



# A Coupled Ground Heat Flux-Surface Energy Balance Model of
# Evaporation Using Thermal Remote Sensing Observations
Devansh Desai[1,11*], Kaniska Mallick[2,10*], Bimal K. Bhattacharya[3], Ganapati S. Bhat[4], Ross
Morrison[5], Jamie Clevery[6], Will Woodgate[7], Jason Beringer[8], Kerry Cawse-Nicholson[9], Siyan
Ma[10], Joseph Verfaillie[10], Dennis Baldocchi[10]
[1]Department of Physics, Electronics & Space Sciences, Gujarat University, Ahmedabad, India
[2]Remote Sensing and Natural Resources Modeling, Department ERIN, Luxembourg Institute of
Science and Technology, Belvaux, L4422, Luxembourg
[3]Agriculture & land Ecosystem Division, Space Applications Center, ISRO, Ahmedabad, India
[4]Centre for Atmosphere and Oceanic Studies, Indian Institute of Sciences, Bengaluru, India
[5]Centre for Ecology and Hydrology, Lancaster, UK
[6]Terrestrial Ecosystem Research Network, College of Science and Engineering, James Cook
University, Cairns, Queensland
[7]CSIRO Land and Water, Private Bag 5, Floreat 6913, Western Australia.
[8]School of Earth and Environment (SEE), The University of Western Australia, WA, 6009,
Australia
[9]Carbon Cycles and Ecosystems, Jet Propulsion Laboratory, California Institute of Technology,
United States
[10]Environemtal Science Policy and Management, University of California, Berkeley, United States
[11]Department of Physics, Institute of Science, Silver Oak University, Ahmedabad, Gujarat, India
*Corresponding authors*: Kaniska Mallick (kaniska.mallick@gmail.com) and Devansh Desai
(ddesai10793@gmail.com)



**Abstract**
The major undetermined problem in evaporation (ET) retrieval using thermal infrared (TIR)
remote sensing is the lack of a physically based ground heat flux (G) model and its amalgamation
with surface energy balance (SEB) model. Here, we present a novel approach based on coupling a
thermal inertia (TI)-based mechanistic G model with an analytical SEB model (Surface
Temperature Initiated Closure) (STIC, version STIC1.2). The coupled model is named as STIC-
TI and it uses noon-night land surface temperature ($T_S$), surface albedo and vegetation index from
MODIS Aqua in conjunction with a clear-sky net radiation model and ancillary meteorological
information. The SEB flux estimates from STIC-TI were evaluated with respect to the *in-situ*
fluxes from Eddy Covariance (EC) measurements in diverse agriculture and natural ecosystems of
contrasting aridity in the northern hemisphere (e.g., India, United States of America) and southern
hemisphere (e.g., Australia). Sensitivity analysis revealed substantial sensitivity of the STIC-TI
derived fluxes due to $T_S$ uncertainty and partial compensation of sensitivity of G to $T_S$ due to the
nature of the equations used in the TI-based G model. An evaluation of STIC-TI G estimates with
respect to *in-situ* measurements showed an error range of 12-21% across six flux tower sites in
both the hemispheres. A comparison of STIC-TI G estimates with other G models revealed
substantially better performance of the former. While the instantaneous noontime net radiation
($R_{Ni}$) and latent heat flux ($LE_i$) was overestimated (15% and 25%), sensible heat flux ($H_i$) was
underestimated with error of 22%. The errors in $G_i$ were associated with the errors in daytime $T_S$
and mismatch of footprint between the model estimates and measurements. Overestimation
(underestimation) of $LE_i$ ($H_i$) was associated with the overestimation of net available energy ($R_{Ni}$
– $G_i$) and use of unclosed SEB measurements. Being independent of any leaf-scale conductance
parameterization and having a coupled sub-model of G, STIC-TI can make valuable contribution
to map and monitor water stress and evaporation in the terrestrial ecosystems using noon-night
thermal infrared observations from existing and future EO missions such as INSAT 4th generation
and TRISHNA.
**Keywords**: Thermal remote sensing, water stress, evaporation, ground heat flux, thermal inertia,
surface energy balance, STIC, terrestrial ecosystem



## 1 Introduction

Ground heat flux (G) is an intrinsic component of the surface energy balance (Sauer and Horton,
2005), affecting the net available energy for evaporation (ET) (the equivalent water depth of latent
heat flux, LE) and sensible heat flux. It represents an energy flow path that couples surface with
atmosphere and has important implications for the underlying thermal regime (Sauer and Horton,
2005). Evaporation is also an integral component of the surface energy balance where water is lost
from and within the soil-vegetation substrate complex through the 'physics of evaporation and
'ecophysiology' of transpiration while regulating the temperature and growth of vegetation (Martel
et al., 2018). Due to complex feedback between the physics of ground heat flux, land-atmosphere
interactions and vegetation ecophysiology, evaporation modelling at different space-time scales
remained a challenging task (Wang et al., 2013; Kiptala et al., 2013). This paper addresses the
challenge of simultaneous estimation of G and ET by combining thermal remote sensing
observations with a mechanistic G model and analytical surface energy balance (SEB) model.
Land surface temperature (LST or $T_S$) retrieved through thermal infrared (TIR) remote sensing
carries imprints of soil water content and is extraordinarily sensitive to evaporative cooling, which
makes it a crucial variable for estimating sensible heat flux (H) ET through the SEB models
(Kustas and Anderson, 2009; Mallick et al., 2014, 2015a, 2018a; Cammalleri and Vogt, 2015;
Anderson et al., 2012). However, it is the aerodynamic temperature ($T_0$) that is responsible for the
sensible heat transfer and the inequality of Ts versus $T_0$ introduces additional uncertainty in ET
retrieval through the SEB models. The differences between Ts and $T_0$ is accommodated either by
using two-source approximation of SEB (Anderson et al., 2012) or through an empirical extra-
resistance in the single-source SEB models (Su, 2002). In the SEB method, $T_S$ represents the lower
boundary condition to estimate both sensible (H) and latent heat fluxes (LE) (Anderson et al.,
2012; Mallick et al., 2014, 2015a, 2018a). SEB models mainly emphasize on estimating H by
resolving the aerodynamic conductance ($g_A$) and resolves LE as a residual SEB component as
follows:

$$LE = R_N - G - H \qquad (1)$$

$R_N$ is the net radiation. The proportion of $R_N$ that is partitioned into conductive heat flux (G)
depends upon soil properties like its albedo, soil moisture, soil thermal properties such as heat





conductance and capacity, which vary with mineral, organic and water fractions. The magnitude
of G varies greatly across different ecosystems from as low as $< 20$ W m$^{-2}$ under dense forest to as
high as 100 W m$^{-2}$ over dry soils in arid and semi-arid landscapes or the rows between crops. In
the humid ecosystems with predominantly dense canopies and high mean fractional vegetation
cover, G contributes to a small proportion in eq. (1). Dense canopy cover leads to less transmission
of downwelling shortwave radiation flux through multiple layers of canopies, which results in low
warming of the soil floor. Due to persistently high soil water content, humid ecosystems generally
show low diurnal and seasonal variability in G. By contrast, the magnitude of G is substantially
large in the arid and semi-arid ecosystems with sparse and open canopy and high water stress. One
of the outstanding challenges in SEB modeling concerns an accurate estimation of G in the open
canopy system such as savanna with mixed vegetation or in ecosystems with low mean fractional
vegetation cover, predominant water stress, and strong seasonality in soil moisture.
While the utility of a surface heat capacity and thermal inertia (TI)-based mechanistic G model
was demonstrated by Murray and Verhoef (2007), Verhoef et al. (2012), and Mallick et al. (2015b);
the potential of an analytical SEB model (Mallick et al., 2014, 2015, 2016, 2018a,b) for mapping
ET in a variety of ecological transects was also demonstrated by Bhattarai et al. (2018, 2019).
Recognizing the significant conclusions of Verhoef et al. (2012), Mallick et al. (2014; 2015a,b;
2016; 2018a,b) and Bhattarai et al. (2018, 2019), there is a need to overcome the challenges of
accurate G estimation and to complement the overarching gaps in SEB modeling in the sparsely
vegetated open canopy systems. Present study coupled the TI-based G model of Murray and
Verhoef (2007), after required modification, with the current version of an analytical ET model,
the Surface Temperature Initiated Closure (STIC, version 1.2; Mallick et al., 2014, 2015a, 2016,
2018a,b) and evaluated this new coupled G-SEB model in different ecosystems of contrasting
aridity.
Remote sensing-based ET models generally use linear and non-linear relationships for estimating
G and such methods generally employ R$_N$, T$_S$, albedo ($\alpha_R$), and NDVI (e.g., Bastiaanssen et al.,
1998; Friedl, 2002; Santanello and Friedl, 2003). While the inclusion of T$_S$ and albedo serves as a
proxy for soil moisture and surface characteristics effects in G, inclusion of NDVI provides a
scaling of G - R$_N$ ratio for different fractional vegetation cover. Unfortunately, all the approaches
are empirical and do not include any information of deep soil temperature or daily temperature



amplitude as lower boundary conditions. These empirical model functions also lack the universal
consensus. Setting G as a fraction of $R_N$ does not solve the energy balance equation and disregards
the role of thermal inertia of the land surface (Mallick et al., 2015b). This could introduce
substantial uncertainty in LE estimation because G effectively couples the surface energy balance
with energy transfer processes in the soil thermal regime. It provides physical feedback to LE
through the effects of soil moisture, temperature, and conductivity (thermal and hydraulic) (Sauer
and Horton, 2005). Such feedbacks are most critical in the arid and semi-arid ecosystems where
LE is significantly constrained by the soil moisture dry-down. The limits imposed on LE by the
water stress consequently result in greater partitioning of the net available energy (i.e., $R_N$ – G)
into H and G (Castelli et al., 1999).
When LE is reduced due to soil moisture dry-down and water stress, both G and $T_S$ tend to show
rapid rise. Therefore, the surface energy balance equation could be linked with mechanistic G
model, $T_S$ harmonics (Verhoef, 2004), and soil moisture availability. Realizing the importance of
direct estimates of G in LE and invigorated by the advent of TIR remote sensing, Verhoef et al.,
(2012) demonstrated the potential of a TI-based mechanistic model (Murray and Verhoef, 2007)
(MV2007 hereafter) for spatio-temporal G estimates in the semi-arid ecosystems of Africa. Some
studies also emphasized the importance of using day-night $T_S$ and $R_N$ for estimating G (Mallick et
al., 2015b; Bennet et al., 2008; Tsuang, 2005). The method of MV2007 has so far been tested in a
stand-alone mode, and no remote sensing method is so far attempted to combine such a mechanistic
G model (e.g., MV2007-TI model) with a SEB model for coupled energy-water flux estimation
and validation.
By integrating $T_S$ into a combined structure of the Penman-Monteith (PM) and Shuttleworth-
Wallace (SW) model, an analytical SEB modeling was proposed by Mallick et al., (2014, 2015a,
2016). The model, Surface Temperature Initiated Closure (STIC), is based on finding analytical
solution for aerodynamic and canopy-surface conductance ($g_A$ and $g_S$) where the expressions of
the conductances were constrained with an aggregated water stress factor. Through physically
linking water stress (Ts derived) with $g_A$ and $g_S$, STIC established a direct feedback between $T_S$,
H and LE, and simultaneously overcame the need of empirical parameterization for estimating the
conductances (Mallick et al., 2016, 2018a). Different versions of STIC have been extensively
validated in different ecological transects (Tropical rainforest to woody savanna) and aridity



gradients (humid to arid) (Trebs et al., 2021; Bai et al., 2021; Mallick et al., 2015a; 2016; 2018a,
b; Bhattarai et al., 2018, 2019). Realizing the significance of mechanistic G model (MV2007) and
the advantage of analytical solution for different turbulent heat fluxes and conductances from the
STIC model, this paper presents the first-ever coupled implementation of MV2007 G with the
most recent version of STIC (STIC1.2). We name this new coupled model as STIC-TI and it
requires day-night Ts and associated remotely sensed land surface variables as inputs. We
performed subsequent evaluation of STIC-TI in nine terrestrial ecosystems in arid, semi-arid and
sub-humid climate in India, the United States of America (USA) (representing northern
hemisphere) and Australia (representing southern hemisphere) at the eddy covariance flux tower
sites. The current study addresses the following research questions and objectives:
(i)   What is the performance of STIC-TI G estimates when compared with contemporary empirical

models in ecosystems having low mean fractional vegetation cover ($f_c$) ($\leq 0.5$) and having larger

soil exposure to radiation for example in Savanna?

(ii)  How do the estimates from STIC-TI LE and H fluxes compare with LE and H observations in

diverse terrestrial ecosystems that represent a varied range of $f_c$ (0.25 – 0.5) covering cropland,

savanna, mulga vegetation spread across arid, semi-arid, sub-humid, humid climates over a vast

range of rainfall (250 to 1730 mm), temperature (-4 to 46°C) and soil regimes?

(iii) What is the seasonal variability of G and evaporative fraction from STIC-TI model in a wide

range of ecosystems having contrasting aridity and vegetation cover?

It is important to mention that assessing the performance of STIC-TI LE and H with respect to
other SEB models is not within the scope of the present study. The prime focus of the current study
is to assess the sensitivity of STIC-TI, temporal variability of the retrieved SEB fluxes, and cross-
site validation of the individual SEB components.
A list of variables, their symbols and corresponding units are given in Table A1 in Appendix A.
**2 Study area and datasets**
**2.1 Study site characteristics**
The present study was conducted at nine flux tower sites (four sites in India; three sites in Australia;
two sites in USA) equipped with Eddy Covariance (EC) measurement systems. The distribution
of the flux tower sites considered for the present study are shown in Fig. 1 below. The sites cover
a wide range of climate, vegetation types, low fractional vegetation cover ($f_c$) of around 0.5 and
have contrasting aridity (Table 1). In India, a network of EC towers was set up under Indo-UK
INCOMPASS (INteraction of Convective Organization and Monsoon Precipitation, Atmosphere,
Surface and Sea) Program (Turner et al., 2019) at Jaisalmer (IND-Jai) in Rajasthan state, Nawagam
(IND-Naw) in Gujarat state, Samastipur (IND-Sam) in Bihar state and under Newton-Bhaba
programme (Morisson et al., 2019 a,b) at Dharwad (IND-Dha) in Karnataka state. The fetch ratio
of EC towers in India varied from 1:50 to 1:100 representing 90% of fetch area. The mean annual
$f_c$ was found to vary from 0.25 to 0.52 with standard deviation (SD) ranging from 0.1 to 0.16.
The IND-Jai site represents arid western zone over desert plains of natural grassland ecosystem.
The region receives very low rainfall (100 – 300 mm) during monsoon and experiences a wide
range in air temperature, high solar radiation, wind speed and high evaporative demand (Raja et
al., 2015). The IND-Naw site represents semi-arid agroecosystem in the middle Gujarat agro-
climatic zone of north-west India and has a pre-dominant rice-wheat cropping system. The IND-
Sam site has sub-humid climate of north-west alluvial plain zone in the Indo-Gangetic Plain (IGP)
situated in the eastern India and this site also follows rice-wheat crop rotation. IND-Dha represents
humid sub-tropical climate of transition zone in the southern India and this site comprises of crops.

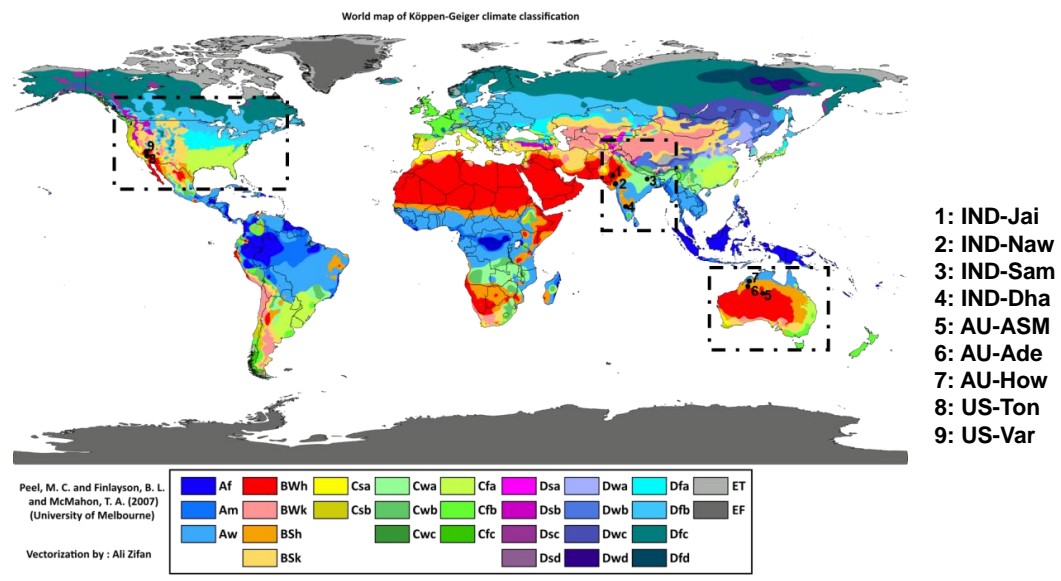





**Figure 1**: Locations of the flux tower sites in India, Australia and USA overlaid on climate type map. (Image Source: By Peel, M. C., Finlayson, B. L., and McMahon, T. A. (University of Melbourne) enhanced, modified, and vectorized by Ali Zifan; Hydrology and Earth System Sciences: "Updated world map of the Köppen-Geiger climate classification" (Supplement) map in PDF (Institute for Veterinary Public Health). Legend explanation, CC BY-SA 4.0, https://commons.wikimedia.org/w/index.php?curid=47086879)

In USA, two EC tower sites were located at Tonzi Ranch (US-Ton) and Vaira Ranch (US-Var), in
the lower foothills of the Sierra Nevada Mountains. Both the EC stations are part of the
AMERIFLUX Management Project (https://ameriflux.lbl.gov/). US-Ton is classified as an oak
savanna woodland on privately owned land. While the overstorey is dominated by blue oak trees
(40% of total vegetation) with intermittent grey pine trees (3 trees ha$^{-1}$), the understory species
include a variety of grasses and herbs. The mean annual rainfall at this site is 559 mm. US-Var is
a grassland dominated site and the growing season is confined to the wet season only, typically
from October to early May. The mean annual rainfall at this site is 559 mm. The mean annual $f_c$
was found to vary from 0.18 to 0.26 and SD of the order of 0.06 to 0.07.
In Australia, three EC tower sites were located at Howard Springs (AU-How), Alice Springs
Mulga (AU-ASM), Adelaide river (AU-Ade) in the Northern Territory as part of the OzFlux
network (Beringer et al., 2016) and the Terrestrial Ecosystem Research Network (TERN), which
is supported by the National Collaborative Infrastructure Strategy (NCRIS)
(http://www.ozflux.org.au/monitoringsites/index.html). The AU-How is situated in the Black
Jungle Conservation Reserve representing an open woodland savanna and the mean annual rainfall
is 1750 mm. The AU-ASM is located on Pine Hill cattle station near Alice Springs. The woodland
is characterized by mulga canopy and mean annual rainfall is 306 mm. AU-Ade represents savanna
with a mean annual rainfall of 1730 mm. The mean annual $f_c$ varied from 0.21 to 0.48 having SD
range of 0.08 - 0.17. A description of Australian flux sites is given in Beringer et al. (2016).












**Table 1:** An overview of the EC flux tower site characteristics in the present study

| Hemisphere | Sites | Latitude (°N), Longitude (°E) | Climate & Vegetation | Mean $f_c$ (SD) | Soil texture | $T_A$ range (°C) | Mean Annual P (mm) | Observation period |
|---|---|---|---|---|---|---|---|---|
| Northern | Jaisalmer (IND-Jai) | 26.99, 71.34 | Arid grassland | 0.25(±0.1) | Loamy fine sand to coarse sand | 8 – 40 | 250 | 2017 – 2018 |
| | Nawagam (IND-Naw) | 22.80, 72.57 | Semi-arid cropland | 0.41(±0.13) | Sandy loam | 9 – 39 | 700 | 2017 – 2018 |
| | Samastipur (IND-Sam) | 26.00, 85.67 | Humid subtropical cropland | 0.52(±0.16) | Sandy loam to loam | 10 – 39 | 1000 | 2017 – 2018 |
| | Dharwad (IND-Dha) | 15.50, 74.99 | Tropical Savanna | 0.36(±0.11) | Shallow to medium black clay and red sandy loam soils | 12 – 40 | 650 | 2016 – 2018 |
| | Tonzi ranch (US-Ton) | 38.43, -120.96 | Woody Savanna | 0.18(±0.06) | Red sandy clay loam | 0 – 40 | 559 | 2011 – 2019 |
| | Vaira ranch (US-Var) | 38.41, -120.95 | Arid grassland | 0.26(±0.07) | Rocky silt loam | 0 – 40 | 559 | 2011 – 2019 |
| Southern | Alice Springs Mulga (AU-ASM) | 22.28, 133.24 | Semi-arid mulga | 0.21(±0.09) | Loamy sand | (-4) – 40 | 305 | 2011 – 2014 |
| | Howard Springs (AU-How) | 12.49, 131.15 | Tropical savanna | 0.48(±0.17) | Red kandasol | 19 – 34 | 1700 | 2011 – 2014 |
| | Adelaide River (AU-Ade) | 13.07, 131.11 | Savanna | 0.42(±0.08) | Yellow hydrosol, shallow, loamy sand with coarse gravel | 16 – 37 | 1730 | 2007 – 2009 |

$T_A$: Air temperature during the observation period; P: rainfall (mm) measured using rain gauge at flux tower site during the study
period. IND is for India, AU is for Australia, and US is for the United States; SD is standard deviation of annual mean fc which is
computed from NDVI as mentioned in section 3.1.





### 2.2 Datasets

### 2.2.1 Micrometeorological data at flux tower sites

Standardized, controlled and harmonized surface energy balance (SEB) flux and meteorological data from nine EC towers were used in the present analysis. In Australia, the SEB measurements were carried out at varying heights of 15 m, 23 m and 11.6 m at AU-Ade, AU-How and AU-ASM, respectively. In India, the EC measurement height was maintained approximately at 8 m above the surface, except at IND-Dha where it was installed at a height of 4.2 m. In USA, the SEB measurements were carried out at tower heights of 23 m at US-Ton and 2 m US-Var. A summary of the instrumentation is given in Table A2 of appendix A. All the flux tower sites were equipped with a range of meteorological instrumentation which measured diurnal air temperature ($T_A$) and relative humidity ($R_H$), four components of the net radiation ($R_N$, consisting of down- and up-welling shortwave and long-wave radiation (SW↓, SW↑, LW↑ and LW↓, respectively)) above the vegetated canopy. In addition, the diurnal soil heat flux (G) and soil temperature ($T_{ST}$) were measured at all the three Australian sites and two US sites. In India, the diurnal soil heat flux was measured only at IND-Dha.

For the Indian sites, the raw EC measurements of the turbulent wind vectors ($u$, $v$ and $w$, for horizontal, meridional and vertical, respectively), sonic temperature (T), and $CO_2$ and water vapor mass density were recorded at a sampling rate of 20 Hz. Raw EC data were post-processed to obtain level-3 quality controlled and harmonized surface fluxes at 30-minute flux averaging intervals using EddyPRO® Flux Calculation Software (LI-COR Biosciences, Lincoln, Nebraska, USA) using the data handling protocol described by Bhat et al. (2019). The EC data from the OzFlux sites was averaged over 30 minutes recorded by the logger and processed through levels using the PyFluxPro standard software processing scripts as mentioned in Isaac et al. (2017). The Level 3 (L3) used in this paper was produced using PyFluxPro (Isaac et al., 2017) employing the Dynamic INtegrated Gap filling and partitioning for Ozflux (DINGO) system as described in Donohue et al. (2014) and Beringer et al. (2016). The quality checked EC data at 30 minute intervals for two AMERIFLUX sites US-Ton and US-Var was acquired from _https://doi.org/10.17190/AMF/1245971&_ _https://doi.org/10.17190/AMF/1245984_, respectively.





**2.2.2 Remote sensing data**
Optical and thermal remote sensing observations available from Moderate Resolution Imaging
Spectroradiometer (MODIS) (Didan et al., 2015) on-board Aqua platform were used in the present
analysis (Table 2) for estimating G and associated SEB fluxes. These include land surface products
(eight-day) of noon-night land surface temperature (LST or $T_S$) and surface emissivity ($\varepsilon_s$)
(MYD11A2), daily surface albedo ($\alpha_R$) (MCD43A3), 16-day NDVI (MYD13A2). The overpass
times of MODIS Aqua are at 1:30 pm and 1:30 am (IST). The noon-night pair of thermal remote
sensing observations from Aqua are close to time of occurrences of maximum and minimum soil
surface temperature (see Figure 2) and are therefore ideal for soil heat flux modeling using thermal
inertia. The MODIS Terra overpass times are at 11 AM and 11 PM and are quite away from time
of occurrences of minimum-maximum soil temperatures. Therefore, MODIS Aqua acquisition
times were used.
**Table 2:** A summary of MODIS Aqua optical and thermal remote sensing products used in the
present study

| Data type | Product ID (version) | Variables used | Spatial resolution (m) | Temporal resolution | Purpose | Inputs to equation numbers |
|---|---|---|---|---|---|---|
| Land surface temperature and emissivity | MYD11A2 (V006) | $T_S$ and $\varepsilon_s$ | 923 | 8-day | For estimating $R_{Ni}$, $G_i$, $LE_{i,}$ $H_i$ | (5), (13), (C6), (C7), (B8) |
| Surface albedo | MCD43A3 (V006) | $\alpha_R$ | 462 | 8-day composite from daily | For estimating $R_{Ni}$, $G_i$ | (5), (B3) |
| Vegetation index | MYD13Q1 (V006) | NDVI | 250 | 16-day | For estimating $G_i$ | (4) |

The key variables of SEB modeling such as LST and $\varepsilon_s$, were retrieved at 923m spatial resolution
from MODIS Aqua noon-night thermal infrared (TIR) observations (MYD11A2) in bands 11.03
µm and 12.02 µm using a generalized split-window algorithm by Wan et al., (2015). The land





surface emissivity was estimated from land cover types, atmospheric column water vapor and
lower boundary air surface temperature that are separated into tractable sub-ranges for optimal
retrieval. The albedo was estimated from MODIS (MCD43A2 Version 6) Bidirectional
Reflectance Distribution Function and Albedo (BRDF/Albedo) daily dataset (Schaaf et al., (2002))
at 462 m spatial resolution. Eight-day compositing for albedo was done from daily products
(MYD11A2). NDVI was estimated from MODIS Vegetation Indices (MYD13Q1) Version 6 data
and are generated every 16-day at 250 meter (m) spatial resolution as a Level 3 product.
MYD13Q1 contains Normalized Difference Vegetation Index (NDVI) and Enhanced Vegetation
Index (EVI). In the present study, NDVI has been used because of its universal applicability (Xue
and Su, 2017; Drori et al. 2020; Bhandari et al., 2012). All the input remote sensing variables
mentioned in table 2 are resampled to spatial resolution of MYD11A2 (V006) product (923 m).
**3 Methodology**
**3.1 Coupled soil heat flux-SEB model**
In this paper, we modified a thermal inertia (TI) based soil heat flux (G) model using noon-night
thermal remote sensing observations and thereafter coupled the TI-based G with STIC1.2. A clear-
sky net radiation ($R_N$) model was also introduced into this coupled model and $R_N$ estimation
algorithm is described in Appendix B. The estimation of G through modifying MV2007-TI
approach and its coupling with STIC1.2 is the most novel component of the modeling scheme, and
it is therefore described in the main body of the paper (section 3.1.1). Such a coupling enabled the
implementation of a mechanistic G model along with an analytical SEB model using optical-
thermal remote sensing data. The coupled model is hereafter referred as STIC-TI. The noteworthy
features of STIC-TI are: (1) estimating G by modifying the mechanistic MV2007-TI model using
noon-night $T_S$ data from thermal remote sensing observations available through polar orbiting
satellite platform (e.g. MODIS Aqua), (2) coupling MV2007-TI G model with STIC1.2 to
simultaneously estimate surface moisture availability (M), G, and SEB fluxes, (3) introducing
moisture availability information in G to better constrain the aerodynamic and canopy-surface
conductances as well as the SEB fluxes, (4) the G model uses fundamental soil physical properties,
moisture constants and soil texture that majorly influence soil heat conduction, (5) derivation of
amplitude of ecosystem-scale surface soil temperature (from top soil to 0.1 m soil depth).





### 3.1.1 MV2007 soil heat flux model based on Thermal Inertia (TI)

The functional form for estimating instantaneous G ($G_i$, hereafter) (eq. 2 below) is based on the harmonic analysis of soil surface temperature and is described in detail by Murray and Verhoef (2007) and Maltese et al. (2013).

$$G_i = \Gamma\left[(1 - 0.5f_C)\left(\sum_{n=1}^{k} A\sqrt{n\omega}\sin\left(n\omega t + \phi'_n + \frac{\pi}{4} - \frac{\pi\Delta t}{12}\right)\right)\right] = \Gamma J_S \qquad (2)$$

$G_i$ is the soil heat flux at the surface at a particular instance (W m$^{-2}$), $\Gamma$ is the soil thermal inertia (J m$^{-2}$ K$^{-1}$ s$^{-0.5}$), k is the total number of harmonics used, A is the amplitude (°C) of the n$^{th}$ soil surface temperature ($T_{ST}$) harmonic, $\omega$ is the angular frequency (rads$^{-1}$), t is the time (s), $\phi'_n$ is the phase shift of the n$^{th}$ soil surface temperature harmonic (rad), $J_S$ is the summation of harmonic terms of soil surface temperature (K), and $\Delta t$(s) is time offset between the canopy composite temperature and the below-canopy soil surface temperature. Here, we represent $G_i$ and A as ecosystem-scale (≤ 1km) soil heat flux and surface soil temperature amplitude (within 0.1 m from the soil top), respectively and assume it to be valid for different vegetated landscape.

Since we have considered a single pair (noon-night corresponding to 1 pm and 1 am) of MODIS aqua LST data in the present study, the phase shift ($\phi'_n$) is taken as zero and number of harmonics is taken as one (k=1) for estimating noontime $G_i$. Thus equation (2) is modified as follows:

$$G_i = \Gamma\left[(1 - 0.5f_C)\left(A\sqrt{\omega}\sin\left(\omega t' + \frac{\pi}{4} - \frac{\pi\Delta t}{12}\right)\right)\right] = \Gamma J_S \qquad (3)$$

$\Delta t$(s) is found to be 1.5 h (Murray and Verhoef, 2007). With the two boundary values (i.e., $\Delta t$ =1.5 h for $f_c$ = 1 and $\Delta t$ = 0 for $f_c$ = 0), a linear approach is proposed here to describe the time offset $\Delta t$ as a function of vegetation fraction ($f_c$) (Murray and Verhoef, 2007; Maltese et al., 2013). The $f_c$ was derived from NDVI on a given day or period and its practically occurring upper-lower limits obtained from annual cycle.





$$\Delta t = 1.5\, f_c \qquad\qquad (4)$$

*3.1.1.1 Scaling function for estimating ecosystem-scale surface soil temperature amplitude (A)*
Estimating ecosystem-scale A involves two steps, (a) computing point-scale soil surface
temperature amplitude (from surface to 0.1m depth) ($T_{STA,}$ hereafter) from the available
measurements of soil surface temperature, and (b) linking $T_{STA}$ with remote sensing variables to
develop scaling functions for A.
Several studies suggested theoretical sinusoidal trajectory of soil surface and sub-surface
temperatures (Gao et al., 2010), where the amplitude is maximum at the surface and it gradually
decreases with depth to become close to zero until the damping depth where soil temperature is
almost invariant through day-night called deep soil temperature. However, the diurnal surface soil
temperature measurements (within top 0.1 m depth) across different flux tower sites showed a
sinusoidal-exponential behavior, i.e. sinusoidal pattern from sunrise until the afternoon and
exponential pattern from afternoon through sunset to the next sunrise. An illustrative example of
the theoretical and observed trajectories of surface soil temperature is shown in Fig. 2. This diurnal
surface soil temperature variation has a single harmonic component (Gao et al., 2010). For
computing $T_{STA}$, theoretical half-curve of sinusoidal pattern is assumed and was derived from
measurements as exemplified in Fig 2.



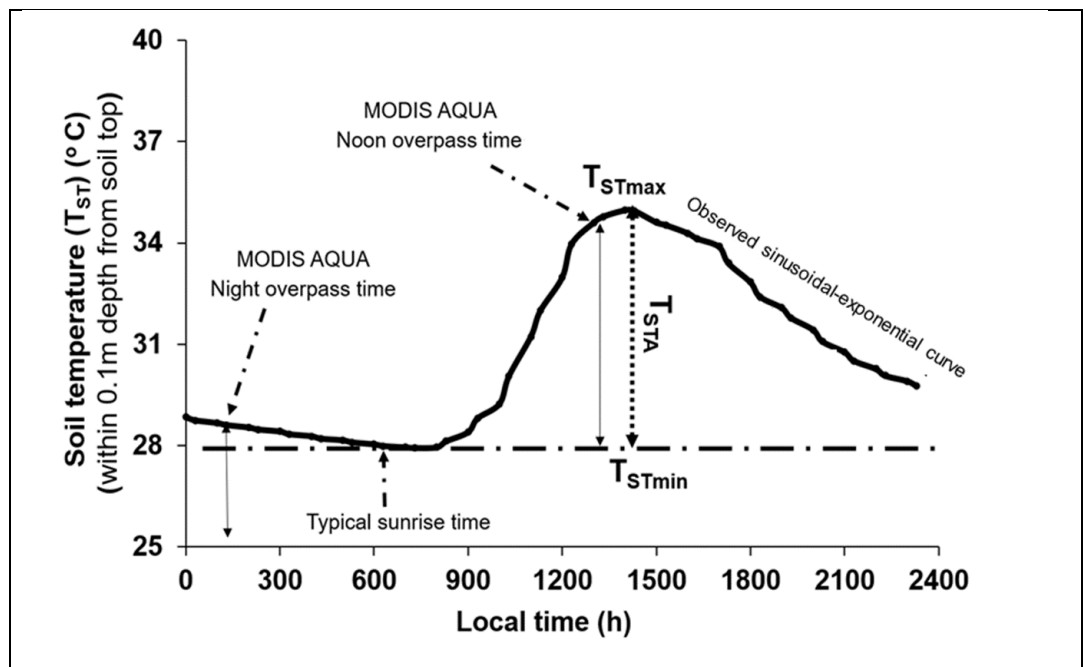

**Figure 2**. An illustrative example of typical diurnal variation of soil temperature ($T_{ST}$) (from surface to 0.1m depth) and timings of MODIS AQUA observations. Here, $T_{STmax}$ and $T_{STmin}$ are maximum and minimum point-scale soil surface temperatures


It is evident from Fig. 2 that $T_{STmin}$ represents minimum surface soil temperature occurring 1-1.5
h after sunrise and $T_{STmax}$ occurs during 12.30 – 15.00 h local time. $T_{STmin}$ is thus close to deep soil
temperature as well as minimum soil temperature of other sub-surface soil layers. Both $T_{STmin}$ and
$T_{STmax}$ represent lower and upper limits of surface soil temperature on a given day and also lower
and upper boundary conditions of soil heat flux conducting through topsoil at noontime. The *in-*
*situ* measured $T_{ST}$ on completely clear-sky days at OzFlux sites were used to extract $T_{STmax}$ and
$T_{STmin}$. The $T_{STA}$ was derived as the difference between $T_{STmax}$ and $T_{STmin}$ from the theoretical half-
curve of sinusoidal pattern.
$T_{STA}$ is generally influenced by several land surface characteristics such as surface temperature
and surface albedo of soil-canopy complex, surface heat capacities, fractional canopy cover and
thermal conductivity (White, 2013). $T_S$ and $\alpha_R$ are the major thermal and reflective land surface
properties that have strong synergy with surface soil temperature dynamics. Hence, we have used
bivariate regression analysis to develop a scaling function for estimating ecosystem-scale $T_{STA}$
(top to 0.1m depth). The bivariate regression is based on the difference of noon (d) and night (n)





$T_S$ data and $\alpha_R$ (Duan et al., 2013, Li Tian et al., 2014) from MODIS Aqua. The scaling function
given in eq. (5) estimates ecosystem-scale $T_{STA}$ (symbolized as 'A' in equation 5) from surface to
0.1 m soil depth:

$$A = B_1(T_{Sd} - T_{Sn}) + B_2(\alpha_R) + B_3 \tag{5}$$

Here, B1, B2, B3 are coefficients of regression model; $T_{Sd}$ and $T_{Sn}$ are noon and nighttime LST,
respectively. The results of this regression analysis are elaborated in section 4.1.

### 3.1.1.2 Estimating $\Gamma$

$\Gamma$ is the key variable for estimating $G_i$ using eq. (2). MV2007 adopted the concept of normalized
thermal conductivity (Johansen, 1975) and developed a physical method to estimate $\Gamma$ as follows:

$$\Gamma = e^{\left[\Upsilon'\left(1 - S_r^{(\Upsilon' - \delta)}\right)\right]}(\tau_* - \tau_0) + \tau_0 \tag{6}$$

where $\tau_*$ and $\tau_0$ are the thermal inertia for saturated and air-dry soil (J m$^{-2}$K$^{-1}$s$^{-0.5}$); $\tau_0 = D_1\theta_* + D_2$;
$\tau_* = D_3$ $(\theta_*^{-1.29})$; $\Upsilon'$ $(-)$ is a parameter depending on the soil texture (Murray and Verhoef, 2007;
Minasny, 2007; Anderson et al., 2007); $S_r$ (m$^3$ m$^{-3}$) is relative saturation and is equal to $(\theta/\theta_*)$; $\delta$
(unitless) is the shape parameter which is dependent on the soil texture. $\theta_*$ (m$^3$ m$^{-3}$) is the soil
porosity (equal to the saturated soil moisture content when soil moisture suction is zero), $\theta$ (cm$^3$
cm$^{-3}$) is the volumetric soil moisture and $D_1$, $D_2$, $D_3$ are coefficients which were derived from a
large number of experimental data. The reported global values of $D_1$, $D_2$, and $D_3$ were taken as -
1062.4, 1010.8, 788.2, respectively (Maltese et al., 2013). The value for $\theta_*$ and shape parameter
for soil textures across study sites were specified according to Van Genuchten et al. (1980). The
details are mentioned in Table E1 of Appendix E.
In the present study, the relative soil moisture saturation, $S_r$ $(\theta/\theta_*)$ is represented in terms of an
aggregated moisture availability (M) of canopy-soil complex through a linear function (eq. 12). In
case of zero canopy cover, M represents the soil moisture availability from surface to 0.1 m depth.
In sparse and open canopy, rates of moisture availability from soil to root and root to canopy were
assumed same.





Theoretically, M is expressed as available soil moisture fraction between field capacity ($\theta_{fc}$) and
permanent wilting ($\theta_{wp}$) point as given in eq. (7) below.

$$M = \frac{\theta - \theta_{wp}}{\theta_{fc} - \theta_{wp}} \tag{7}$$

Where, $\theta_{fc}$ ($m^3\ m^{-3}$) is the volumetric soil moisture at the field capacity (at a suction of 330 hpa)
and $\theta_{wp}$ ($m^3\ m^{-3}$) is the volumetric soil moisture at the permanent wilting point (at suction of 15000
hpa) (Singh, 2007). Since $\theta_{fc}$, $\theta_*$, $\theta_{wp}$ are soil moisture constants and depends on the soil texture,
dividing the numerator and denominator in eq. (7) by $\theta_*$ gives the following expression:

$$M = \frac{\dfrac{\theta}{\theta_*} - \dfrac{\theta_{wp}}{\theta_*}}{\dfrac{\theta_{fc}}{\theta_*} - \dfrac{\theta_{wp}}{\theta_*}} \tag{8}$$

Due to their dependence on soil texture, the ratios ($\theta_{fc}/\theta_*$) and ($\theta_{wp}/\theta_*$) are treated as constants.
These are represented as C and C′ in the later equations (eq. 9, 10, and 11). The constants, C and
C′ vary from 0.3 to 0.8 and from 0.1 to 0.4 (Murray and Verhoef, 2007; Minasny et al., 2011;
Anderson et al., 2007), respectively over different soil textures.

$$M = \frac{\dfrac{\theta}{\theta_*} - C'}{C - C'} \tag{9}$$

$$M(C - C') = \left(\frac{\theta}{\theta_*}\right) - C' \tag{10}$$

By replacing $S_r$ in eq. (6) as $\theta/\theta_*$ and by rearranging eq. (10), the following linear function is
obtained.

$$S_r = \frac{\theta}{\theta_*} = M\,(C - C') + \ C' = M' \tag{11}$$

Thus, the modified equation to calculate $\Gamma$ is given by eq. (12) as follows:

$$\Gamma = e^{\left[\Upsilon'\left(1 - M'^{(\Upsilon' - \delta)}\right)\right]}(\tau_* - \tau_0) + \tau_0 \tag{12}$$





By substituting the values obtained from eq. (4), (5) and (12) into eq. (3), we obtained the
instantaneous ecosystem-scale $G_i$ corresponding to MODIS Aqua noontime overpass. The intrinsic
link between $G_i$ estimates through MV2007-TI and SEB scheme in STIC1.2 is made through M,
where the computation of M follows the procedure as described in Mallick et al. (2016, 2018a, b)
and Bhattarai et al. (2018). (description in Appendix C).

### 393 *3.1.1.3 Estimating M*

In STIC1.2, an aggregated moisture availability (M) of canopy-soil complex is expressed as the
ratio of the 'vapor pressure difference' between the aerodynamic roughness height of the canopy
(i.e., source/sink height) and air to the 'vapor pressure deficit' between aerodynamic roughness
height to the atmosphere:

$$M = \frac{(e_0 - e_A)}{(e_0^* - e_A)} = \frac{(e_0 - e_A)}{\kappa(e_S^* - e_A)} = \frac{s_1(T_{0D} - T_D)}{\kappa s_2(T_S - T_D)} \tag{13}$$

Where $e_0$ and $e_0^*$ are the actual and saturation vapor pressure at the source/sink height; $e_A$ is the
atmospheric vapor pressure; $e_S^*$ is the saturation vapor pressure at the surface; $T_{0D}$ is dew point
temperature at the source/sink height; $T_S$ is the LST; $T_D$ is the air dew point temperature; $s_1$ and $s_2$
are the psychrometric slopes of the saturation vapor pressure and temperature between $(T_{0D} - T_D)$
versus $(e_0 - e_A)$ and $(T_S - T_D)$ versus $(e_S^* - e_A)$ relationship; and $\kappa$ is the ratio between $(e_0^* - e_A)$
and $(e_S^* - e_A)$. To solve the eq. (13), estimation of $T_{0D}$ is necessary. An initial estimate of $T_{0D}$ [$T_{0D}$
$= [(e_S^* - e_A) - s_3 T_S + s_1 T_D]/(s_1 - s_3)]$ and M were obtained following Venturini et al. (2008) where
$s_1$ and $s_3$ were approximated in $T_D$ and $T_S$, respectively. However, eq. (13) cannot be directly
solved because there are two unknowns in one equation. However, since $T_{0D}$ also depends on LE
(Mallick et al., 2016, 2018a), an iterative updation of $T_{0D}$ (and M) was carried out by expressing
$T_{0D}$ as a function of LE [$T_{0D} = T_D + (\gamma LE/\rho c_p g_A s_1)$] which is described in detail by Mallick et al.
(2016, 2018a) and Bhattarai et al. (2018). In the numerical iteration, $s_1$ was not updated to avoid
numerical instability and it was expressed as a function of $T_D$.

### 411 **3.1.2 STIC-TI: Coupling modified MV2007-TI and STIC 1.2**

The initiation of the coupling between MV2007-TI and STIC1.2 was executed through linking $G_i$
estimates from the modified MV2007-TI with M estimates from STIC1.2. Having the initial




estimates of M (through eq. 13), an initial estimation of $G_i$ was made from eq. (2) where $S_r$ in eq.
11 was replaced with the initial estimates of $M'$. Given the initial estimates of $G_i$ (eq. 2) and $R_{Ni}$
(equations in Appendix B), initial estimation of the conductances, $LE_i$ and $H_i$ were obtained. The
process was then iterated by updating $T_{0D}$ [$T_{0D} = T_D + (\gamma LE /\rho c_p g_A s_1)$] and M in every time step
(as mentioned in Mallick et al., 2016, 2018a), and re-estimating $G_i$ (using eq. 3), net available
energy ($R_{Ni} - G_i$), conductances, $LE_i$ and $H_i$, until stable estimates of $LE_i$ were obtained. The
conceptual block diagram and algorithm flow of STIC-TI is shown in Fig. 3a and Fig 3b,
respectively.












(a)

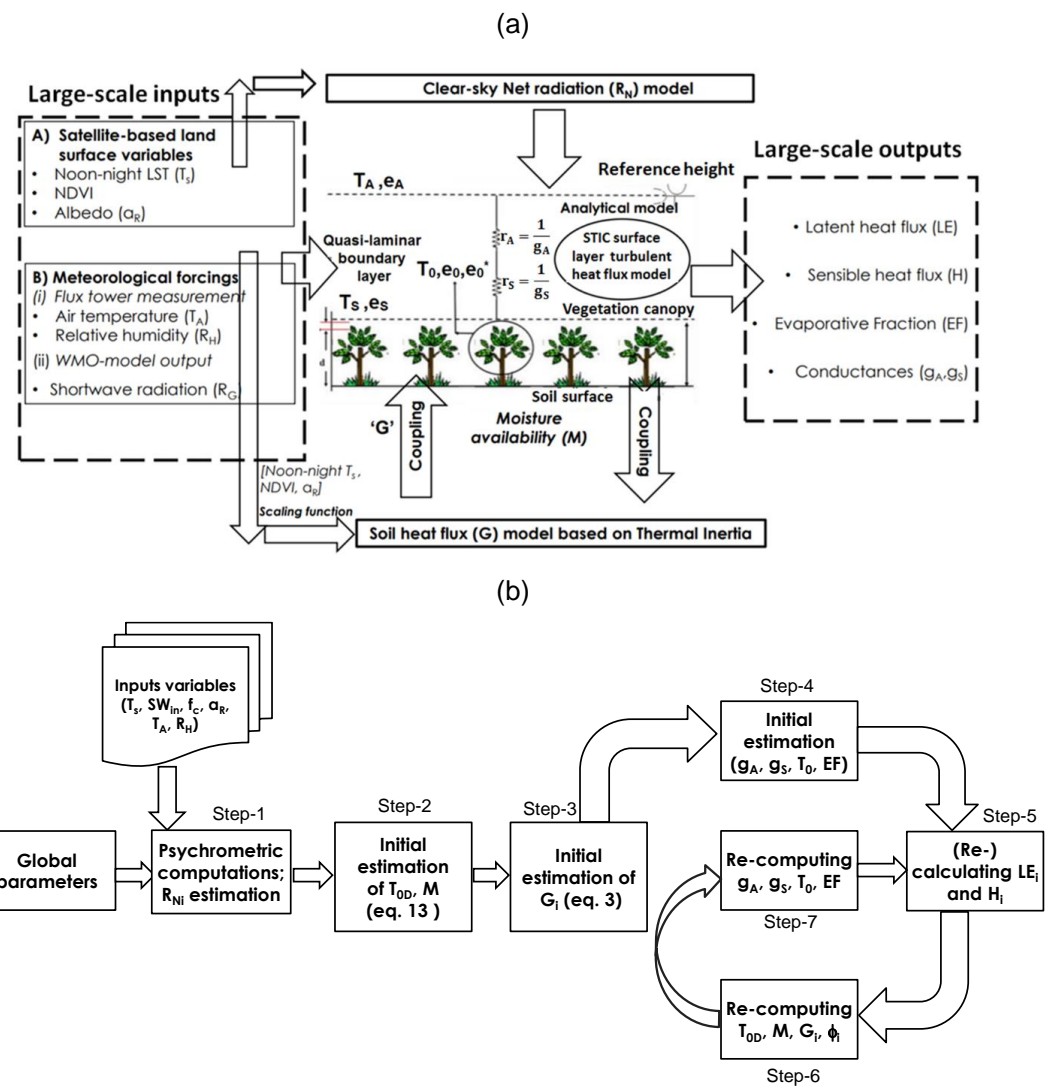

(b)

**Figure 3**: (a) Conceptual diagram of STIC-TI model showing different input variables and model outputs; (b) Algorithmic flow for estimating G and associated SEB fluxes through STIC-TI.

Examples of iterative stabilization of $G_i$ and $LE_i$ for Indian, Australian and US ecosystems of India
are shown in Fig. 4. The iterative stabilization of $G_i$ and $LE_i$ was obtained between 8-25 iterations
for all sites.





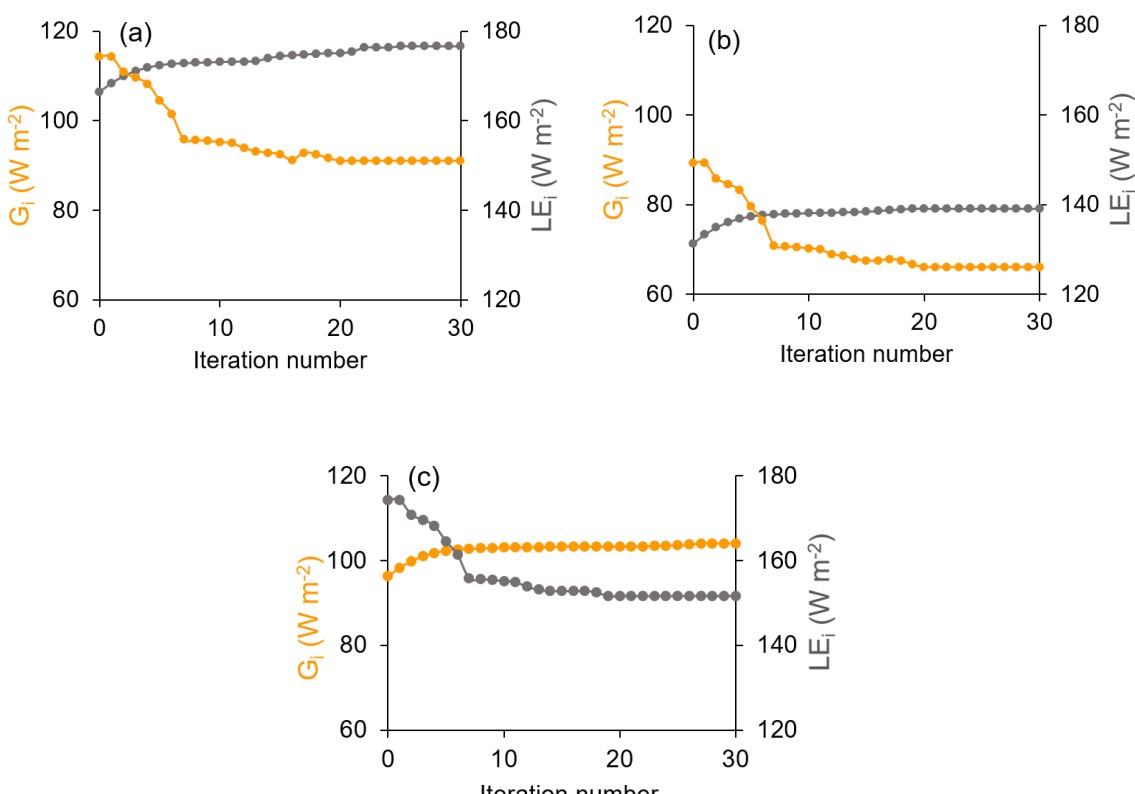

**Figure 4**: Illustrative examples of iterative stabilization of STIC-TI $G_i$ (yellow marker line) and $LE_i$ (grey marker line) in (a) IND-Jai, (b) AU-ASM, (c) US-Ton

**3.2 Sensitivity and statistical analysis**
The accuracy of STIC-TI heavily depends on the accuracy of $T_S$, NDVI, and $\alpha_R$ due to the dual
role of $T_S$ in estimating M and $G_i$, the role of NDVI in $G_i$, and the combined role of $T_S$ and $\alpha_R$ in
estimating $R_{Ni}$. Therefore, one-dimensional sensitivity analysis was conducted to assess the
impacts of uncertainty in $T_S$, NDVI and $\alpha_R$ on $G_i$, $H_i$ and $LE_i$. The sensitivity was assessed by
varying noon-time $T_S$ by $\pm0.5$ K, $\pm1.5$ K and $\pm1.5$ K (keeping nighttime $T_S$ constant so that
amplitude can vary automatically); varying NDVI by $\pm0.05$; $\pm0.10$, $\pm0.15$; and varying albedo by
$\pm0.02$, $\pm0.05$, $\pm0.10$, respectively. SEB fluxes were computed by using $T_S$, NDVI, and $\alpha_R$ for three
different periods of the year in all the eight ecosystems. Sensitivity analyses were conducted by
increasing and decreasing systematically $T_S$, NDVI, $\alpha_R$ from its central value while keeping the
other variables and parameters constant. This procedure was selected because the fluxes and



intermediate outputs of the STIC-TI model reflect an integrated effect due to uncertainty in $T_S$. In
the first run, SEB fluxes were computed using *in-situ* $T_S$ measurements obtained from the flux
tower outgoing longwave radiation measurements. Then $T_S$ was increased and decreased at
constant interval and a new set of fluxes were estimated. In the similar way, $\alpha_R$ and NDVI were
increased and decreased at constant intervals and new set of fluxes were computed. The sensitivity
of STIC-TI was assessed by the equation 14.

$$\text{Sensitivity} = \frac{E_{i0} - E_{iM}}{O_i} * 100 \tag{14}$$

$E_{i0}$ is the estimated (original) model output and $E_{iM}$ is the estimated (modified) output obtained by
changing the variable whose sensitivity is to be tested. $O_i$ is actual measurements. Apart from the
sensitivity analysis, the following set of statistical metrics were used to assess model performances.

$$R^2 = \left( \frac{\sum_{i=1}^{n}(E_i - \bar{E})(O_i - \bar{O})}{\sqrt{\sum_{i=1}^{n}(E_i - \bar{E})^2}\sqrt{\sum_{i=1}^{n}(O_i - \bar{O})^2}} \right)^2 \tag{15}$$

$$\text{RMSE} = \sqrt{\sum_{i=1}^{n}\frac{(E_i - O_i)^2}{n}} \tag{16}$$

$$\text{BIAS} = \frac{\sum_{i=1}^{n}(E_i - O_i)}{n} \tag{17}$$

$$\text{MAPD} = \frac{100}{n}\sum_{i=1}^{n}\left|\frac{E_i - O_i}{O_i}\right| \tag{18}$$

$$\text{KGE} = 1 - \sqrt{(r-1)^2 + \left(\frac{\sigma_E}{\sigma_o} - 1\right)^2 + \left(\frac{\bar{E}}{\bar{O}} - 1\right)^2} \tag{19}$$

Where $R^2$ is the coefficient of determination, RMSE is root-mean-square error, BIAS is the mean
bias, MAPD is the mean absolute percent deviation, KGE is Kling-Gupta efficiency, n is the total
number of data pairs, the bar indicates mean value of the measured variable and model estimates
of the same variable. $E_i$ and $O_i$ are the model estimated and measured SEB fluxes, r is the Pearson's
correlation coefficient and $\bar{O}$ is the average of measured values and $\bar{E}$ is the average of estimated





values and $\sigma_o$ is standard deviation of observation values and $\sigma_E$ is the standard deviation of
estimated values. The KGE has been widely used for calibration and evaluation hydrological
models in recent years and it combines the three components of Nash-Sutcliffe efficiency (NSE)
of model errors (i.e. correlation, bias, ratio of variances or coefficients of variation) in a more
balanced way. But it has not been widely used for analyzing the ET model performances. KGE = 1
indicates perfect agreement between modelled estimates and observations. The performance of a
model is considered 'poor' for KGE between 0 and 0.5 and models with negative KGE values is
considered 'not satisfactory'.
**4 Results**
**4.1 Ecosystem- scale surface soil temperature amplitude (A)**
The scaling functions developed to estimate ecosystem-scale (1 km) surface soil temperature
amplitude (A) from point-scale $T_{STA}$ were used to estimate $G_i$. However, before the development
of the scaling functions, analysis was carried out to investigate the relationship of soil temperature
amplitude between the two different spatial scales. The scatterplot (Fig. 5a) of noon-night LST
difference ($\Delta T_s$) versus $T_{STA}$ for different albedo classes showed a linear increase in $\Delta T_s$ with
increasing $T_{STA}$. However, some divergence of data points within the cluster were also noticed
which could be associated with different albedo ($\alpha_R$) levels. Bivariate linear function was fitted
between $T_{STA}$ as predictand (Y) versus $\Delta T_s$ ($T_{sd} - T_{sn}$) and $\alpha_R$ as predictors (X1 and X2,
respectively). The function was found to be $Y = 0.59X1 - 51.3X2 + 8.66$ by combining the data
of nine ecosystems (r = 0.86). The coefficients in the above expressions correspond to B1 (0.59),
B2 (51.3), B3 (8.66) of eq. 5 in section 3.1.1.1. The estimated amplitude from this ecosystem-scale
predictors and scaling functions was treated as ecosystem-scale surface soil temperature amplitude
(A).





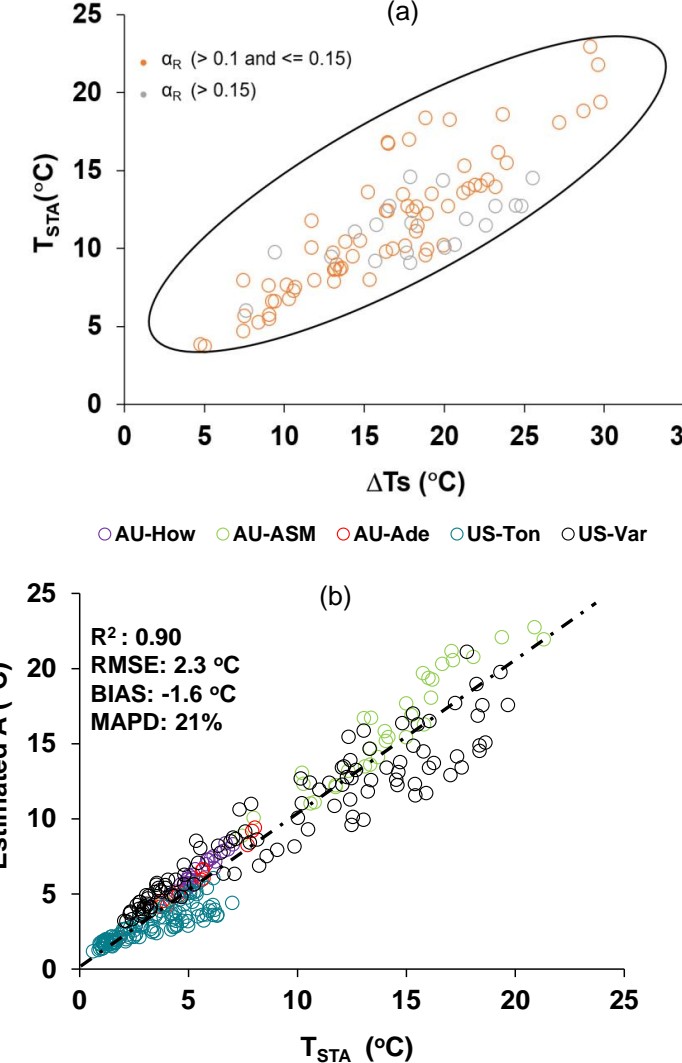

**Figure 5**. (a) Two-dimensional scatterplots between (ΔTs) versus $T_{STA}$ at different $\alpha_R$ levels over different ecosystems. Here $T_{STA}$ in y-axis is the observed soil temperature amplitude that is used to develop the scaling function and delta ΔTs is noon-night LST difference of MODIS AQUA; (b) Validation of the ecosystem-scale estimates of A from the above functions over different ecosystems and for independent years.

The validation of the ecosystem-scale estimates of A from the above functions over different
ecosystems is shown in Fig. 5b with respect to $T_{STA}$ for the independent datasets. The estimated A





was found to have MAPD of 21%, bias of -1.6 °C and $R^2 = 0.90$ over different ecosystems. The
temporal variation of estimated A and $T_{STA}$ is shown in Fig D1 in Appendix D.

**4.2 Sensitivity analysis of STIC-TI $G_i$, $LE_i$ and $H_i$ to land surface variables**

**4.2.1 Sensitivity of $G_i$ to land surface variables**

The average sensitivity of $G_i$ to three land surface variables ($T_S$, NDVI, $\alpha_R$) by combining the
estimates of wet and dry periods is shown in Fig. 6. $G_i$ was found to be substantially sensitive to
$T_S$ with error magnitude ranging from 2 – 18% due to $T_S$ uncertainties of ±0.5 – 2.5 K (Fig. 6a),
with greater sensitivity to $T_S$ during the summer season as compared to other seasons. The median
sensitivity of $G_i$ due to ±5 – 10% uncertainty in $\alpha_R$ varied from 5 to 12% in all the ecosystems (Fig.
6b). The uncertainties in NDVI revealed 2 to 15% error in $G_i$ estimates (Fig. 6c), and no significant
difference in the mean sensitivity due to NDVI uncertainties was noted between the ecosystems.
The sensitivity of $G_i$ decreased with increasing values of NDVI.

**4.2.2 Sensitivity of $LE_i$ and $H_i$ to land surface variables**

Both $LE_i$ and $H_i$ were sensitive to $T_S$ to the order of 2 – 29% ($LE_i$) and 5 – 35% ($H_i$) for $T_S$
uncertainty of ±0.5 – 2.5 K from its mean values (Table 3). Interestingly, $LE_i$ was more sensitive
to $T_S$ uncertainties as compared to $H_i$ in the rainfed ecosystems. The highest mean sensitivity of
$LE_i$ to $T_S$ was found in arid (IND-Jai: 2 – 28%), semi-arid (AU-ASM: 5 – 21%), tropical savanna
(IND-Dha: 3 – 26%), savanna (US-Ton: 4-29%) and arid (US-Var: 3-26%) ecosystems. The mean
sensitivity of $H_i$ to $T_S$ was maximum in sub-humid (IND-Sam: 2 – 32%), semi-arid (IND-Naw: 2
– 28%), savanna (AU-Ade: 8 – 17%) (Table 3). A greater sensitivity of the SEB fluxes due to $\alpha_R$
uncertainties was found than due to NDVI. The median sensitivity of $LE_i$ and $H_i$ due to 10%
uncertainty from mean $\alpha_R$ varied within 2 – 16% in all the ecosystems (Table 3). By contrast,
errors in the two SEB fluxes were substantially low (2 – 13%) due to ±0.05 – 0.15 uncertainty
from mean NDVI (Table 3).

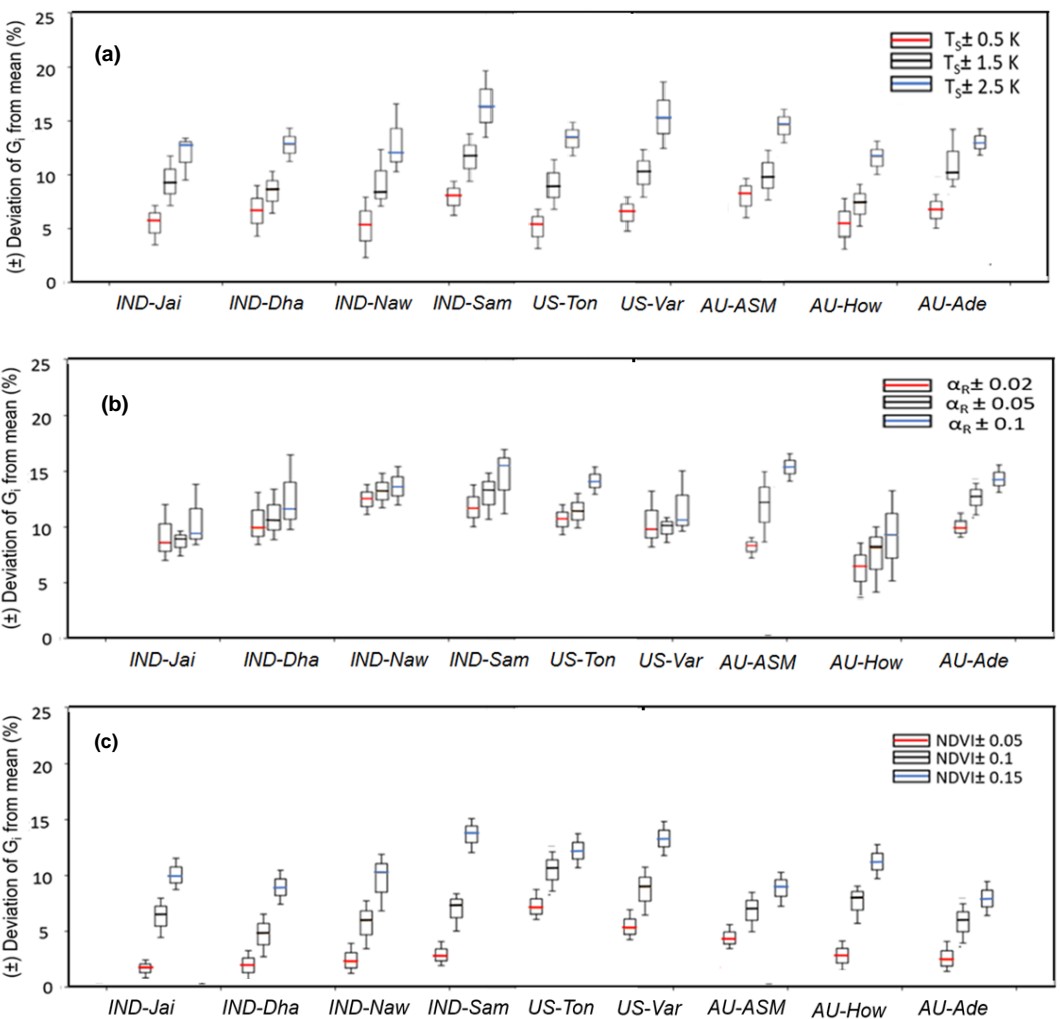

**Figure 6:** Sensitivity of STIC-TI $G_i$ due to uncertainties in $T_S$ (a), $\alpha_R$ (b), and NDVI (c) for eight flux tower sites in India and Australia. The uncertainties were introduced by taking the mean values of these variables during three different periods (summer, rainy and winter) of a year. Mean uncertainties of the three periods are presented in the figure.





**Table 3:** Sensitivity (in percent) of $LE_i$ and $H_i$ due to $T_S$, NDVI, and $\alpha_R$ uncertainties

| Study Sites | Sensitivity of $LE_i$ and $H_i$ to $T_S$, NDVI and $\alpha_R$ (% range) | | | | | |
| | $T_S$ uncertainty ($\pm0.5 – 2.5$ K) | | $\alpha_R$ uncertainty ($\pm5 – 10\%$) | | NDVI uncertainty ($\pm0.05 – 0.15$) | |
| | $LE_i$ | $H_i$ | $LE_i$ | $H_i$ | $LE_i$ | $H_i$ |
|---|---|---|---|---|---|---|
| IND-Jai | 2-28 | 1-6 | 3-14 | 2-13 | 2-8 | 2-6 |
| IND-Dha | 3-26 | 2-8 | 2-12 | 3-12 | 3-10 | 3-9 |
| IND-Naw | 1-20 | 2-28 | 2-10 | 3-10 | 2-7 | 2-6 |
| IND-Sam | 1-16 | 5-32 | 4-13 | 6-11 | 2-5 | 2-7 |
| US-Ton | 4-29 | 4-12 | 3-12 | 4-12 | 3-8 | 5-7 |
| US-Var | 3-26 | 6-14 | 4-11 | 2-10 | 4-10 | 2-8 |
| AU-ASM | 5-21 | 2-10 | 3-12 | 2-13 | 2-10 | 2-11 |
| AU-How | 8-13 | 2-15 | 2-11 | 4-16 | 3-12 | 3-13 |
| AU-Ade | 2-17 | 8-17 | 3-12 | 2-10 | 3-10 | 3-9 |


### 4.3 Comparative evaluation of STIC-TI and contemporary $G_i$ models

The performances of STIC-TI and existing $G_i$ models were evaluated and compared with respect
to *in-situ* $G_i$ measurements. The existing models reported by Moran et al. (1989), Bastiaanssen et
al. (1998), Su (2002), and Boegh et al. (2004) have been considered for comparing with TI-based
model. These four existing models are referred here as MOR89, BAS98, SU02 and BO04,
respectively. While the models MOR89, SU02 and BO04 are based on linear regression between
G versus NDVI, BAS98 is based on multivariate regression of G with NDVI, LST and $\alpha_R$. The
performance of the STIC-TI was substantially better as compared to MOR89, SU02 and BO04
with respect to MAPD (19%), RMSE (22 Wm$^{-2}$) and coefficient of determination ($R^2 = 0.8$) when
compared with *in-situ* measurements over one Indian, three Australian and two US flux tower sites
(Table 4) and also comparable with BAS98 $G_i$ model. The validation plot of retrieved noontime
Gi from STIC-TI is shown in Fig. 7.



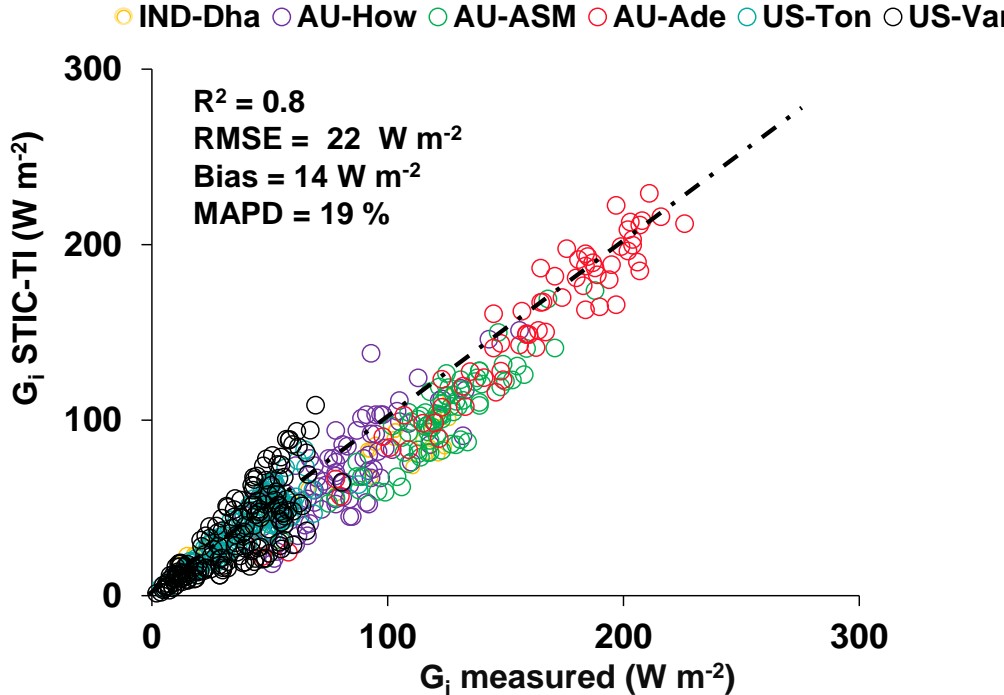

**Figure 7:** Validation of STIC-TI derived $G_i$ estimates with respect to *in-situ* measurements in different ecosystems. The regression between the two sources of $G_i$ is $G_i$ (STIC-TI) = 0.90$G_i$ (tower) -0.10.

**Table 4**: A comparison of error statistics of $G_i$ estimates from STIC-TI and existing $G_i$ models
over different ecosystems

| G models | $R^2$ | RMSE (W m$^{-2}$) | MAPD (%) | KGE |
|:---:|:---:|:---:|:---:|:---:|
| STIC-TI | 0.80 | 22 | 19 | 0.74 |
| MOR89 | 0.70 | 31 | 29 | 0.46 |
| BAS98 | 0.80 | 20 | 18 | 0.61 |
| SU02 | 0.80 | 30 | 26 | 0.54 |
| BO04 | 0.70 | 35 | 29 | 0.48 |

The RMSE varied from 9 to 20 W m$^{-2}$ with MAPD ranging from 12 to 21% across individual flux
tower sites. High magnitude of $G_i$ was predicted in the arid and semi-arid systems (120 – 240 W
m$^{-2}$) as compared to the humid systems (20 – 90 W m$^{-2}$), which was in close correspondence with
the observations. The model also captured the range of $G_i$ that are generally found in different
biomes (20 – 140 W m$^{-2}$ for grasslands, 20 – 90 W m$^{-2}$ for cropland) (Purdy et al., 2016). Due to





the paucity of $G_i$ measurements, direct validation of $G_i$ was only possible for 32 days (concurrent
to MODIS overpass) at the IND-Dha site. Overall, STIC-TI tends to provide reasonable G
estimates for the terrestrial ecosystems having soil temperature amplitude above 5ºC.
**4.4 Evaluation of STIC-TI $LE_i$, $H_i$, and EF**
The modelled versus measured $LE_i$ and $H_i$ showed good agreement in all the nine ecosystems with
RMSE in $LE_i$ and $H_i$ estimates to the order of 29 – 62 W m$^{-2}$ and 26 – 61 W m$^{-2}$, MAPD of 9 –
31% and 20 – 36%, BIAS of -29 to 38 W m$^{-2}$ and -44 to 32 W m$^{-2}$ (Fig. 8a, b; Table 5) and high
$R^2$ of 0.8.

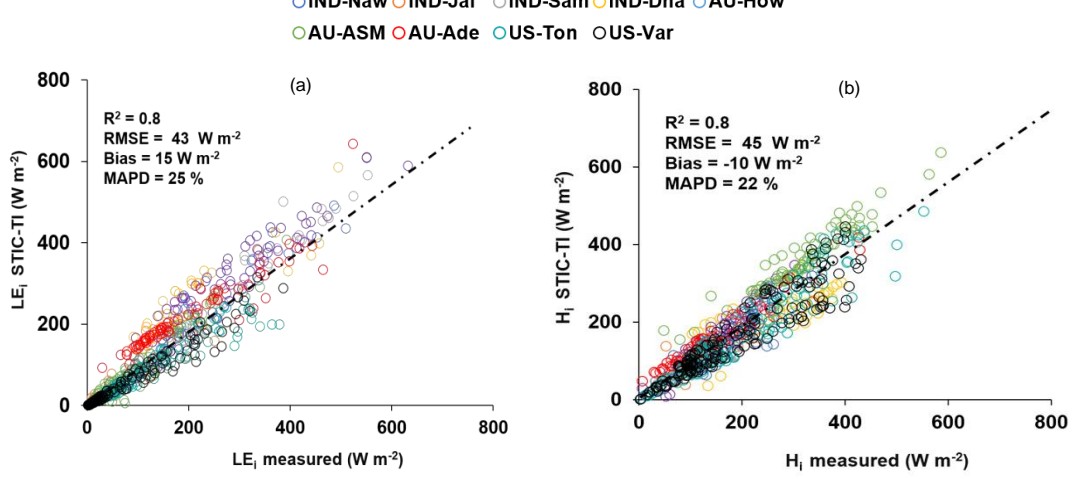

**Figure 8:** (a) Validation of STIC-TI $LE_i$ estimates with respect to *in-situ* measurements in different ecosystems.; (b) Validation of STIC-TI $H_i$ estimates with respect to *in-situ* measurements in different ecosystems.


**Table 5**: Error statistics of STIC-TI $LE_i$ and $H_i$ estimates with respect to EC measurements in different
ecosystems of India, US, and Australia.

| Sites | STIC- TI ($LE_i$ and $H_i$) | | | | |
|---|---|---|---|---|---|
| | $R^2$ | BIAS (W m$^{-2}$) | RMSE (W m$^{-2}$) | MAPD (%) | KGE |





|  | $LE_i$ | $H_i$ | $LE_i$ | $H_i$ | $LE_i$ | $H_i$ | $LE_i$ | $H_i$ | $LE_i$ | $H_i$ |
|---|---|---|---|---|---|---|---|---|---|---|
| IND-Jai | 0.87 | 0.85 | -21 | 12 | 57 | 27 | 31 | 22 | 0.80 | 0.76 |
| IND-Naw | 0.89 | 0.85 | 19 | -26 | 44 | 51 | 17 | 28 | 0.92 | 0.71 |
| IND-Dha | 0.92 | 0.91 | 38 | -44 | 43 | 35 | 27 | 25 | 0.71 | 0.64 |
| IND-Sam | 0.85 | 0.81 | 12 | -10 | 32 | 61 | 9 | 27 | 0.95 | 0.70 |
| US-Ton | 0.86 | 0.88 | -29 | -32 | 53 | 34 | 25 | 17 | 0.85 | 0.91 |
| US-Var | 0.84 | 0.79 | -19 | -28 | 49 | 39 | 27 | 20 | 0.82 | 0.89 |
| AU-ASM | 0.91 | 0.89 | -3 | 22 | 46 | 26 | 29 | 20 | 0.94 | 0.83 |
| AU-How | 0.88 | 0.86 | 16 | -25 | 42 | 27 | 17 | 21 | 0.89 | 0.85 |
| AU-Ade | 0.86 | 0.85 | 21 | 15 | 29 | 53 | 28 | 36 | 0.77 | 0.80 |

Arid ecosystems in India (IND-Jai), US (Ton and Var) and semi-arid ecosystem in Australia (AU-
ASM) revealed relatively high MAPD (31%, 25%, 27%, and 28%) (Table 5). In general, STIC-TI
was able to produce the dominant convective heat fluxes with respect to the EC measurements as
evident through low RMSE for $H_i$ and high RMSE for $LE_i$ in the IND-Jai, US-Ton, US-Var, and
AU-Ade where $LE_i$ is inherently low except few rainy days. A uniform distribution of data points
around 1:1 validation line (Fig. 8a) indicated overall low BIAS in $LE_i$ estimates. However,
modeled $H_i$ was consistently lower than the observations (negative BIAS) in the tropical savanna
(IND-Dha and AU-How) and semi-arid (IND-Naw) ecosystems [(-44) – (-25) W m$^{-2}$ and -26 W
m$^{-2}$) while a consistent positive BIAS was observed in the AU-ASM (semi-arid) and AU-Ade
(savanna), US-Var (arid) (Fig. 8b; Table 5). This consequently led to overall low negative BIAS
(-10 W m$^{-2}$), relatively low $R^2$ in $H_i$ ($R^2 = 0.8$) as compared to the errors in $LE_i$ (BIAS = 15 W m$^{-2}$, $R^2 = 0.9$). The regression between the modeled and tower measurements of $LE_i$ is $LE_i$(STIC-TI)



= 0.98LE$_i$(tower) – 0.266. The regression between the modeled and tower measurements of H$_i$ is
H$_i$ (STIC-TI) = 0.93H$_i$(tower) + 4.90. The KGE statistics varied in the range of 0.71 – 0.95 for LE$_i$
and in the range of 0.64 –0.91 for H$_i$, respectively across all nine flux tower sites, thus revealed
reasonably high efficiency of the model to capture the magnitude and variability of SEB fluxes.

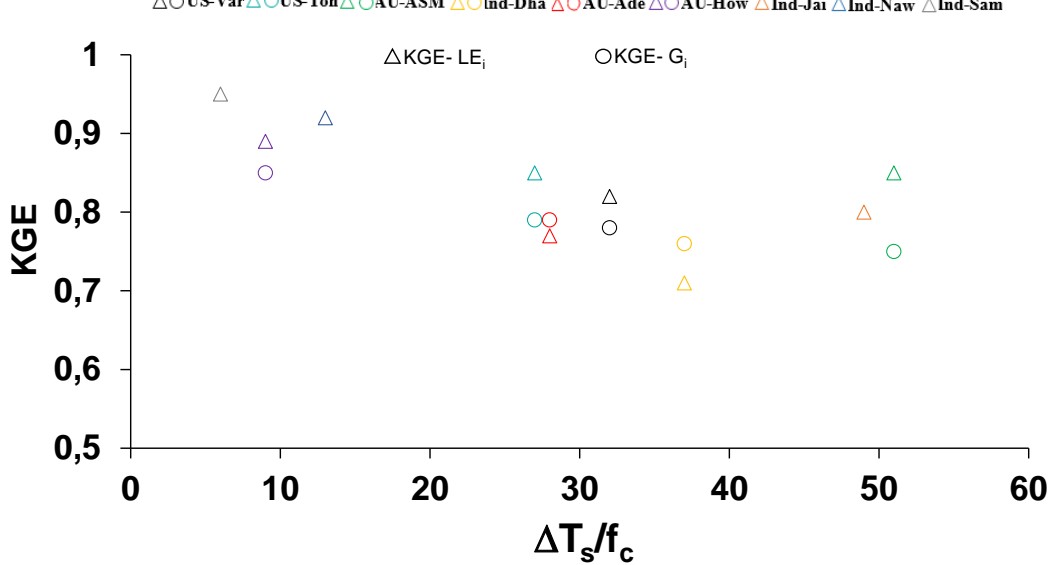

**Figure 9:** Relationship between KGE of STIC-TI (G$_i$ and LE$_i$) with $\Delta T_s/f_c$ in different terrestrial ecosystems.

Further investigation was made on whether KGE for STIC-TI G$_i$ and LE$_i$ follow any systematic
pattern and the ratio $\Delta T_S$ and f$_c$ was used as proxy for surface heterogeneity and dryness. The plot
of KGE of G$_i$ and LE$_i$ with this ratio is shown in Fig. 9. KGE-G$_i$ was found to show a systematic
decrease with increase in $\Delta T_s$-fc ratio up to 40, after which it remained unchanged with increase
in the ratio. Although KGE of LE$_i$ also decreased (20% reduction) with increase in $\Delta Ts$-fc ratio,
KGE-LE$_i$ was found to increase beyond $\Delta Ts$-fc 40. This revealed that the model efficiency
remained high (>0.8) within certain dryness limits ($\Delta Ts$-fc ratio <20 and >50) and the efficiency
reduced moderately (within 0.7 – 0.8) for intermediate dryness.
An independent evaluation of multi-temporal heat fluxes over two US flux sites for the years 2016-
2018 is shown in Fig. 10. STIC-TI G$_i$ estimates showed close match with *in-situ* measurements
with respect to intra and inter-annual variability in G$_i$ followed by LE$_i$ and H$_i$. This further





demonstrates the merit of the coupled model for reproducing ecosystem-scale $G_i$ estimates
especially for shorter and open canopies.

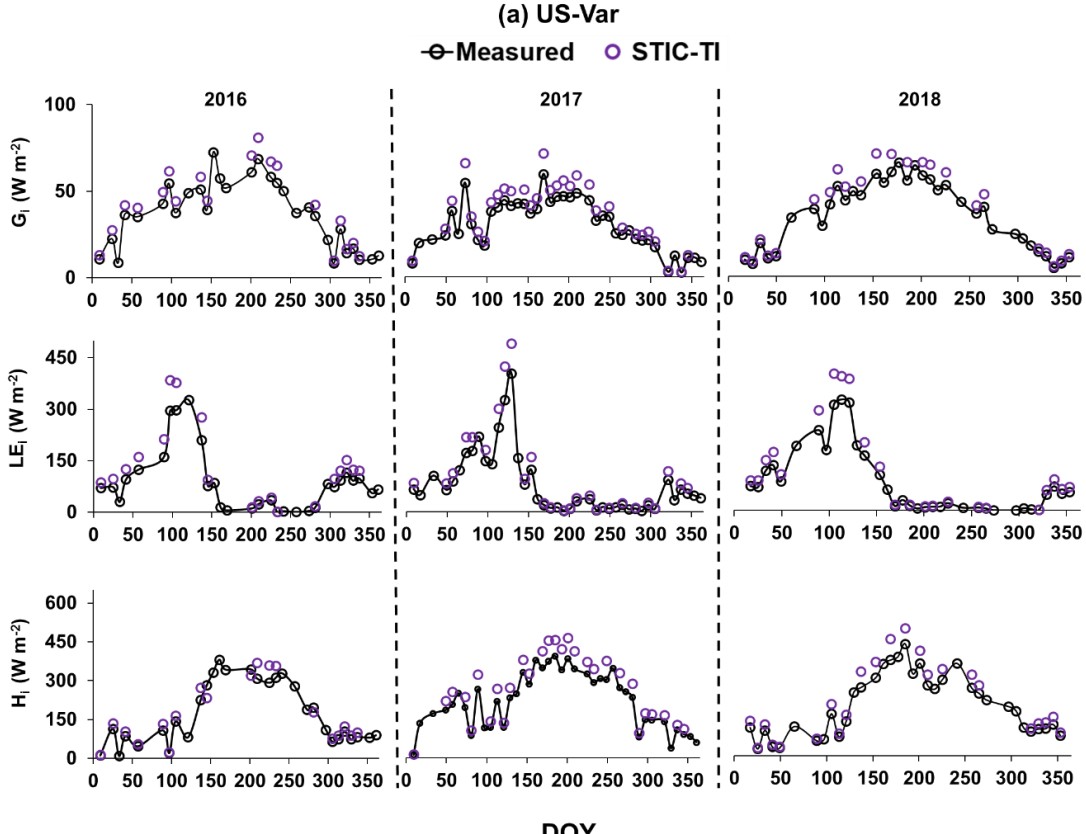

**Figure 10 (a):** Illustrative examples of temporal evolution of the STIC-TI derived versus observed SEB fluxes for three consecutive years from 2016 to 2018 in a grassland ecosystem in United States (e.g., US-Var).

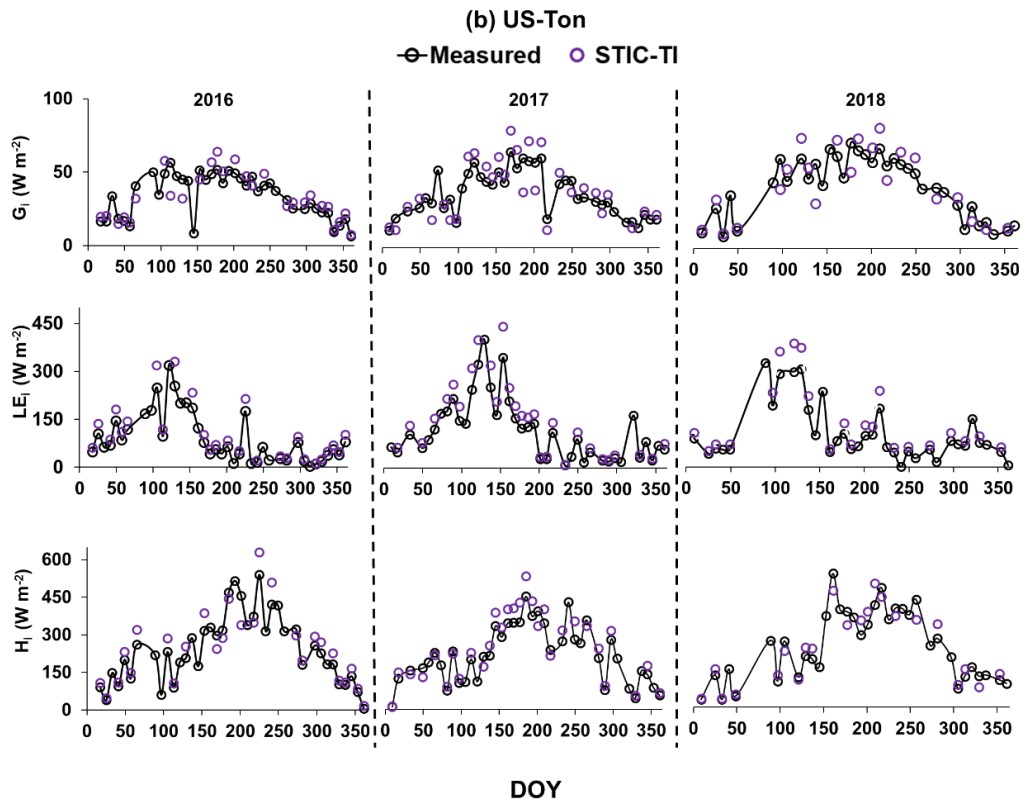

**Figure 10 (b):** Illustrative examples of temporal evolution of the STIC-TI derived versus observed SEB fluxes for three consecutive years from 2016 to 2018 in a woody savanna ecosystem in the United States (e.g., US-Ton).

Temporal behavior of STIC-TI and observed evaporative fraction (EF) (ratio of LE and $R_N$ – G) (Fig. 11a) along with observed monthly rainfall (P) distinctly captured the substantial temporal variability in EF during the dry-to-wet transition in the Indian study sites, which also corresponded to low (high) θ and P. In IND-Naw and IND-Sam, a marked rise (>0.4) in STIC-TI EF was noted during day-of-the-year (DOY) 25 to 75 where wheat is grown under assured irrigation. The impact of irrigation is thus captured by the substantial increase in EF in the absence of P. In contrast, the rainfed grassland system (IND-Jai) showed peak EF (~0.8), which corresponded to south-west monsoon rainfall during June to September and a progressive decline in EF during the dry down period in October to April corresponding to post south-west monsoon phase. Some intermittent spikes in EF was also noted during dry-down phase in both STIC-TI and observations. This could be due to extra latent heat energy transported through micro-advection from surrounding irrigated



agricultural land through the 'clothesline effect' which frequently occurs in semi-arid and arid
ecosystems. In addition to IND-Jai, the response of both modelled and measured EF to wet and
dry spells was also noted during south-west monsoon period at all other flux tower sites of India.

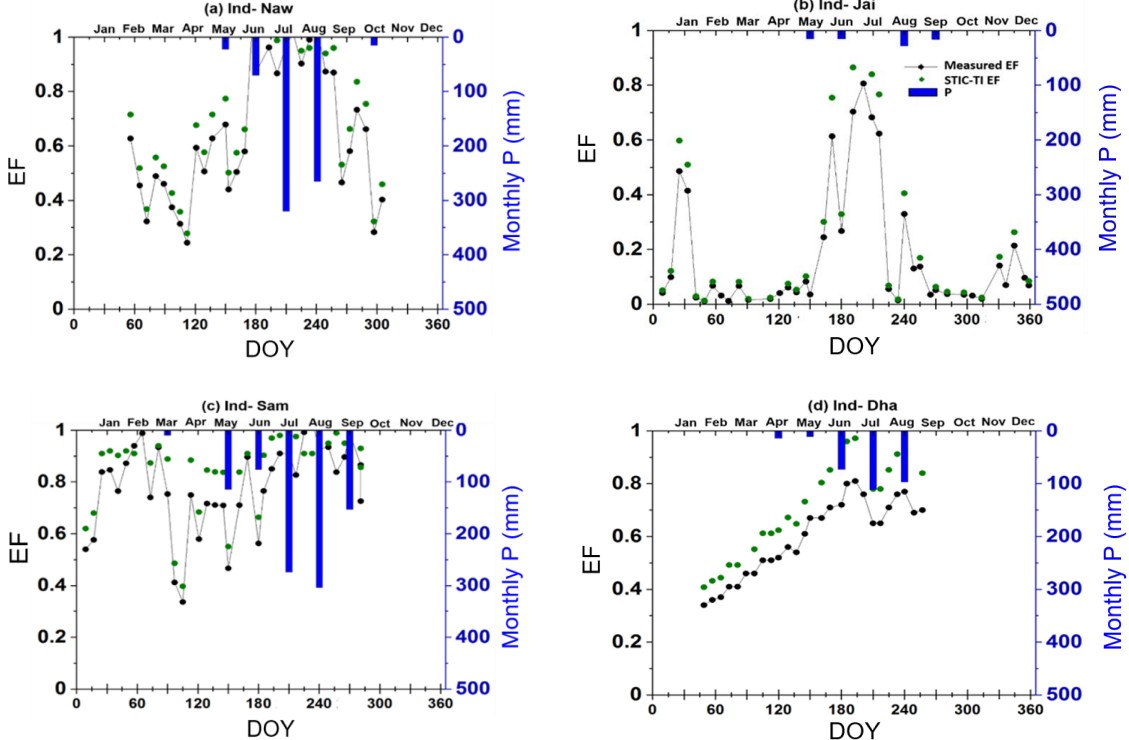

**Figure 11a**: Illustrative examples of temporal variation of STIC-TI derived EF with respect to measured EF and P in (a) IND-Naw, (b) IND-Jai, (c) IND-Sam, and (d) IND-Dha

The temporal behavior of EF from STIC-TI and EC measurements along with measured $\theta$ and P
at the two OzFlux and AmeriFlux sites also revealed (Fig. 11b) close correspondence of STIC-TI
with EC observations. Low EF (0.05 – 0.40) during the dry season around DOY 100 – 250 and
high EF (>0.4) during the wet season (DOY 1 – 120 and 300 to 360) in AU-ASM, US-Ton and
US-Var was observed. The analysis showed that STIC-TI EF can capture the annual variability of
observed EF and its responses across different ecosystems during wet and dry seasons. The plots
of STIC-TI EF versus measured $\theta$ (in the inset of Fig. 11b) revealed triangular scatter close to
right-angled triangle with positive slope of hypotenuse in three ecosystems AU-ASM, US-Var and
US-Ton. This showed that in the water-controlled ecosystems, where distinct wet-dry seasons





exist, the positive EF-θ relationship is an outcome of the soil moisture controls on transpiration
during the dry season.

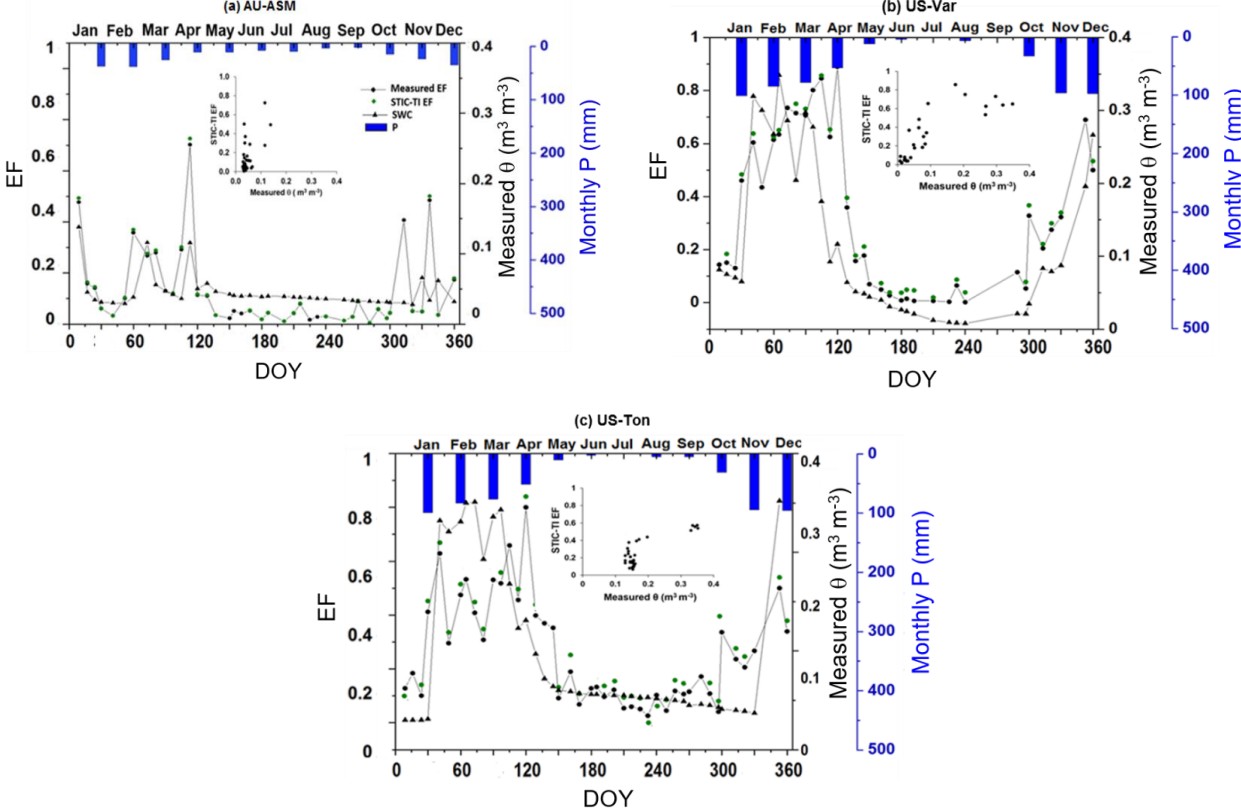

**Figure 11b**: Comparison of temporal variation of STIC-TI derived EF with respect to measured EF, θ and P in (a) AU-ASM, (b) US-Var, (c) US-Ton. The scatterplots in the inset shows the relationship between STIC-TI EF with respect to measured θ.

**5 Discussion**
**5.1 Interaction of flux and internal SEB metrices**
From the section 4.1 we found relatively reduced sensitivity of $G_i$ to Ts uncertainties. In any given
condition, if an over(under) estimation of M due to noontime $T_S$ uncertainties (through eq. 13)
leads to an over(under) estimation of $\Gamma$, the effects of such over(under) estimation of $\Gamma$ (due to
noontime $T_S$ uncertainties) tend to be compensated by the under(over) estimation of amplitude A
(in eq. 5) (Fig. 12d), ultimately leading to a reduction of the sensitivity of $G_i$ to $T_S$. While the





scatter between Γ-M and Γ- $T_S$ (Fig. 12a, b) revealed the sensitivity of $G_i$ to $T_S$ in arid (IND-Jai)
and tropical savannah (IND-Dha); which were due to the strong relationship between Γ and
daytime $T_S$ (Fig. 12b); the scatter between $G_i$, Γ, and A (Fig. 12c, d) revealed that the sensitivity
of $G_i$ to $T_S$ in semi-arid (IND-Naw) and sub-humid (IND-Sam) ecosystems were due to the strong
association between $G_i$ and A.

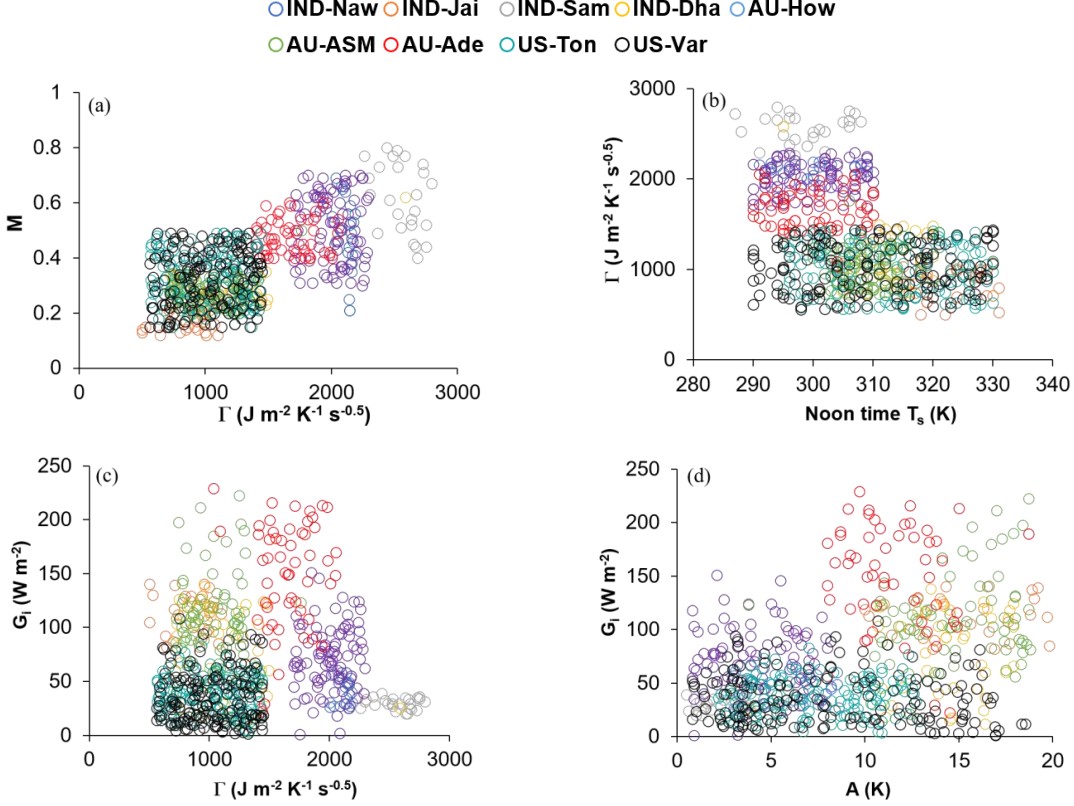

**Figure 12:** Response plots among parameters of TI-based $G_i$ model, such as (a) Γ vs. M, (b) Γ vs. noon-time $T_S$, (c) $G_i$ vs. Γ, and (d) $G_i$ vs. A over different ecosystems.

Concerning $LE_i$ and $H_i$, dual uncertainties could be propagated in both the fluxes through
daytime $T_S$ (through M and $G_i$), leading to high sensitivity of these two SEB fluxes due to $T_S$
perturbations. The relatively high sensitivity of $LE_i$ to $T_S$ (as compared to $H_i$) in the non-
irrigated ecosystems could be due to partial compensation of $g_A/g_S$ in both numerator and
denominator of the PMEB equation for H (eq. C7 of Appendix C). A recent study (Fig.10 in
Mallick et al., 2018a) showed high sensitivity of $g_S$ due to $T_S$ (1% change in $T_S$ led to 5.2–7.5%





change in $g_S$) as compared to $g_A$ sensitivity to $T_S$ (1% change in $T_S$ led to 1.6–2% change in $g_A$),
suggesting that errors in $g_S$ due to $T_S$ uncertainty tend to be larger than errors in $g_A$. Partial
cancellation of the conductance errors in the numerator of eq. (C7 of Appendix C) might have
resulted in compensation of $H_i$ errors in the water-limited ecosystems. In this environment, the
variability of $LE_i$ is mainly dominated by $g_A/g_S$, which makes $LE_i$ highly sensitive due to $T_S$
uncertainties. Combined uncertainty due to $g_A/g_S$ in the denominator and $g_A$ in the numerator
of eq. (C6 of Appendix C) resulted into greater sensitivity in $LE_i$ to $T_S$ in the arid and tropical
savannah ecosystems (Mallick et al., 2015, 2018a; Winter & Eltahir, 2010). The very low
sensitivity of $LE_i$ and Hi due to uncertainties in NDVI is because NDVI was not used in the
conductance parameterizations and effects due to NDVI in STIC-TI was only propagated
through $G_i$. The sensitivity of $LE_i$ and $H_i$ to albedo was mainly due to the dependence of net
radiation ($R_{Ni}$) on albedo, and any resultant uncertainty in $R_{Ni}$ (due to albedo) tends to be
reflected in the sensitivity of $LE_i$ and $H_i$ to albedo.
**5.2 Possible sources of errors in SEB flux evaluation**
In STIC-TI, underestimation and overestimation errors in $G_i$ in different ecosystems (Fig. 7) could
originate due to the errors in MOD11A1 LST product. A host of studies previously reported Ts
error of MOD11A1 LST product in the range of 2-3 K with a standard deviation of 0.009, which
is mainly due to errors in surface emissivity correction (Duan et al., 2017; Wan, 2014; Lei et al.,
2018). In the present analysis, we found an overestimation error of MODIS $T_S$ in the range of 0.5
– 1.5 K when compared with *in-situ* infrared temperature measurements at the tropical savanna
site. As mentioned in section 3.1, a positive (negative) bias in $T_S$ would tend to an overestimation
(underestimation) of amplitude (A) in eq. (5); underestimation (overestimation) of M in eq. (13),
and consequent underestimation (overestimation) of $\Gamma$ (eq. 12) and $G_i$, respectively. Furthermore,
the standard deviation of NDVI surrounding the tower sites varied from 0.01 – 0.05 when
compared to the ground measurements, which could be another source of error in the STIC-TI
model. In addition, NDVI saturates at LAI > 3. However, STIC-TI provides direct estimates of
ecosystem G and is independent of $R_N$. The higher accuracies of TI-based thermal diffusion model
as compared to $R_N$ dependent empirical G models were also reported by Purdy et al. (2016) at
daily or longer time scales in cropland, grassland. All these G model estimates many a times differ



from in situ measurements because of the no accounting of leaf litter presence or layer on soil floor
in the remote sensing-based G-model.
The overestimation (underestimation) of $LE_i$ ($H_i$) is also due to the effects of spatial resolution of
different input variables on these two SEB fluxes and conducted statistical evaluation with respect
to the measured SEB fluxes. Eswar et al. (2017) demonstrated the need for spatial disaggregation
models for monitoring $LE_i$ at field scale using contextual models by disaggregation of evaporative
fraction ($\Lambda$) and downwelling shortwave radiation ratio ($R_G$). Using different disaggregation
models, they estimated $LE_i$ at 250m spatial resolution and reported RMSE of $30 - 32$ W m$^{-2}$ as
compared to $LE_i$ obtained at 1000m spatial resolution with RMSE of $40 - 70$ Wm$^{-2}$ over different
sites in India. Anderson et al. (2007) reviewed different validation experiments conducted in
diverse agricultural landscapes (Anderson et al., 2004, 2005; Norman et al., 2003) and reported
RMSE in $LE_i$ in the range of $35 - 40$ W m$^{-2}$ (15%) at $30 - 120$ m disaggregated spatial resolution.
Current analysis also brought out the need for noon-night thermal imaging with spatial resolution
finer than 1000m to adequately capture the magnitude and variability of $LE_i$ in the terrestrial
ecosystems especially agroecosystems where average field sizes are less ($< 0.5$ ha) and fragmented
such as in India and other sub-continents.
As seen in Fig. 8a and Table 5, there is a gross overestimation of $LE_i$ with respect to the tower
observations. The consistent positive BIAS in STIC-TI $LE_i$ in five out of nine sites is presumably
due to the overestimation of $R_{Ni}$ (Figure B1 of Appendix B) and underestimation of $G_i$. Figure 7
shows overestimation of $G_i$ for three OzFlux sites and US sites and underestimation of $G_i$ for Indian
site with $G_i$ (STIC-TI) = 0.90 $G_i$(tower) - 0.10 and overestimation of $R_{Ni}$ at the ecosystem-scale,
with $R_{Ni}$ (STIC-TI) = 0.78$R_{Ni}$ (tower) + 58.92 (Appendix-B2). This means a systematic
overestimation of the net available energy ($R_{Ni} - G_i$) will be obvious in cases where STIC-TI shows
underestimation of $G_i$, which consequently leads to an overestimation of retrieved $LE_i$.
**5.3 Effects of SEB closure**
Using the unclosed SEB observations for Indian sites in absence of *in-situ* $G_i$ observations also
added to the consistent positive BIAS in the statistical evaluation of $LE_i$. A widespread lack of
energy balance closure to the order of $10 - 20\%$ worldwide at most of the EC sites is reported in
the literature (Stoy et al., 2013; Wilson et al., 2002), which implies a systematic underestimation
(overestimation) of $LE_i$(EC tower) (and/or $H_i$(EC tower)). Accommodating an average 15%





imbalance in $LE_i$(EC tower) would tend to diminish the positive BIAS in STIC-TI. Therefore, the
pooled gain (0.98) and positive BIAS between the STIC-TI and tower $LE_i$ is determined by the
overestimation of ($R_{Ni} – G_i$), combined with the underestimation of measured $LE_i$ from the EC
towers. An underestimation of $H_i$(negative BIAS) is associated with two reasons; (a) ignoring the
two-sided aerodynamic conductance of the leaves (Jarvis and McNaughton, 1986; Monteith and
Unsworth, 2013; Schymanski et al., 2017), which could lead to substantial underestimation of $H_i$,
and (b) due to the complementary nature of the PMEB equation, if $LE_i$ is overestimated, $H_i$ will
be underestimated. In addition, frequent micro-advection fluxes alter measured in situ H and LE
fluxes. But these advection conditions are not explicitly accounted in the current STIC-TI model.
**6 Summary and conclusions**
This study addressed one of the outstanding challenges in retrieving ground heat flux (G) and
evaporation (ET) in open canopy, water-controlled and radiation-controlled ecosystems. It
demonstrated coupling of a thermal inertia (TI)-based mechanistic G model with an analytical
surface energy balance (SEB) model (Surface Temperature Initiated Closure, STIC) using
satellite-based land surface temperature ($T_s$) and associated biophysical variables and has minimal
independence on *in-situ* measurements. The model is called STIC-TI, and this is the first ever
implementation of a coupled G-SEB model that does not require any empirical parameterization
of aerodynamic and canopy-surface conductance. By linking $T_S$ with thermal inertia ($\Gamma$) and
surface moisture availability (M), STIC-TI derives G through the harmonics equation between G
and $\Gamma$, and subsequently coupled G with the SEB fluxes. For estimating $\Gamma$, this paper also
developed scaling functions for ecosystem-scale surface soil temperature amplitude (A) through
bivariate regression between the observed soil temperature versus remote sensing derived $T_s$ and
surface albedo. Independent validation of STIC-TI using measured flux data from nine terrestrial
ecosystems in arid, semi-arid and sub-humid climate in India, USA (representing northern
hemisphere) and Australia (representing southern hemisphere) led us to the following conclusions:
(i)   The retrieved $G_i$ and associated SEB fluxes through STIC-TI were reasonably sensitive to

uncertainties in $T_S$ and vegetation index. However, a compensation effect was evident due to

the partial cancellation of overestimated TI and underestimated A in the harmonics equation

of G. Both, latent and sensible heat fluxes (LE and H), were extremely sensitive to $T_S$





uncertainties. While the maximum sensitivity of LE to $T_S$ was found in the arid and semi-arid
ecosystems, the sensitivity of H to $T_S$ was maximum in the sub-humid ecosystems.
(ii)  $G_i$ estimates through STIC-TI performed better as compared to most of the contemporary
empirical G models. It showed lower mean absolute percent deviation (MAPD) of 19% and
higher correlation coefficient (0.8) with respect to *in-situ* measurements for different
ecosystems. Despite the error statistics, G from STIC-TI was comparable to the existing semi-
empirical G model of Bastiaanssen et al. (1998) (BAS98), this coupled model has certain
advantages such as, (a) it provides direct estimates of G and is not dependent on net radiation
estimates, (b) the ecosystem-scale surface soil temperature amplitude used in G model can
advance our understanding on associated terrestrial ecosystem processes.
(iii) Overall, the STIC-TI explained significant variability in the measured SEB fluxes with a
MAPD of 19% for instantaneous G and 22 – 25% for instantaneous LE and H. The model
efficiency (KGE) was greater than 0.7 for G and LE in all the nine ecosystems having
contrasting aridity and canopy cover. Underestimation tendency of G in some ecosystems was
primarily attributed to the inherent bias in MODIS $T_S$ product, NDVI saturation at higher LAI
(>3) in conjunction with the spatial scale mismatch between single MODIS pixel and the
footprint of G measurements. The consequent overestimation (underestimation) of LE (H) in
some ecosystems was associated with the overestimation of the net available energy, use of
'unclosed' SEB observation in the validation of LE and H, the spatial scale discrepancy
between MODIS pixel versus eddy covariance measurement footprint, the complementary
nature of the Penman Monteith Energy Balance equation (for H), and possibly due to ignoring
the two-sided aerodynamic conductance by the leaves (for H).
The requirement of few input variables in STIC-TI generates promise for surface-atmosphere
exchange studies using readily available data from the current generation remote sensing satellites
(e.g., MODIS, INSAT) that have noon-night TIR observations. Current findings also provide
motivation in refining G simulation in the land surface models.  STIC-TI can be potentially used
for distributed ET mapping using current and future 4th generation Indian Geostationary satellite
observations from INSAT as well as future high spatial resolution (~ 60m) TIR observations with
3-day revisit from polar orbiting platform (Lagouarde et al., 2018, 2019) through the planned Indo-
French space-borne mission, TRISHNA (**T**hermal infrared **I**maging **S**atellite for **H**igh-resolution
**N**atural Resource **A**ssessment). This simple approach will also help in catering the need for a




reliable, space-time continuous ET datasets in data-poor regions like Indian sub-tropics, South-
East Asia and other parts of the world from thermal remote sensing observation.
**Author contributions**
KM and BKB conceptualized the idea; DD conducted STIC-TI model coding, simulations and
data analysis in consultation with KM and BKB; DD and BKB wrote the first version of the
manuscript with KM writing the introduction, discussions and conclusions; all authors contributed
to discussions, editing and corrections; BKB and KM jointly finalized the manuscript.
**Acknowledgement**
The authors gratefully acknowledge Ministry of Earth Sciences (MoES), Govt. of India and
Natioanl Environmental Research Council for providing necessary support through Indo-UK
INCOMPASS programme (NE/L013819/1, NE/L013843/1, NE/L01386X/1, NE/P003117/1).
BKB acknowledges Deputy Director, EPSA, SAC-ISRO and Director, SAC–ISRO for providing
necessary support to participate and contribute to Indo-UK INCOMPASS programme. DD
acknowledges Prof. P.D. Lele and Head from Department of Physics, Electronics and Space
Sciences, Gujarat University Ahmedabad and for providing the necessary support to carry out this
work. KM was supported by the Luxembourg Institute of Science and Technology (LIST) and
through the doctoral training unit and through the Mobility OUT fellowship of Luxembourg
National Research Fund (FNR) (PRIDE15/10623093/HYDROCSI;
INTER/MOBILITY/2020/14521920/MONASTIC). KCN is supported by the Jet Propulsion
Laboratory, California Institute of Technology, under contract with the National Aeronautics and
Space Administration and Government sponsorship is acknowledged. DDB acknowledges support
from NASA Ecostress project and the US Department of Energy, Office of Science which supports
the AmeriFlux project
**Data and code availability**
Harmonized time series datasets over the study grids are available in
https://doi.org/10.5281/zenodo.5806501. The model code is available to the first author upon
reasonable request.



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





**Appendix A**
**Table A1: A list of symbols, their descriptions and units used in the present study**

| Attributes | Symbol | Description |
|---|---|---|
| Temperature | $T_A$ | Air temperature ($^o$C) |
| | $T_{Max}$ | Maximum air temperature ($^o$C) |
| | $T_{Min}$ | Minimum air temperature ($^o$C) |
| | $T_D$ | Air dew-point temperature ($^o$C) |
| | $T_{STA}$ | point-scale soil temperature amplitude |
| | $\Delta T_S$ | noon-night LST difference ($^o$C) |
| | $T_{ST}$ | Soil temperature ($^o$C) |
| | $T_S$ | Land surface temperature (LST) ($^o$C) |
| Humidity, vapor pressures | $R_H$ | Relative humidity (%) |
| | $e_A$ | Atmospheric vapor pressure at the level of $T_A$ measurement (hPa) |
| | $e_A^*$ | Saturation vapor pressure at the level of $T_A$ measurement (hPa) |
| | $e_S^*$ | Saturation vapor pressure at surface (hPa) |
| | $D_A$ | Atmospheric vapor pressure deficit at the level of $T_A$ measurement (hPa) |
| Radiation | $R_G$ | Downwelling shortwave radiation (or global radiation) (W m$^{-2}$) |
| | $R_R$ | Upwelling or reflected shortwave radiation (W m$^{-2}$) |
| | $R_L\downarrow$ | Downwelling longwave radiation (W m$^{-2}$) |
| | $R_L\uparrow$ | Upwelling longwave radiation (W m$^{-2}$) |
| | $\tau_{sw}$ | Atmospheric transmissivity for shortwave radiation (unitless) |





| | | |
|---|---|---|
| | $\alpha_R$ | Broadband shortwave surface albedo (unitless) |
| SEB components | $LE_i$ | Latent heat flux (W m$^{-2}$); subscript 'i' signifies 'instantaneous' |
| | $H_i$ | Sensible heat flux (W m$^{-2}$); subscript 'i' signifies 'instantaneous' |
| | $G_i$ | Ground heat flux (W m$^{-2}$); subscript 'i' signifies 'instantaneous' |
| | $R_{Ni}$ | Net radiation (W m$^{-2}$); subscript 'i' signifies 'instantaneous' |
| | $\phi$ | Net available energy (W m$^{-2}$); i.e., $R_N - G$ |
| MV2007 model | $A$ | Ecosystem-scale surface soil temperature amplitude (°C) |
| | $T_{Sd}$ | Daytime $T_S$ (°C) |
| | $T_{Sn}$ | Nighttime $T_S$ (°C) |
| | $\omega$ | Angular frequency (rad s$^{-1}$) |
| | $\phi'_n$ | Phase shift of the n$^{th}$ soil surface temperature harmonic (rad) |
| | $\Delta$ | Shape parameter (unitless) |
| | $S_r$ | Relative soil moisture saturation (m$^3$ m$^{-3}$) |
| | $f_s$ | Sand fraction (unitless) |
| | $\theta_{fc}$ | Soil water content at field capacity (m$^3$ m$^{-3}$) |
| | $\theta_{wp}$ | Soil water content at permanent wilting point (m$^3$ m$^{-3}$) |
| | $\theta_*$ | Soil porosity (cm$^3$ cm$^{-3}$) |
| | $J_S$ | Summation of harmonic terms of soil surface temperature (K) |
| | $\Upsilon'$ | Soil textural parameter (unitless) |
| | $\Gamma$ | Soil thermal inertia (J K$^{-1}$ m$^{-2}$ s$^{-0.5}$) |
| | $\tau_0$ | Thermal inertia of air-dry soil (J K$^{-1}$ m$^{-2}$ s$^{-0.5}$) |
| | $\tau_*$ | Thermal inertia of saturated soil (J K$^{-1}$ m$^{-2}$ s$^{-0.5}$) |



| | | |
|---|---|---|
| | t' | Time of satellite overpass (seconds) |
| | $\Delta t$ | Time offset between the canopy composite temperature and the below-canopy soil surface temperature (seconds) |
| | $\kappa$ | Total number of harmonics used (unitless) |
| | $f_c$ | Vegetation fraction (unitless) |
| | $\theta$ | Volumetric soil moisture (cm cm$^{-3}$) |
| Clear-sky $R_{Ni}$ model | $R_{ns}$ | Net shortwave radiation (W m$^{-2}$) |
| | $R_{nl}$ | Net long wave radiation (W m$^{-2}$) |
| | $G_{sc}$ | Solar constant (1367 W m$^{-2}$) |
| | $\beta_e$ | Sun elevation angle ($^0$). |
| | $\varepsilon_s$ | Infrared surface emissivity (unitless) |
| | $\varepsilon_a$ | Atmospheric emissivity (unitless) |
| | E | Eccentricity correction factor due to variation in Sun-Earth distance (unitless) |
| | M | Aggregated moisture availability (0-1) |
| | $g_A$ | Aerodynamic conductance (m s$^{-1}$) |
| | $g_S$ | Canopy-surface conductance (m s$^{-1}$) |
| | $T_0$ | Aerodynamic temperature (or source/sink height temperature) ($^{o}$C) |
| | $T_{0D}$ | Dewpoint temperature at the source/sink height ($^{o}$C) |
| | $\Lambda$ | Evaporative fraction (unit less) |
| | $e_0$ | Vapor pressure at the source/sink height (hPa) |
| | $e_0^{*}$ | Saturation vapor pressure at the source/sink height (hPa) |





| STIC-TI model | $D_0$ | Vapor pressure deficit at source/sink height (hPa) |
|---|---|---|
| | $s_1$ | Psychrometric slope of vapor pressure and temperature between $(T_{0D} -T_D)$ versus $(e_0 -e_A)$ (h Pa $K^{-1}$) |
| | $s_2$ | Psychrometric slope of vapor pressure and temperature between $(T_S-T_D)$ versus $(e_s^*-e_A)$ (h Pa $K^{-1}$) |
| | $s_3$ | Psychrometric slope of vapor pressure and temperature between $(T_{0D} -T_D)$ versus $(e_s^*-e_A)$. |
| | $\kappa$ | Ratio between $(e_0^* - e_A)$ and $(e_s^* - e_A)$ (unitless) |
| | $s$ | Slope of saturation vapor pressure vs. temperature curve (h Pa $K^{-1}$) |
| | $\alpha$ | Priestley-Taylor coefficient (unitless) |
| Ancillary meteorological variables | $U$ | Wind speed at 8 m height (m $s^{-1}$) |
| | $u^*$ | Friction velocity (m $s^{-1}$) |
| Constants | $P$ | Precipitation (mm $d^{-1}$) |
| | $\gamma$ | Psychrometric constant (h Pa $k^{-1}$) |
| | $c_P$ | Specific heat capacity of air at constant pressure (MJ $kg^{-1}$ $K^{-1}$) |
| | $\rho$ | Density of air (Kg $m^{-3}$) |
| | $\sigma$ | Stefan–Boltzmann constant ($5.67 \times 10^{-8}$ $Wm^{-2}K^{-4}$) |








**Table A2:** Summary of instruments used, height or depth and period of measurements, measured
variables at nine EC flux tower sites

| Type of primary instruments used for in situ data recording at flux tower sites | Measurement Height/ Depth (m) at different sites | Measured variables |
|---|---|---|
| Net radiometer | • 3m (IND-Naw, IND-Jai, IND-Sam)<br>• 15m (AU-Ade)<br>• 12.2m (AU-ASM)<br>• 23m (AU-How)2m (US-Ton, US-Var) | Four radiation flux components: shortwave incoming ($R_G$) and outgoing ($R_R$); longwave incoming ($R_L\downarrow$) and outgoing ($R_L\uparrow$) |
| EC assembly with IRGA (Infrared Gas Analyzer), three-dimensional sonic anemometer, TC probe | • 8m (IND-Naw; IND-Jai; IND-Sam)<br>• 4.5m (IND-Dha)<br>• 15m (AU-Ade)<br>• 11.6m(AU-ASM)<br>• 23m (AU-How)<br>• 2m (US-Ton, US-Var) | High response wind vectors ($u$, $v$ and $w$), sonic temperature, and $CO_2$- water vapor mass at 10/20 Hz frequency |
| Humidity and temperature probe | • 8m (IND-Naw, IND-Jai, IND-Sam)<br>• 4.5m (IND-Dha)<br>• 15m (AU-Ade), 11.6m (AU-ASM)<br>• 23m (AU-How), 70m (AU-How)<br>• 2m (US-Ton, US-Var) | $T_A$ and $R_H$ |
| Soil temperature probe | • -0.1m (IND-Dha)<br>• -0.15m (AU-Ade)<br>• (-0.02, -0.06m) (AU- ASM)<br>• -0.08m (AU- How)<br>• -0.02m, -.04m, -0.08m, and -0.16m (US-Ton, US-Var) | $T_{ST}$ |
| Soil heat flux plates | • Ground, 0.1 m (IND-Dha)<br>• Ground, -0.15 m (AU-Ade)<br>• Ground, -0.08 m (AU-ASM)<br>• Ground, -0.15 m (AU-How)<br>• -0.01m (US-Ton, US-Var) | Soil heat flux (G) |

**Appendix B**
**B1: Clear-sky instantaneous net radiation ($R_{Ni}$) model**
Net radiation ($R_N$) is defined as the difference between the incoming and outgoing radiation fluxes,
which includes both longwave and shortwave radiation at the surface of earth.



Terrestrial $R_N$ has four components: downwelling and upwelling shortwave radiation ($R_G$ and $R_R$),
downwelling and upwelling longwave radiation ($R_L\downarrow$ and $R_L\uparrow$), respectively.

$$R_N = (R_G - R_R) + (R_{L\downarrow} - R_{L\uparrow}) \qquad (B1)$$

Out of these four terms mentioned in eq.(B1), $R_G$ and $R_L\downarrow$ are dependent on various factors such
as geographic location, season, cloudiness, aerosol loading, atmospheric water vapor content and
less on surface properties. On the other hand, the upwelling radiations in eq. (B1) strongly depends
on the surface properties such as surface reflectance and emittance, land surface temperature, and
soil water content (Zerefos and Bais, 2013).
Instantaneous net radiation ($R_{Ni}$) can be derived using eq. B2 as follows (Mallick et al., 2007):

$$R_{Ni} = R_{ns} - R_{nl} \qquad (B2)$$

$$R_{ns} = (1 - \alpha_R)\, R_G \qquad (B3)$$

$$R_{nl} = R_{L\downarrow} - R_{L\uparrow} \qquad (B4)$$

Where, $R_{ns}$ is net shortwave radiation (W m$^{-2}$), $R_{nl}$ is net longwave radiation (W m$^{-2}$).and $\alpha_R$ is
the broadband surface albedo shortwave spectrum.
A WMO (World Meteorological Organization) shortwave radiation model (Cano et al.,1986)
calibrated over Indian conditions (Mallick et al., 2007, 2009) was used to compute $R_G$ using the
following equation:

$$R_G = \tau_{sw} G_{sc} E\, (\sin\beta_e)^{1.15} \qquad (B5)$$

Where, $\tau_{sw}$ is the is the global clear sky transmissivity for the shortwave radiation (0.7), $G_{sc}$ is the
solar constant (1367 Wm$^{-2}$), $\varepsilon$ is the eccentricity correction factor due to variation in Sun-Earth
distance and $\beta_e$ is the sun elevation in degrees.
$R_L\downarrow$ at any instance was calculated as follows:

$$R_{L\downarrow} = \varepsilon_a\, \sigma\, (273.14 + T_A)^4 \qquad (B6)$$





Where, σ is the Stefan–Boltzmann constant (5.67 x$10^{-8}$ $Wm^{-2}K^{-4}$); $T_A$ is the air temperature ($^0$C);
$\varepsilon_a$ is the atmospheric emissivity.
Atmospheric emissivity ($\varepsilon_a$) was computed using the following equation (Bastiaanssen et
al.,1998):

$$\varepsilon_a = 0.85 - \ln\tau_{sw}{}^{0.09} \tag{B7}$$

$R_L\uparrow$ at any particular instance was calculated as follows:

$$R_{L\uparrow} = \varepsilon_s \, \sigma(273.14 + T_s)^4 \tag{B8}$$

Where, $\varepsilon_s$ is the surface emissivity in thermal infrared (8 – 14 μm) spectrum and $T_S$ is the land
surface temperature ($^0$C).
**B2: Evaluation of STIC-TI $R_{Ni}$**
Comparison of the clear-sky $R_{Ni}$ estimates with respect to *in situ* measurements revealed RMSE in
$R_{Ni}$ to the order of 27 – 72 W $m^{-2}$, MAPD 8 –24%, BIAS (-67) – 50 W $m^{-2}$, and $R^2$ varying from
0.62– 0.90 across all the sites (Fig. B2, Table B2). Among the nine sites, a consistent
underestimation of $R_{Ni}$ was noted in IND-Dha, US-Ton, and US-Var (with BIAS of -23 W $m^{-2}$, -
61 W $m^{-2}$ and -67 W $m^{-2}$), whereas substantial overestimation of $R_{Ni}$ was found in IND-Sam, IND-
Naw, and AU-ASM with a BIAS of 50 W $m^{-2}$, 37 W $m^{-2}$ and 43 W $m^{-2}$, respectively (Table B2).



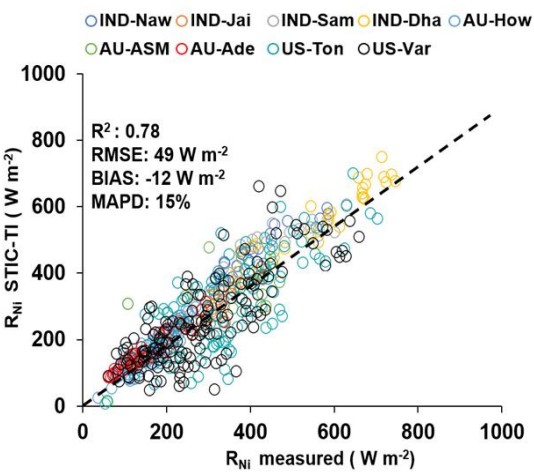

Figure B2: Validation of STIC-TI derived $R_{Ni}$ estimates with respect to *in situ* measurements in different ecosystems. The regression equation between modeled versus in-situ $R_{Ni}$ is, $R_{Ni}$ (STIC-TI) = 0.78$R_{Ni}$ (tower) +58.92.

**Table B2:** Performance evaluation statistics of clear-sky $R_{Ni}$ estimates in nine different
agroecosystems

| Sites | Error statistics of clear-sky $R_{Ni}$ model estimates | | | |
|---|---|---|---|---|
| | $R^2$ | BIAS (W m$^{-2}$) | RMSE (W m$^{-2}$) | MAPD (%) |
| IND-Jai | 0.81 | -9 | 32 | 8 |
| IND-Naw | 0.81 | 37 | 56 | 12 |
| IND-Dha | 0.81 | -23 | 42 | 9 |
| IND-Sam | 0.64 | 50 | 67 | 15 |
| US-Ton | 0.68 | -61 | 69 | 21 |
| US-Var | 0.62 | -67 | 72 | 24 |
| Au-How | 0.87 | 7 | 27 | 15 |
| AU-ASM | 0.88 | 43 | 50 | 14 |
| AU-Ade | 0.90 | 11 | 27 | 16 |





**Appendix C**
**C1: Estimating SEB fluxes using STIC1.2 analytical model and thermal remote sensing data**
STIC1.2 (Mallick et al., 2014, 2015a,b, 2016, 2018a) is a one-dimensional physically based SEB
model and is based on the integration of satellite LST observations into the Penman–Monteith
Energy Balance (PMEB) equation (Monteith, 1965). In STIC1.2, the vegetation–substrate
complex is considered as a single slab. Therefore, the aerodynamic conductances from individual
air-canopy and canopy-substrate components is regarded as an 'effective' aerodynamic
conductance ($g_A$), and surface conductances from individual canopy (stomatal) and substrate
complexes is regarded as an 'effective' canopy-surface conductance ($g_S$) which simultaneously
regulate the exchanges of sensible and latent heat fluxes (H and LE) between surface and
atmosphere. One of the fundamental assumptions in STIC1.2 is the first order dependence of these
two critical conductances on M through $T_S$. Such an assumption enabled an integration of satellite
LST in the PMEB model (Mallick et al., 2016). The common expression for LE and H according
to the PMEB equation is as follows:

$$LE = \frac{s\phi + \rho c_P g_A D_A}{s + \gamma\left(1 + \frac{g_A}{g_S}\right)} \qquad (C6)$$

$$H = \frac{\gamma\phi\left(1 + \frac{g_A}{g_S}\right) - \rho c_P g_A D_A}{s + \gamma\left(1 + \frac{g_A}{g_S}\right)} \qquad (C7)$$

In the above equations, the two biophysical conductances ($g_A$ and $g_S$) are unknown and the
STIC1.2 methodology is based on finding analytical solutions for the two unknown conductances
to directly estimate LE (Mallick et al., 2016, 2018a). The need for such analytical estimation of
these conductances is motivated by the fact that $g_A$ and $g_S$ can neither be measured at the canopy
nor at larger spatial scales, and there is no universally agreed appropriate model of $g_A$ and $g_S$ that
currently exists (Matheny et al., 2014; van Dijk et al., 2015). By integrating $T_S$ with standard SEB
theory and vegetation biophysical principles, STIC1.2 formulates multiple state equations in order



to eliminate the need to use the empirical parameterizations of the $g_A$ and $g_S$ and also to bypass the
scaling uncertainties of the leaf-scale conductance functions to represent the canopy-scale
attributes. The state equations for the conductances are expressed as a function of those variables
that are mostly available as remote sensing observations and weather forecasting models. In the
state equations, a direct connection to $T_S$ is established by estimating M as a function of $T_S$. The
information of M is subsequently used in the state equations of conductances, aerodynamic
variables (aerodynamic temperature, aerodynamic vapor pressure), and evaporative fraction,
which is eventually propagated into their analytical solutions. M is a unitless quantity, which
describes the relative wetness (or dryness) of a surface and also controls the transition from
potential to actual evaporation; which implies M→1 under saturated surface conditions and M→0
under extremely dry conditions. Therefore, M is critical for providing a constraint against which
the conductances are estimated. Since $T_S$ is extremely sensitive to the surface moisture variations,
it is extensively used for estimating M in a physical retrieval scheme (detail in Appendix A3 of
Bhattarai et al., 2018; Mallick et al., 2016, 2018a). It is hypothesized that linking M with the
conductances will simultaneously integrate the information of $T_S$ into the PMEB model. To
illustrate, we express the state equations by symbols, $sv_1= f \{c_1, c_2, c_3, v_1, v_2, v_3, v_4, sv_3, sv_5\}$; $sv_2$
$= f \{v_4, sv_1, sv_5, sv_6\}$; $sv_3 = f \{c_3, v_3, v_4, sv_4, sv_5\}$; $sv_4 = f \{c_3, v_3, sv_1, sv_2, sv_7, sv_8\}$. Here, f, sv, v,
and c denote the function, state variables, input variables (5 input variables; radiative and
meteorological), and constants (3 constants), respectively. Here $sv_1$ to $sv_4$ are $g_A$, $g_S$, aerodynamic
temperature ($T_0$), evaporative fraction (Λ), and $sv_8$ is M. Given the estimates of M, net radiative
energy ($R_{Ni}- G_i$), $T_A$, $R_H$, the four state equations are solved simultaneously to derive analytical
solutions for the four state variables and to produce a surface energy balance "closure" that is
independent of empirical parameterizations for $g_A$, $g_S$, $T_0$, and Λ. However, the analytical solutions
to the four state equations contain three accompanying unknown state variables (effective vapor
pressures at source/sink height, and Priestley-Taylor variable), and as a result there are four
equations with seven unknowns. Consequently, an iterative solution was found to determine the
three additional unknown variables as detailed in this section above and also described in Mallick
et al. (2016, 2018a) and Bhattarai et al. (2018). The state equations of STIC are given below.





$$g_A = \frac{\phi}{\rho c_P \left[ (T_0 - T_A) + \left( \frac{e_0 - e_A}{\gamma} \right) \right]} \tag{C1}$$

$$g_S = g_A \frac{(e_0 - e_A)}{(e_0^* - e_0)} \tag{C2}$$

$$T_0 = T_A + \left( \frac{e_0 - e_A}{\gamma} \right) \left( \frac{1 - \Lambda}{\Lambda} \right) \tag{C3}$$

$$\Lambda = \frac{2\alpha s}{2s + 2\gamma + \gamma \frac{g_A}{g_S} (1 + M)} \tag{C4}$$

Detailed derivations of these four state equations are given in Mallick et al. (2016). Given the
values of M, $R_N$, G, $T_A$, and $R_H$ or $e_A$, the four state equations can be solved simultaneously to
derive analytical solutions for the four unobserved variables and to simultaneously produce a
'closure' of the PMEB model that is independent of empirical parameterizations for both $g_A$ and
$g_S$. However, the analytical solutions to the four state equations contain three accompanying
unknowns; $e_0$ (vapor pressure at the source/sink height), $e_0^*$ (saturation vapor pressure at the
source/sink height), and Priestley-Taylor coefficient ($\alpha$), and as a result there are four equations
with seven unknowns. Consequently, an iterative solution was needed to determine the three
unknown variables (as described in Appendix A2 in Mallick et al. 2016). Once the analytical
solutions of $g_A$ and $g_S$ are obtained, both variables are returned into eq. (13) to directly estimate
LE.
In STIC-TI, an initial value of $\alpha$ was assigned as 1.26; initial estimates of $e_0^*$ were obtained from
$T_S$ through temperature-saturation vapour pressure relationship, and initial estimates of $e_0$ were
obtained from M as, $e_0 = e_A + M(e_0^* - e_A)$. Initial $T_{0D}$ and M were estimated according to
Venturini et al. (2008) as described in section 3.2, and initial estimation of G was performed from
initial M using the equation sets eq. (2) – eq. (11). With the initial estimates of these variables;
first estimate of the conductances, $T_0$, $\Lambda$, H, and LE were obtained. The process was then iterated
by updating $e_0^*$, $D_0$, $e_0$, $T_{0D}$, M, and $\alpha$ (using eq. A9, A10, A11, A17, A16 and A15 in Mallick et
al., 2016), with the first estimates of $g_S$, $g_A$, $T_0$, and LE, and re-computing G, $\phi$, $g_S$, $g_A$, $T_0$, $\Lambda$, H,
and LE in the subsequent iterations with the previous estimates of $e_0^*$, $e_0$, $T_{0D}$, M, and $\alpha$ until the



convergence of LE was achieved. Stable values of G, conductances, LE, H, $T_0$, $e_0^*$, $e_0$, $T_{0D}$, M, and
α were obtained within ~25 iterations. The inputs needed for computation of $LE_i$ (eq.C6) are air
temperature ($T_A$), land surface temperature ($T_S$), relative humidity ($R_H$), net radiation ($R_{Ni}$) and
soil heat flux ($G_i$).
**Appendix D**
The temporal variation of estimated A and $T_{STA}$ is shown in Fig. D1. The annual variations of $T_{STA}$
in different ecosystem was found to be within the ranges of 1 - 4°C.

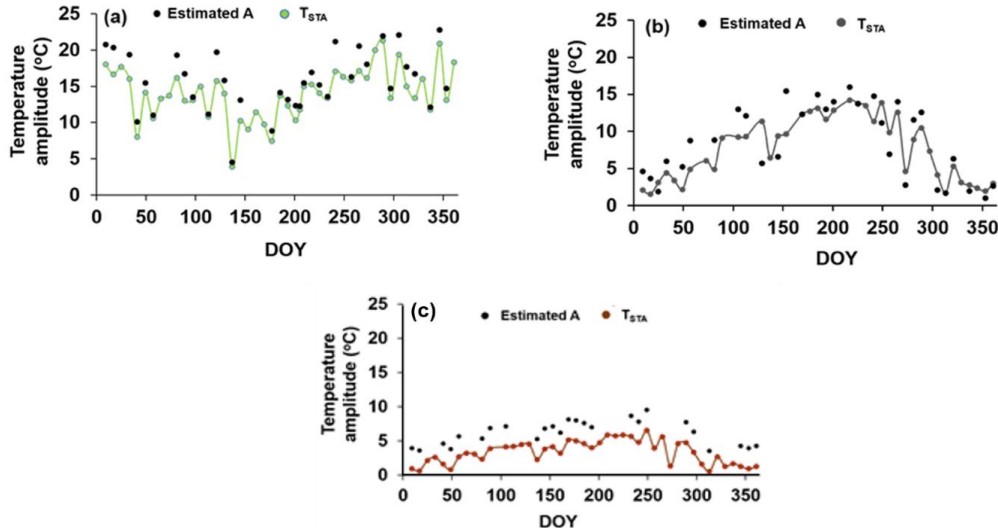


**Figure D1:** Temporal variation of A and $T_{STA}$ in (a) AU-ASM (2013), (b) US-Ton (2014), (c) US-
Var (2014)**.**








**Appendix E**
**Table E1**: Soil textural properties and their values used in the present study (Murray and Verhoef,
2007; Minasny et al., 2011; Anderson et al., 2007)

| Soil texture | Water retention Shape parameter ($\delta$) | Field capacity (vol/vol) (%) $\theta_{fc}$ | Wilting point (vol/vol) (%) $\theta_{wp}$ | Sand fraction ($f_s$) | Saturated soil moisture (vol/vol) (%) $\theta_*$ |
|---|---|---|---|---|---|
| Sand | 2.77 | 10 | 5 | 0.92 | 43 |
| Loamy Sand | 2.39 | 12 | 5 | 0.82 | 41 |
| Sandy loam | 2.27 | 18 | 8 | 0.58 | 41 |
| Loam | 2.20 | 28 | 14 | 0.43 | 43 |
| Silty loam | 2.22 | 31 | 11 | 0.17 | 45 |
| Sandy clay loam | 2.17 | 27 | 17 | 0.58 | 39 |
| Clay loam | 2.14 | 36 | 22 | 0.40 | 41 |
| Silty clay loam | 2.14 | 38 | 22 | 0.10 | 43 |
| Sandy clay | 2.11 | 36 | 25 | 0.52 | 38 |
| Silty clay | 2.12 | 41 | 27 | 0.06 | 46 |
| Clay | 2.10 | 42 | 30 | 0.22 | 38 |





