# Peer review of "A Coupled Ground Heat Flux-Surface Energy Balance Model of 1 Evaporation Using Thermal Remote Sensing Observations 2"

_Biogeosciences, 2021_

## Referee Comment (RC1)

**Review Desai et al. (2022).**

**General comments**

This paper looks at improved estimates of the instantaneous noon-time surface energy balance (SEB) obtained from a variety of field and remote sensing data. The main emphasis of this paper is on the improved estimation of soil heat flux G, by employing the thermal inertia and soil heat flux models proposed by Murray and Verhoef amongst others. There are some good to marginal improvements compared to simpler methods used to calculate G. The STICS-TI model does a good job at predicting the noon-time latent and sensible heat fluxes for complex sparse canopies in a range of climates.

It should be applauded that the authors have managed to structure the description and analyses of a large amount of data and theory in this manuscript, also with the help of a large number of Appendices. Perhaps some of these could be Supplementary Information?. However, the English could be improved in many places. There are a number of native English speakers on the author list, so I expect to see a much improved revised version.

The introduction rambles quite a bit, and contains a number of inaccuracies and 'red herrings' as per my many minor comments below.

I still have my doubts about the validity of the STICS equations and assumptions, from a micrometeorological and soil physical perspective, but seeing that this theory has been peer-reviewed and published many times in recent years, I will not argue with this again..

I am not sure how useful these peak values of LE are from a practical point of view. Can they be used for irrigation? For land surface model data-assimilation?? They cannot really be used in water balance estimates for example.

**Introduction**

Line 73-74: I have two issues with this part of the sentence "....which makes it a crucial variable for estimating sensible heat flux (H) ET through the SEB models". What is meant by "sensible heat flux (H) ET"? Is there a word missing here? Also: what are " the SEB models" exactly? You mention an "analytical surface energy balance (SEB) model" in line 71, but that is a more specific model.

Line 79-80: Remove the hyphen in "extra-resistance". Also, it is often called "excess resistance", see e.g. Verhoef et al., (1997), and it relates to the differences in roughness lengths for momentum and heat/water vapour. Is this worth mentioning here?

https://journals.ametsoc.org/view/journals/apme/36/5/1520-0450_1997_036_0560_spnotp_2.0.co_2.xml

Line 83: how is the aerodynamic conductance (gA) related to the extra resistance? According to the Appendix it is not, so is all this discussion here about roughness lengths, aerodynamic and thermodynamic temperatures etc. simply a distraction?

Line 86-87: replace "heat conductance" by "thermal conductivity". This sentence is not fully correct, because soil thermal properties themselves depend on soil moisture content.

Line 87-89: Quoting these values is okay, but you need to make clear over what time period these are. Are these instantaneous values, or daily averages?

Line 99 -102: You seem to be comparing apples and pears here? I.e., a mechanistic G model with an analytical SEB model? This feels a bit contrived.

Line 103: What is meant with "Recognizing the significant conclusions of..". ? Did you mean "Based on/in light of the conclusions.."?

Line 105: What exactly is meant by "...complement the overarching gaps in SEB modelling"? Line 107: Is an "analytical ET model" the same as a "SEB-model'? ("new coupled G-SEB model"). And how does this fit with "Remote sensing-based ET models". The whole introduction has been written in this very woolly fashion.

Line 117: "...any information of deep soil temperature or daily temperature amplitude". Do we need deep soil temperature to calculate an estimate of G? Also: what temperature amplitude are you actually referring to here?

Line 127-128: ..."When LE is reduced due to soil moisture dry-down and water stress, both G and TS tend to show rapid rise".. A rapid rise over what period? During the day? A season a year? Multiple years? Another illustration of the imprecise language used throughout this manuscript.

Line 133: What is meant by "day-night TS"? Is this day minus night TS? Is this not the same as amplitude (x 2)?

Line 135: It should be "has so far"

Line 157-158: What are these "contemporary empirical models'? Also: Later on you say that you are not comparing to SEB models (line 167). Then why exactly were these discussed in so much detail in the introduction?

Line 162: Most of the readers will not have heard of mulga vegetation. Can you explain it briefly between brackets?

**2 Study area and datasets**

Line 173: "The present study was conducted at nine flux tower sites". To me this means that you were running your models while being physically based at these sites. Can you not say".. used data from nine flux tower sites.."?

Line 181-182: Did you mean "The fetch-to-height ratios of EC towers ? and what is meant with "representing 90% of fetch area". Please read a paper or 2 on these topics and express it properly. Perhaps you meant that 90% of the vegetated area was within the footprint of the mast? Or something along those lines?

Line 192: This reads as if there were 4 towers in total.. .
Line 195: "on privately owned land". Is this relevant??
Line 227: What does it mean "SEB measurements were carried out"..? Did you mean EC measurements? Or Rn too? Certainly not G because one can't measure this above ground.
Line 228-231: These measurements heights mean nothing if you don't also give the vegetation heights...

Line 236: soil temperature at what depths?
Line 256: What is meant by "noon-night land surface temperature" and why is this not apparent from the entries in Table 2? Why not just say "LST at 1.30pm and am"?

Line 269: Why exactly do you need land surface emissivity and albedo for that matter? I guess to calculate Rn? You need to refer forward to Chapter 3 then, or describe the methods first, then the data?

Line 296: What are "moisture constants"?

Line 290: there is a long list here of 5 noteworthy features of the STICS-TI model, but some of them are phrased inaccurately and it is hard to picture what is actually meant, because the theory has not yet been presented... Maybe describe the theory first?

Line 308-309: what exactly is meant by "surface soil temperature amplitude (within 0.1 m from the soil top"? Is this any soil temperature between the soil surface at 10 cm depth? Or the integrated/average soil temperature?

Line 310: I thought it was 1.30?

Line 327-328: ... "decreases with depth to become close to zero until the damping depth where soil temperature is almost invariant through day-night called deep soil temperature". This is not my soil physical understanding? The amplitude is still 30% or so at damping depth?

Line 339-340: ..." TSTmin is thus close to deep soil temperature as well as minimum soil temperature of other sub-surface soil layers" . What evidence do you have for this statement? I am not sure that this (always) holds?

Line 340-342:.. "Both TSTmin and TSTmax represent ....lower and upper boundary conditions of soil heat flux conducting through topsoil at noontime" I do not understand this statement? The soil heat flux will be determined by the gradient in soil temperatures at two depths, at the same time??

Line 379: I think you mean "soil moisture contents", not "soil moisture constants"? Or did you mean to say that they are parameters that remain constant?

Line 393-410: Despite the fact that I have reviewed the STICS approach a number of times (and each time I expressed my bewilderment at the 'moisture equation') I still do not understand how you estimate M (Eq. 9), that is clearly a below-ground soil moisture related equation, for variables that are related to atmospheric moisture that is above ground? Also, why do you rearrange Eq. 6 in terms of M' if you are then still deriving M in Section 3.1.1.3?

Line 416: Le and H are not conductances? They are fluxes! What is meant here? I believe the word 'and' is missing after the comma. Also, how were the conductances calculated? No detail is given about this?

Line 429 (Figure 3). Is surface emissivity not used to calculate net radiation?

**Results**

Line 481: I am not sure that I fully understand the caption for Fig. 5b. How are the different years denoted?

Line 520: Figure 7. It is actually not clear to me what these data are exactly. These are instantaneous values but for what time? You have used noon and midnight $T_s$ values here, but your $G_i$ cannot be a daily average because its G values would be much smaller. You mention noon time in the text, so I guess this must be it.

Figures 10 & 11 look promising, but of course all these efforts only give you one value of the fluxes (around noon) for each day. How useful is this?

**Discussion**

Should Fig. 12 be part of the results section? Also why are you not plotting thermal intertia on the y-axis versus M? That would make more sense? Finally why are we not seeing the flattening off of the curve at large M values? And why is there so much scatter? Because of the different soil types at the different sites?

Line 685-687: "...this is the first ever implementation of a coupled G-SEB model that does not require any empirical parameterization of aerodynamic and canopy-surface conductance". I am somewhat confused by this statement. I thought you were calculating $g_A$ and $g_S$? I see equations for these in the Appendices?

---

## Author Comment (AC1)

**We would like to thank Reviewer#1 for the detailed comments. Please find below our responses.**

*RC 1: Line 73-74: I have two issues with this part of the sentence "....which makes it a crucial variable for estimating sensible heat flux (H) ET through the SEB models". What is meant by "sensible heat flux (H) ET"? Is there a word missing here? Also: what are " the SEB models" exactly? You mention an "analytical surface energy balance (SEB) model" in line 71, but that is a more specific model.*

**AC:** We will modify the sentence to read as "which makes it a crucial variable for estimating sensible heat flux (H) and ET through the SEB models". The word 'and' was missing and will be added in the revised version. SEB models are Surface Energy Balance models.

Analytical surface energy balance model is a specific SEB model. We will remove 'SEB' from the parentheses in the text "analytical surface energy balance (SEB) model"

*RC 2: Line 79-80: Remove the hyphen in "extra-resistance". Also, it is often called "excess resistance", see e.g., Verhoef et al., (1997), and it relates to the differences in roughness lengths for momentum and heat/water vapour. Is this worth mentioning here?*

**AC:** We will correct it as 'excess resistance' in the revised version. We will also mention that the excess resistance was introduced to accommodate the differences between roughness lengths for momentum and heat/water transfers.

*RC 3: Line 83: how is the aerodynamic conductance (gA) related to the extra resistance? According to the Appendix it is not, so is all this discussion here about roughness lengths, aerodynamic and thermodynamic temperatures etc. simply a distraction?*

**AC:** The corrections related to RC 2 will take care of this suggestion.

*RC 4: Line 86-87: replace "heat conductance" by "thermal conductivity". This sentence is not fully correct, because soil thermal properties themselves depend on soil moisture content.*

**AC:** We will replace 'heat conductance' with 'thermal conductivity'. We will modify the text as 'soil thermal properties such as thermal conductivity and heat capacity, which vary with mineral, organic and soil water fractions.'

*RC 5: Line 87-89: Quoting these values is okay, but you need to make clear over what time period these are. Are these instantaneous values, or daily averages?*

**AC:** These are instantaneous time scale, and we will mention this in the revised version.

**_RC 6_**: Line 99 -102: You seem to be comparing apples and pears here? i.e., a mechanistic G model with an analytical SEB model? This feels a bit contrived.

**AC:** We are not comparing mechanistic G model with analytical SEB model. We mentioned about coupling a mechanistic G model with an analytical surface energy balance model for estimating evaporation. Thermal remote sensing-based evaporation estimation through SEB models commonly use empirical sub-models of G and these G sub-models are run in a stand-alone mode. Despite the utility of mechanistic G models is demonstrated in different studies, no thermal remote sensing-based evaporation study attempted to couple such a mechanistic G model with a SEB model. Therefore, the major goal of this work is to couple the mechanistic G model of Murray and Verhoef (2007) with an analytical surface energy balance model, STIC. This text will be added in line 99 – 102.

**_RC 7_**: Line 103: What is meant with "Recognizing the significant conclusions of..". ? Did you mean "Based on/in light of the conclusions.."?

**AC:** Yes. We agree to replace "Recognizing the significant conclusions of.." with "Based on/in light of the conclusions.."

**_RC 8_**: Line 105: What exactly is meant by "...complement the overarching gaps in SEB modelling"?

**AC:** The overarching gaps are, (i) accurate estimation of G and ET in sparse vegetation, (ii) testing the utility of coupling a thermal inertia-based G model with analytical SEB model for accurately estimating G and ET, and (iii) detailed evaluation of a coupled G-SEB model at the ecosystem scale. We will mention this before the sentence containing text "……complement the overarching gaps in SEB modelling". We will replace the word 'complement' by 'mitigate' in the revised version.

**_RC 9_**: Line 107: Is an "analytical ET model" the same as a "SEB-model'? ("new coupled G-SEB model"). And how does this fit with "Remote sensing-based ET models". The whole introduction has been written in this very woolly fashion.

**AC**: We will modify the text "analytical ET model" with "analytical SEB model". The analytical SEB model is STIC, and in the present study, STIC is coupled with a TI-based mechanistic G model. This coupled SEB-G model is known as STIC-TI model. We will mention this after line 107 and before the start of new paragraph.

We will modify the text **'**Remote sensing-based ET models' as **'**Remote sensing-driven SEB models for evapotranspiration (ET) estimation'.

**_RC 10_**: Line 117: "...any information of deep soil temperature or daily temperature amplitude". Do we need deep soil temperature to calculate an estimate of G? Also: what temperature amplitude are you actually referring to here?

**AC:** We will modify the text as "…..any information on soil temperature". The TI-based G model does not require any deep soil temperature and it requires daily amplitude of surface soil temperature to compute noontime instantaneous G for a given day.

**_RC 11_**: Line 127-128: "When LE is reduced due to soil moisture dry-down and water stress, both G and TS tend to show rapid rise".. A rapid rise over what period? During the day? A season a year? Multiple years? Another illustration of the imprecise language used throughout this manuscript.

**AC:** We mean here rapid intra-seasonal rise of G and $T_S$. We will modify the text as: "When LE is reduced due to soil moisture dry-down and water stress, both G and $T_S$ tend to show rapid intra-seasonal rise"

**_RC 12_**: Line 133: What is meant by "day-night TS"? Is this day minus night TS? Is this not the same as amplitude (x 2)? Line 135: It should be "has so far"

**AC:** We will replace day-night Ts with "noontime and nighttime Ts" to avoid confusion. Soil temperature amplitude computation for TI-based G modeling is given in section 3.1.1.1. We will replace 'is so far' with 'has so far'.

**_RC 13_**: Line 157-158: What are these "contemporary empirical models'? Also: Later on you say that you are not comparing to SEB models (line 167). Then why exactly were these discussed in so much detail in the introduction?

**AC:** Here, we intend to say conventionally used empirical G models. We will modify text as 'conventionally used empirical G models' instead of 'contemporary empirical models' to avoid confusion.

**_RC 14_**: Line 162: Most of the readers will not have heard of mulga vegetation. Can you explain it briefly between brackets?

**AC:** Mulga is the name given to woodlands and open-forests dominated by the mulga tree (_Acacia aneura_). We will explain this in the revised version.

**_RC 15_**: Line 173: "The present study was conducted at nine flux tower sites". To me this means that you were running your models while being physically based at these sites. Can you not say".. used data from nine flux tower sites.."?

**AC:** We will modify the text as "The present study was conducted using data from nine flux tower sites"

**RC 16**: Line 181-182: Did you mean "The fetch-to-height ratios of EC towers? and what is meant with "representing 90% of fetch area". Please read a paper or 2 on these topics and express it properly. Perhaps you meant that 90% of the vegetated area was within the footprint of the mast? Or something along those lines?

**AC:** The flux footprint for EC towers in India varied from 500m-1km (Bhat et al., 2019). In the present study, about 90% of the fluxes came from an area within 500 m to 1 km from the EC tower. This represents that relative contribution of the vegetated land surface area to the flux is close to 90% (Schmid, 2002; Vesala et al., 2008). The remaining few percentage of flux is coming from an area beyond the flux footprint.

**RC 17**: Line 192: This reads as if there were 4 towers in total.

**AC:** In the beginning of section 2.1, it is clearly written as "……at nine flux tower sites (four sites in India; three sites in Australia; two sites in USA) equipped with Eddy Covariance (EC) measurement systems. So, there should not be any confusion that 4 towers in total.

**RC 18**: Line 195: "on privately owned land". Is this relevant??

**AC:** We will remove the text "on privately owned land".

**RC 19**: Line 227: What does it mean "SEB measurements were carried out"..? Did you mean EC measurements? Or Rn too? Certainly not G because one can't measure this above ground.

**AC:** We intend to say measurements of H, LE fluxes from EC systems and Rn from net radiometers. We will modify the text accordingly.

**RC 20**: Line 228-231: These measurements heights mean nothing if you don't also give the vegetation heights...

**AC**: Average heights of vegetation are 1.15 m at IND-Naw, 1 m at IND-Jai, 1.23 m at IND-Sam, 1.5 m at IND-Dha, 6.5 m at AU-Asm, 15m at AU-How, 7 m at AU-Ade; 2m at US-Ton, 1.5m at US-Var.

We will add this information for each site in the revised manuscript.

**RC 21**: Line 236: soil temperature at what depths?

**AC:** -0.1m (IND-Dha); -0.15m (AU-Ade); (-0.02, -0.06m) (AU- ASM); -0.08m (AU- How)
    -0.02m, -.0.04m, -0.08m, -0.16m (US-Ton, US-Var)

This information is already included in Table A2 of Appendix.

**RC 22**: What is meant by "noon-night land surface temperature" and why is this not apparent from the entries in Table 2? Why not just say "LST at 1.30 pm and am"?

**AC:** We will mention land surface temperature at 1.30 pm and am instead of noon-night land surface temperature. We will also mention Ts at 1.30 pm and am in Table 2.

**RC 23**: Line 269: Why exactly do you need land surface emissivity and albedo for that matter? I guess to calculate Rn? You need to refer forward to Chapter 3 then, or describe the methods first, then the data?

**AC:** We needed land surface emissivity and albedo for computing Rn. The Rn model is described in Appendix B. In addition, albedo is required to develop model for soil temperature amplitude. This has been mentioned in section 4.1.

We have used products of MODIS Aqua LST, Emissivity, albedo, NDVI as key variables to SEB modeling. As suggested by R1, it will be appropriate to keep a brief account of their generation / retrieval process as sub-section 3.2 of section 3. In that case present sub-section of 3.2 will be 3.3. Therefore, the relevant text given in sub-section 2.2.2 will be shifted to new sub-section 3.2.

**RC 24:** Line 296: What are "moisture constants"?

**AC:** We mean soil moisture characteristic limits. We will modify the text accordingly.

**RC 25:** Line 290: there is a long list here of 5 noteworthy features of the STICS-TI model, but some of them are phrased inaccurately and it is hard to picture what is actually meant, because the theory has not yet been presented... Maybe describe the theory first?

**AC:** We agree that those 5 noteworthy features should be presented after describing the theory. We will transfer the sentences from line 290-297 after the section 3.2.1. We will modify the points as follows.

The noteworthy features of STIC-TI are: (1) estimating G by modifying the mechanistic MV2007-TI model using noon and midnight TS information from thermal remote sensing observations available through polar orbiting satellite platform (e.g. MODIS Aqua), (2) coupling the mechanistic MV2007-TI G model with STIC1.2 to simultaneously estimate surface moisture availability (M), G, and SEB fluxes, (3) introducing water stress information in G to better constrain the aerodynamic and canopy-surface conductances as well as the SEB fluxes, (4) derivation of amplitude of ecosystem-scale surface soil temperature (from top soil to 0.1 m soil depth).

**RC 26**: Line 308-309: what exactly is meant by "surface soil temperature amplitude (within 0.1 m from the soil top"? Is this any soil temperature between the soil surface at 10 cm depth? Or the integrated/average soil temperature?

**AC:** It is average soil temperature between surface to 10 cm depth.

**RC 27**: Line 310: I thought it was 1.30?

**AC:** It is 1.30 indeed. We will correct this in the revised version.

**RC 28**: Line 327-328: ... "decreases with depth to become close to zero until the damping depth where soil temperature is almost invariant through day-night called deep soil temperature". This is not my soil physical understanding? The amplitude is still 30% or so at damping depth?

**AC**: We agree with reviewer suggestion on this. We will modify the text to read as "where the amplitude is maximum at the surface and it gradually decreases with depth to become 37% of surface amplitude until the damping depth (Hillel, 1982)." However, at deeper depths, soil temperatures remain constant with time and do not show much fluctuations as compared to surface or near-surface soil temperatures. This invariant soil temperature is called deep soil temperature (Mihailovic et al., 1999).  We will add this sentence after the above-mentioned quoted text.

The soil depth of deep soil temperature is higher than the damping depth.

**RC 29**: Line 339-340: ..." TSTmin is thus close to deep soil temperature as well as minimum soil temperature of other sub-surface soil layers". What evidence do you have for this statement? I am not sure that this (always) holds?

**AC:** We agree with this comment. This sentence will be deleted in the revised version.

**RC 30**: Line 340-342: "Both TSTmin and TSTmax represent ....lower and upper boundary conditions of soil heat flux conducting through topsoil at noontime" I do not understand this statement? The soil heat flux will be determined by the gradient in soil temperatures at two depths, at the same time??

**AC:** This sentence will be deleted in the revised version.

**RC 31**: Line 379: I think you mean "soil moisture contents", not "soil moisture constants"? Or did you mean to say that they are parameters that remain constant?

**AC:** We mean $\theta_{fc}$, $\theta_*$, $\theta_{wp}$ are the characteristic volumetric soil moisture content at field capacity, maximum retentive capacity and permanent wilting point corresponding to soil water potentials at

-33 kPa, 0 kPa and -1500 kPa, respectively. We will modify the text in Line 379 in the revised version.

**_RC 32_**: Line 393-410: Despite the fact that I have reviewed the STICS approach a number of times (and each time I expressed my bewilderment at the 'moisture equation') I still do not understand how you estimate M (Eq. 9), that is clearly a below-ground soil moisture related equation, for variables that are related to atmospheric moisture that is above ground? Also, why do you rearrange Eq. 6 in terms of M' if you are then still deriving M in Section 3.1.1.3?

**AC**: Equation 9 is a classical expression for estimating soil moisture availability (M) from the observations of actual soil moisture, soil moisture at the field capacity and permanent wilting point, respectively. The thermal inertia equation (eq. 6) was rearranged in terms of M to arrive at equation 9, and M is unknown here. Therefore, the estimation of M is done according to the procedure mentioned in 3.1.1.3, through which STIC is coupled with TI-based estimation of G. In the revised version, we will make this description more explicit.

**_RC 33_**: Line 416: Le and H are not conductances? They are fluxes! What is meant here? I believe the word 'and' is missing after the comma. Also, how were the conductances calculated? No detail is given about this?

**AC**: We intend to say that the initial estimates of conductances, LE, and H are obtained. The word 'and' is missing indeed. We will modify this section description as follows:

From the initial estimates of $G_i$(eq. 2) and $RN_i$ (equations in Appendix B), initial estimation of $LE_i$ and $H_i$ were obtained through the PMEB equation. **The initial conductances for estimating H and LE through the PMEB equation was obtained by solving the state equations as described in the Appendix.** The process was then iterated by updating $T_{0D}$ [$T_{0D} = T_D + (\gamma LE/\rho c_p g_a s_1)$] and M in every time step (as mentioned in Mallick et al., 2016, 2018a), and re-estimating $G_i$(using eq. 3), net available energy ($RN_i-G_i$), conductances, $LE_i$, and $H_i$, until stable estimates of $LE_i$ were obtained. The conceptual block diagram and algorithm flow of STIC-TI is shown in Fig. 3a and Fig 3b, respectively.

**_RC 34_**: Line 429 (Figure 3). Is surface emissivity not used to calculate net radiation?

**AC:** Yes, surface emissivity was used to calculate net radiation. In the revised version, we will incorporate surface emissivity in the block showing inputs in Figure 3.

**_RC 35_**: Line 481: I am not sure that I fully understand the caption for Fig. 5b. How are the different years denoted?

**AC:** We will modify the caption for Fig. 5b as "Validation of the ecosystem-scale estimates of A from the above functions over different ecosystems"

**_RC 36_**: _Line 520: Figure 7. It is actually not clear to me what these data are exactly. These are instantaneous values but for what time? You have used noon and midnight Ts values here, but your Gi cannot be a daily average because its G values would be much smaller. You mention noon time in the text, so I guess this must be it._

**AC:** These are noontime (1.30 pm) $G_i$ estimates. We will modify in the caption of Figure 7 accordingly to read as 'Validation of STIC-TI derived noontime (1.30 pm) $G_i$ estimates with respect to in-situ measurements in different ecosystems.

**_RC 37_**: _Figures 10 & 11 look promising, but of course all these efforts only give you one value of the fluxes (around noon) for each day. How useful is this?_

**AC:** Ideally diurnal latent heat fluxes are needed to estimate daily evapotranspiration (ET). In a cloudless day, the diurnal variation of latent heat flux between sunrise to sunset has sinusoidal pattern with its amplitude or peak corresponding to around noontime. Several studies revealed (Mallick et al, 2009, Bhattacharya et al, 2010) that the noontime latent heat flux computed through SEB model from polar orbiting remote sensing observation during a day can be converted to daily ET (in terms of mm depth of water) through sinusoidal integration. Validation of daily ET through such method is found to be close to in situ ET measurements. Therefore, validation of noontime LE and H fluxes and their accuracies are also very useful.

**_RC 38_**: _Should Fig. 12 be part of the results section? Also why are you not plotting thermal intertia on the y-axis versus M? That would make more sense? Finally why are we not seeing the flattening off of the curve at large M values? And why is there so much scatter? Because of the different soil types at the different sites?_

**AC:** Figure 12 is put in the discussion to highlight the reasons for reduced sensitivity of G due to Ts uncertainties. However, we agree with the suggestion to put thermal inertia in y-axis of Fig 12a. Overall, we will modify Figure 12 in the revised version which will be as follows:

    (i)    We will merge Fig. 12 (a) and (b): thermal inertia in y-axis and noontime Ts in x-axis with color segregation according to M. The plot of thermal inertia versus Ts is flattening due to the effects of M and we see so much scatter due to different soil types across the sites.

    (ii)    We will merge Fig. 12 (c) and (d): G in y-axis and Amplitude (A) in x-axis with color segregation according to the thermal inertia.

The description in the text will be modified accordingly.

**_RC 39_**: _Line 685-687: "...this is the first ever implementation of a coupled G-SEB model that does not require any empirical parameterization of aerodynamic and canopy-surface conductance". I am somewhat confused by this statement. I thought you were calculating g_A and g_S? I see equations for these in the Appendices?_

**AC:** We are directly estimating ga and gs by solving the state equations as mentioned in the manuscript (Line 1093 to 1124). While the estimation of ga does not require any parameterization of surface roughness and atmospheric stability, estimation of gs does not involve any empirical

function or look-up table for biome or plant functional attributes. **We will modify the sentence in L685-687 for a better clarity.**

**References:**

Bhattacharya, B.K., Mallick, K., Patel, N.K., & Parihar, J.S. (2010). Regional clear sky evapotranspiration over agricultural land using remote sensing data from Indian geostationary meteorological satellite. Journal of Hydrology, 387, 65-80.

Hillel, D (1982). Introduction to Soil Physics, San Diego, US, ISBN 9780123485205.

Mallick, K., et al. (2009). Latent heat flux estimation in clear sky days over Indian agroecosystems using noontime satellite remote sensing data. Agricultural and Forest Meteorology, 149 (10), 1646–1665, doi:10.1016/j.agrformet.2009.05.006.

Mihailovic, D. T., Kallos, G., Aresenic, I.D., Lalic, B., Rajkovic, B. and Papadopoulos, A. (1999). Sensitivity of soil surface temperature in a Force-Restore Equation to heat fluxes and deep soil temperature. International Journal of Climatology, 19, 1617-1632.

Schmid, H.P. (2002). Footprint modelling for vegetation atmosphere exchange studies: a review and perspective. Agricultural and Forest Meteorology 113, 159-183.

Vesala, T., Kljun, N., Rannik, U., Rinne, A. Sogachev, Markkanen, T., Sabelfeld, K., Foken, T. and Leclerc, M.Y. (2008). Flux and concentration footprint modelling: State of the art. Environmental Pollution, 152, 653-666

---

## Author Comment (AC2)

**REVIEWER 2:**

**We thank Reviewer#2 for the detailed comments. Please find the responses below.**

_**RC_1**_: The manuscript focusses on arid and semi-arid ecosystems where G can be a large component of the surface energy balance. Further, the authors have reported that the LE and G is sensitive to the LST observations. Over such arid or semi-arid ecosystems, it has been reported that the MYD21 LST (based on the Temperature Emissivity Separation algorithm) product might work better than the MYD11 product. May I suggest the authors to test if MYD21 product is better suited than the MYD 11 product for this coupled model?

**AC:** We have carried out the sensitivity analysis of the new coupled model (STIC-TI). It showed that Both LEi and Hi were sensitive to $T_S$ to the order of 2 – 29% (LEi) and 5 – 35% (Hi) for $T_S$ uncertainty of ±0.5 – 2.5 K from its mean values (Table 3). This provides a quantification of improvement of flux estimates with better accuracy of Ts. Therefore, the answer to reviewer's suggestion is already built-in the sensitivity analysis. However, in subsection 5.2 we will mention about better expected accuracy of Ts from MYD21 over arid and semi-arid ecosystems that can lead to little higher accuracy in LEi.

**In the revised version, we will incorporate model results by using MYD21 product particularly over the arid and semi-arid sites.**

_**RC 2**_: Among the different components of surface energy fluxes, the in-situ observations of G and soil temperature are indeed strictly point observations when compared with LE, H or even Rn. In Equation 12, I think the ground observed soil temperature amplitude and satellite LST are combined towards developing the regression relationship. Is this valid considering that there can be varying soil conditions that can affect the amplitude of the soil temperature in a pixel?

**AC:** Soil temperature amplitude and noon-night LST as well as noontime surface albedo are taken from soils over varying soil types of Northern hemisphere (Indian site) and Southern hemisphere (Australian site), respectively. Based on the soil temperature amplitude model, STIC-TI estimate of G is independently validated over several US sites having varying soil types and vegetation cover for multiple years. This showed a good accuracy of G estimates. Therefore, we conclude that the proposed approach is valid for a wide range of soil, plant, and climate types.

_**RC 3**_: Further, satellite LST can represent soil temperature only over bare surface. Till what level of fraction vegetation cover, can we use LST to estimate the amplitude of soil temperature? Can the results in Section 4.1 be separated in terms of fc to see if there is any decrease in accuracy in estimating A with increasing vegetation cover?

**AC:** The present soil temperature amplitude model has been developed from noon-night LST difference as well as surface albedo for mean vegetation fraction around 0.5. In the revised version, **we will segregate the temperature amplitude evaluation and mention the results for both fc <0.5 and fc >0.5.**

_**RC 4**_: The STIC-TI model uses difference in LST between day and night. MODIS observations of LST can vary significantly in terms of viewing angle or increase in GIFOV due to the ground point being away from the nadir between day and night observations. Is this taken into consideration? Can the results be separated for LST observations made with similar viewing angles and different viewing angles during day and night?

**AC: This is an excellent suggestion. In the revised version, we will mention the errors in LE and H with respect to the view zenith angle.**

_RC 5_: As mentioned clearly, the method by Bastiaanssen (1998) was able to retrieve G with accuracy closer to that of the STIC-TI model. Is this due to the fact that both the models use LST, NDVI and albedo as key inputs to the 'G' model? Also, what benefit does the STIC-TI brings over the model by Bastiaanssen? We can look at the diurnal variation of LST to get more information about the ecosystem processes than the soil temperature amplitude, I think.

**AC:** This is a very good observation by the Reviewer followed by suggestion. The modelled soil temperature amplitude can be used to quantify diurnal variation of soil temperature and G fluxes which in turn can help to quantify diurnal latent heat fluxes and plant water stress. Although Bastiaanssen (1998) G model uses similar inputs and provides comparable accuracy of G estimates as STIC-TI, the former is not applicable for diurnal G, LE and water stress simulations. Although diurnal simulation of fluxes from STIC-TI is not within the scope of this paper, we will include the above future scope of STIC-TI in the sub-section 5.2.

_RC 6_: There can be a considerable difference between the diurnal behavior of soil temperature at the surface (skin temperature) and soil temperature at 10 cm depth. Further, for modelling G, the soil skin temperature is the actual boundary condition. Can the authors please explain the reason for using soil temperature measured over 10 cm depth as the boundary condition rather than the skin temperature?

**AC:** Soil temperature measured at different depths within top 10 com soil layer was averaged and considered as representative surface soil temperature (0 – 10 cm). Studies also showed that LST carries some signal beneath the skin of the surface. We will clarify this in section 3.1.1.1 of the revised version of the manuscript.

_RC 7_: In addition, over two sites (Ind-Dha and AUS-Ade), soil temperature observations are done at or below 10 cm depth. How this has been used in the study in place of soil temperature up to 10 cm depth? When there are more than one soil temperature observations within 10 cm depth (AU-ASM, US-Ton and US-Var) were they averaged?

**AC:** Soil temperature observations at different depths within 10 cm were averaged (AU-ASM, US-Ton and US-Var). For Ind-Dha and AUS-Ade, single-depth (10 cm) soil temperature observation was used. We will state this explicitly in section 3.1.1.1 of the revised version.

**OTHER COMMENTS:**

_RC_: Lines 103-104, Please mention the key conclusions of the studies that you are referring to and explain why this study is needed.

**AC:** We will modify this portion in the revised manuscript mentioning the key conclusions from the previous studies and explain why the present study is needed

_RC_: Lines 181-182 The fetch ratio of …90% of fetch area: This sentence is not clear. Are the authors trying to say that the assumed fetch area around the towers accounted for 90% of the energy fluxes? Figure 1 caption can be shortened and made precise.

**AC:** The flux footprint for EC towers in India varied from 500m-1km (Bhat et al. 2019). In present study, about 90% of fluxes came from an area within 500 m to 1 km from the EC tower. This represents that relative contribution of the vegetated land surface area to the flux is close to 90% (Schmid, 2002; Vesala et al, 2008). The remaining few percentage of flux is coming from an area beyond flux footprint.

We will shorten the Figure 1 caption by removing the text '(Supplement) map in PDF (Institute for Veterinary Public Health)' in the revised version.

_**RC**_: Lines 237-238: Non-availability of G over Indian sites is an issue. The authors have explained about the this at the end of the discussions section. But any studies supporting the use of unclosed energy budget from EC observations and lack of G observations will be helpful here.

**AC:** In the micrometeorological research, it is well known that the turbulent fluxes (H + LE) measured with eddy covariance (EC) systems do not usually equal the available energy (Rn-G) on instantaneous, hourly, daily time scale. However, it tends to be show closure at longer time scales such as weekly, monthly and season. Hence, qualitative knowledge of the impact of different vegetation types, and climatic variables on this 'non-closure' on shorter time scale is essential. The study by Dare-Idowu et al. (2021) analyzed a unique database of EC flux measurements covering 8 growing seasons of 3 crops (maize, wheat, and rapeseed) cultivated over two close agricultural sites (FR-Lam and FR-Aur) in southwestern France. For data analysis, some dry and wet cropping seasons of the same crop type were selected; then, their phenological stages were identified to investigate their effect on the energy balance closure (EBC), and flux partitioning. The results showed that the systematic effect of each site on the EBC was stronger than the influence of crop type and stage, as EBC was generally higher at FR-Aur (82%) than at FR-Lam (67%), even for the same crop type. The assessed effect of rainfall, and phenological stages on energy partitioning revealed that during the wet seasons, over 42% of the net radiation (Rn) was accounted for by the latent heat flux (LE), which was 9% higher than the recorded LE in the dry year during the active vegetation period. Similarly, the ground heat flux (G) was observed to be very sensitive to vegetation; G accounted for 30% of Rn when vegetation was low, whereas at the peak of vegetation, it fell below 16% due to canopy shading. Closure was also assessed under various atmospheric stability conditions and wind sectors, and it was observed to be higher under unstable conditions, and in prevailing wind directions. Analysis of the sensible heat advection (AH) revealed that AH accounts for more than half of the imbalance at both sites.

The unclosed energy budget from EC observations is helpful to model flux partitioning (LE/Rn, H/Rn, G/Rn) with respect to vegetation/crop growth characteristics or biophysical properties for a given crop type at local scale. Especially, where G observations are lacking such as in many Indian sites, the TI-based G model can be used to fill up the gaps of G observations and estimate noontime G/Rn to simulate vegetation / crop growth characteristics.

Though the use of unclosed energy balance is not the main emphasis of this paper, we will mention the above findings and conclusions of Dare-Idowu et al (2021) briefly in section 5.3 where SEB closure is discussed.

_**RC**_: Lines 268-269: The land surface emissivity was estimated from land cover types, atmospheric water vapour and …retrieval': I think MODIS emissivity is estimated from land cover

type and anisotropy factor. The other variables mentioned here are used in the retrieval of LST with the split window algorithm. Please rewrite the sentence.

**AC:** Yes, we will rewrite the sentence as suggested by the reviewer.

*RC*: Lines 271-272: How albedo was estimated from the white sky and black sky albedo available in the MCD43 product?

**AC:** Actual albedo is a value which is interpolated between white-sky and black-sky albedo as a function of the fraction of diffuse skylight which is itself a function of the aerosol optical depth.

**We will add this text in the new sub-section (in section 3.1) on input data generation (as also suggested by R1)**

*RC*: Line 273-274: The line 'eight-day compositing … (MYD11A2)' appears little confusing. Do you mean that the compositing dates were obtained from MYD11 product? Please rewrite clearly.

**AC:** The 8-day average values of clear-sky LSTs retrieved from MYD11A2 data were used (Source: https://vip.arizona.edu/documents/viplab/MYD11A2.pdf). We will clarify this in L273 – 274 of the revised manuscript.

*RC*: Lines 295-296: Is point (4) a contribution of this study? I think that is the nature of the TI model selected for coupling. Also, what is meant by 'moisture constants'?

**AC:** We agree that point (4) is not a contribution of this study but is a feature of STIC-TI. We will delete this point to remove confusion.

We mean soil moisture characteristic limits. We will modify the text accordingly.

*RC*: Lines 303 – 304: Will the amplitude vary with each harmonic component? If yes, the A has to be replaced with An.

**AC:** In the present study, we have considered a single pair (noon-night corresponding to 1:30 pm and 1:30 am) of MODIS aqua LST data for amplitude. Therefore, 'A' is appropriate here. If we take multiple (n) LST pairs within a day, then 'An' will be appropriate.

*RC*: Lines 310-312: How $\varphi_n$' can be taken as zero and k can be assumed to be one when noon-night data is considered? Please explain clearly. In equation (2), the first term in the inner bracket of sin term is 'n$\omega$t'. However, this becomes $\omega$t' in eqn. (3). Are the symbols consistent here?

**AC:** Since we have considered a single pair (noon-night corresponding to 1:30 pm and 1:30 am) of MODIS aqua LST data for amplitude in the present study, the phase shift ($\varphi'_n$) is taken as zero and number of harmonics is taken as one (k=1) for estimating noontime $G_i$. We are not doing it here diurnally. **The explanation will be given in Line 310 – 312.**

**The symbols are consistent between Eq. (2) and (3).**

**_RC_**: Lines 331-332: It is mentioned that figure 2 contains theoretical and observed trajectories of soil temperature. However, only one curve is seen in the figure and I am not sure if it is theoretical or observed. Please check.

**AC**: This curve is based on observed soil temperature variation. We will clarify it in the revised manuscript.

**_RC_**: Lines 339-340: How TSTmin can be assumed to be closer top deep soil temperature as well as minimum soil temperature of other layers? Any data/studies to show this?

**AC:** This sentence will be deleted in the revised version to remove confusion.

**_RC_**: Lines 342-344: Have you used the OzFlux sites data to create Figure 2 or for the entire analysis?

**AC:** We have used OzFlux sites data to create Figure 2 to show example of diurnal variation of surface soil temperature.

**_RC_**: Line 438: 1.5 K repeats twice. Please check.

**AC:** The sensitivity was assessed by varying noon-time $T_S$ by ±0.5 K, ±1.0 K and ±1.5 K (keeping nighttime $T_S$ constant so that amplitude can vary automatically). We will modify accordingly in the revised version.

**_RC_**: Line 579: What is 'clothesline effect'?

**AC:** The 'clothesline effect' is the effect on soil moisture when warm, dry air moves by advection through vegetation (e.g. field crops or forest). Close to the point of entry the warm air warms the soil-canopy system thus increasing the rate of evapotranspiration and drying the soil. Deeper into the vegetation canopy the temperature of moving air falls.

We will revise the text in the line 579 in the following way:

The intermittent EF spikes in the soil moisture dry down phase could be due to enhanced latent heat fluxes through evapotranspiration due to passage of warm air transported by micro-advection from surrounding vegetation causing additional evaporation than expected. This is known as the 'clothesline effect' which frequently occurs in semi-arid and arid ecosystems.

**_RC_**: Figure 12: Expect the relationship between M and thermal inertia in the first plot, there seem to be no direct link between other variables in the other plots. What is the key take home message from this figure?

**AC:** We will modify Figure 12 in the revised version considering comments from Reviewer 1 also. The revision will be as follows:

(i) We will merge Fig. 12 (a) and (b): thermal inertia in y-axis and noontime Ts in x-axis with color segregation according to M. The plot of thermal inertia versus Ts is flattening due to the effects of M and we see so much scatter due to different soil types across the sites.

(ii) We will merge Fig. 12 (c) and (d): G in y-axis and Amplitude (A) in x-axis with color segregation according to the thermal inertia.

The description in the text will be modified accordingly**.**

**_RC_**: Line 627: …'range of 2-3 K with a standard deviation of 0.009'. I think the 0.009 corresponds to the standard deviation in surface emissivity in MODIS product. Please check.

**AC:** The standard deviations of errors in retrieved emissivity in bands 31 and 32 are 0.009, and the maximum error in retrieved LST values falls within 2-3 K. We will correct this in the revised version.

Reference:

Dare-Idowu, O., Brut, A., Cuxart, J., Tallec, T., Rivalland, V., Zawilski, B., Ceschia, E. and Jarlan, L. (2021). Surface energy balance and flux partitioning of annual crops in southwestern France. Agricultural and Forest Meteorology, 308 – 309, 108529, https://doi.org/10.1016/j.agrformet.2021.108529.

Schmid, H.P. (2002). Footprint modelling for vegetation atmosphere exchange studies: a review and perspective. Agricultural and Forest Meteorology 113, 159-183.

Vesala, T., Kljun, N., Rannik, U., Rinne, A. Sogachev, Markkanen, T., Sabelfeld, K., Foken, T. and Leclerc, M.Y. (2008). Flux and concentration footprint modelling: State of the art. Environmental Pollution, 152, 653-666.

---

## Author Response (AR1)

**Reviewer 1: We would like to thank Reviewer#1 for the detailed comments. Please find below our responses. All the line numbers correspond to the clean version of the manuscript. A track change version is also submitted.**

*RC 1: Line 73-74: I have two issues with this part of the sentence "....which makes it a crucial variable for estimating sensible heat flux (H) ET through the SEB models". What is meant by "sensible heat flux (H) ET"? Is there a word missing here? Also: what are " the SEB models" exactly? You mention an "analytical surface energy balance (SEB) model" in line 71, but that is a more specific model.*

**AC:** The text in the section 1 has been revised in line 59-62 and in line 66-74. SEB models are Surface Energy Balance models. Analytical surface energy balance model is a specific SEB model. We have used 'SEB' only to represent surface energy balance (SEB) model

*RC 2: Line 79-80: Remove the hyphen in "extra-resistance". Also, it is often called "excess resistance", see e.g., Verhoef et al., (1997), and it relates to the differences in roughness lengths for momentum and heat/water vapour. Is this worth mentioning here?*

**AC:** The relevant text has been deleted in the revised version to remove confusion.

*RC 3: Line 83: how is the aerodynamic conductance (gA) related to the extra resistance? According to the Appendix it is not, so is all this discussion here about roughness lengths, aerodynamic and thermodynamic temperatures etc. simply a distraction?*

**AC:** The relevant text has been deleted in the revised version to remove confusion.

*RC 4: Line 86-87: replace "heat conductance" by "thermal conductivity". This sentence is not fully correct, because soil thermal properties themselves depend on soil moisture content.*

**AC:** We have replaced 'heat conductance' with 'thermal conductivity'. We have modified the text in line 82-83 as 'soil thermal properties such as thermal conductivity and heat capacity, which vary with mineral, organic and soil water fractions.'

*RC 5: Line 87-89: Quoting these values is okay, but you need to make clear over what time period these are. Are these instantaneous values or daily averages?*

**AC:** These are instantaneous time scale, and we have mentioned this in the revised version in line 67.

*RC 6: Line 99 -102: You seem to be comparing apples and pears here? i.e., a mechanistic G model with an analytical SEB model? This feels a bit contrived.*

**AC:** We are not comparing mechanistic G model with analytical SEB model. We mentioned about coupling a mechanistic G model with an analytical surface energy balance model for estimating evaporation.

Evaporation estimation through SEB models commonly employ empirical sub-models of G in a stand-alone mode. Despite the utility of mechanistic G models is demonstrated in different studies (Verhoef, 2004; Murray and Verhoef, 2007; Verhoef et al., 2012), no TIR-based evaporation study attempted to couple a mechanistic G model with a SEB model.

This text has been added in line 88 – 92.

*RC 7*: Line 103: What is meant with "Recognizing the significant conclusions of..". ? Did you mean "Based on/in light of the conclusions.."?

**AC:** Yes.We have replaced "Recognizing the significant conclusions of.." in line 130 with "Based on/in light of the conclusions."

*RC 8*: Line 105: What exactly is meant by "...complement the overarching gaps in SEB modelling"?

**AC:** The text has been revised in line 131-135 to read as 'there is a need to address some of the challenges in SEB modeling, which are, (i) accurate estimation of G and ET in sparse vegetation, (ii) testing the utility of coupling a TI-based G model with an analytical SEB model for accurately estimating G and ET, and (iii) detailed evaluation of a coupled G-SEB model at the ecosystem scale.'

We have replaced the word 'complement' by 'mitigate' in the revised version in line 136.

*RC 9*: Line 107: Is an "analytical ET model" the same as a "SEB-model'? ("new coupled G-SEB model"). And how does this fit with "Remote sensing-based ET models". The whole introduction has been written in this very woolly fashion.

**AC**: We have modified the text "analytical ET model" with "analytical SEB model". The analytical SEB model is STIC, and in the present study, STIC is coupled with a TI-based mechanistic G model. This coupled SEB-G model is known as STIC-TI model. We have mentioned this in line 137-139.

We have modified the text in line 93 **'**Remote sensing-based ET models' as **'**Remote sensing-driven SEB models for evapotranspiration (ET) estimation'.

*RC 10*: Line 117: "...any information of deep soil temperature or daily temperature amplitude". Do we need deep soil temperature to calculate an estimate of G? Also: what temperature amplitude are you actually referring to here?

**AC:** We have modified the text in line 98 to 99 as "…..any information on soil temperature or daily temperature amplitude". The TI-based G model does not require any deep soil temperature

and it requires daily amplitude of surface soil temperature to compute noontime instantaneous G for a given day.

**RC 11**: Line 127-128: "When LE is reduced due to soil moisture dry-down and water stress, both G and $T_S$ tend to show rapid rise".. A rapid rise over what period? During the day? A season a year? Multiple years? Another illustration of the imprecise language used throughout this manuscript.

**AC:** We mean here rapid intra-seasonal rise of G and $T_S$.We have modified the text in line 109-110  as: "When LE is reduced due to soil moisture dry-down and water stress, both G and $T_S$ tend to show rapid intra-seasonal rise."

**RC 12**: Line 133: What is meant by "day-night TS"? Is this day minus night TS? Is this not the same as amplitude (x 2)? Line 135: It should be "has so far"

**AC:** We have replaced day-night Ts with "noontime and nighttime Ts" to avoid confusion in line 115. Soil temperature amplitude computation for TI-based G modeling is given in section 3.1.1.1. We have replaced 'is so far' with 'has so far' in line 117.

**RC 13**: Line 157-158: What are these "contemporary empirical models'? Also: Later on you say that you are not comparing to SEB models (line 167). Then why exactly were these discussed in so much detail in the introduction?

**AC:** Here, we intend to say conventionally used empirical G models. We have modified text as 'conventionally used empirical G models' in line 145 to 146 instead of 'contemporary empirical models' to avoid confusion.

**RC 14**: Line 162: Most of the readers will not have heard of mulga vegetation. Can you explain it briefly between brackets?

**AC:** Mulga is the name given to woodlands and open-forests dominated by the mulga tree (*Acacia aneura*). We have explained this in line 150 to 151 in the revised version.

**RC 15**: Line 173: "The present study was conducted at nine flux tower sites". To me this means that you were running your models while being physically based at these sites. Can you not say".. used data from nine flux tower sites.."?

**AC:** We have modified the text in line 162 as "The present study was conducted using data from nine flux tower sites"

**RC 16**: Line 181-182: Did you mean "The fetch-to-height ratios of EC towers? and what is meant with "representing 90% of fetch area". Please read a paper or 2 on these topics and express it properly. Perhaps you meant that 90% of the vegetated area was within the footprint of the mast? Or something along those lines?

**AC:** The flux footprint for EC towers in India varied from 500m-1km (Bhat et al., 2019). In the present study, about 90% of the fluxes came from an area within 500 m to 1 km from the EC tower. This represents that relative contribution of the vegetated land surface area to the flux is close to 90% (Schmid, 2002; Vesala et al., 2008). The remaining few percentage of flux is coming from an area beyond the flux footprint.

The text in the manuscript is revised accordingly in line 171 to 174.

**RC 17**: Line 192: This reads as if there were 4 towers in total.

**AC:** In the beginning of section 2.1, it is clearly written as "……at nine flux tower sites (four sites in India; three sites in Australia; two sites in USA) equipped with Eddy Covariance (EC) measurement systems. So, there should not be any confusion that 4 towers in total.

**RC 18**: Line 195: "on privately owned land". Is this relevant??

**AC:** We have removed the text "on privately owned land".

**RC 19**: Line 227: What does it mean "SEB measurements were carried out"..? Did you mean EC measurements? Or Rn too? Certainly not G because one can't measure this above ground.

**AC:** We intend to say measurements of H, LE fluxes from EC systems and Rn from net radiometers. We have modified the text accordingly in line 224 to 225.

**RC 20**: Line 228-231: These measurements heights mean nothing if you don't also give the vegetation heights...

**AC**: Average heights of vegetation are 1.15 m at IND-Naw,  1 m at IND-Jai, 1.23 m at IND-Sam, 1.5 m at IND-Dha, 6.5 m at AU-Asm, 15m at AU-How, 7 m at AU-Ade; 2m at US-Ton, 1.5m at US-Var.

We have added this information for each site in the revised manuscript in line 205 to 207.

**RC 21**: Line 236: soil temperature at what depths?

**AC:** -0.1m (IND-Dha); -0.15m (AU-Ade); (-0.02, -0.06m) (AU- ASM); -0.08m (AU- How)
    -0.02m, -.0.04m, -0.08m, -0.16m (US-Ton, US-Var)

This information is already included in Table A2 of Appendix.

**_RC 22_**: What is meant by "noon-night land surface temperature" and why is this not apparent from the entries in Table 2? Why not just say "LST at 1.30 pm and am"?

**AC:** We have now mentioned land surface temperature at 1.30 pm and am instead of noon-night land surface temperature. We have also mentioned Ts at 1.30 pm and am in Table 2.

**_RC 23_**: Line 269: Why exactly do you need land surface emissivity and albedo for that matter? I guess to calculate Rn? You need to refer forward to Chapter 3 then, or describe the methods first, then the data?

**AC:** We needed land surface emissivity and albedo for computing Rn. The Rn model is described in Appendix B. In addition, albedo is required to develop model for soil temperature amplitude. This has been mentioned in section 4.1.

We have used products of MODIS Aqua LST, Emissivity, albedo, NDVI as key variables to SEB modeling. As suggested, it is appropriate to keep a brief account of their generation / retrieval process as sub-section 3.1.3 of section 3. Therefore, the relevant text given in sub-section 2.2.2 has been shifted to new sub-section 3.1.3 in the revised version.

**_RC 24:_** Line 296: What are "moisture constants"?

**AC:** We mean soil moisture characteristic limits. We have modified the text accordingly.

**_RC 25:_** Line 290: there is a long list here of 5 noteworthy features of the STICS-TI model, but some of them are phrased inaccurately and it is hard to picture what is actually meant, because the theory has not yet been presented... Maybe describe the theory first?

**AC:** We agree that those 5 noteworthy features should be presented after describing the theory. We have transfered the text to the last paragraph of the section 3.1.2 in line 419 to 426. We have modified the points as follows:

The noteworthy features of STIC-TI are: (1) estimating G by modifying the mechanistic MV2007TI model using noon and midnight TS information from thermal remote sensing observations available through polar orbiting satellite platform (e.g. MODIS Aqua), (2) coupling the mechanistic MV2007-TI G model with STIC1.2 to simultaneously estimate surface moisture availability (M), G, and SEB fluxes, (3) introducing water stress information in G to better constrain the aerodynamic and canopy-surface conductances as well as the SEB fluxes, (4) derivation of amplitude of ecosystem-scale surface soil temperature (from top soil to 0.1 m soil depth).

**_RC 26_**: Line 308-309: what exactly is meant by "surface soil temperature amplitude (within 0.1 m from the soil top"? Is this any soil temperature between the soil surface at 10 cm depth? Or the integrated/average soil temperature?

**AC:** It is average soil temperature between surface to 10 cm depth. Modifications in the text have been made accordingly in line 306 to 309.

*RC 27*: Line 310: I thought it was 1.30?

**AC:** It is 1.30 indeed. We have corrected this in the revised version.

*RC 28*: Line 327-328: ... "decreases with depth to become close to zero until the damping depth where soil temperature is almost invariant through day-night called deep soil temperature". This is not my soil physical understanding? The amplitude is still 30% or so at damping depth?

**AC**: We agree with reviewer suggestion on this. We have modified the text in line 312 to 316 to read as "where the amplitude is maximum at the surface and it gradually decreases with depth to become 37% of surface amplitude until the damping depth (Hillel, 1982)." However, at deeper depths, soil temperatures remain constant with time and do not show much fluctuations as compared to surface or near-surface soil temperatures. This invariant soil temperature is called deep soil temperature (Mihailovic et al., 1999).  We have added this sentence after the above-mentioned quoted text.

The soil depth of deep soil temperature is higher than the damping depth.

*RC 29*: Line 339-340: ..." TSTmin is thus close to deep soil temperature as well as minimum soil temperature of other sub-surface soil layers". What evidence do you have for this statement? I am not sure that this (always) holds?

**AC:** We agree with this comment. This sentence has been deleted in the revised version.

*RC 30*: Line 340-342: "Both TSTmin and TSTmax represent ....lower and upper boundary conditions of soil heat flux conducting through topsoil at noontime" I do not understand this statement? The soil heat flux has been determined by the gradient in soil temperatures at two depths, at the same time??

**AC:** This sentence has been deleted in the revised version.

*RC 31*: Line 379: I think you mean "soil moisture contents", not "soil moisture constants"? Or did you mean to say that they are parameters that remain constant?

**AC:** We mean $\theta_{fc}$, $\theta_*$, $\theta_{wp}$ are the characteristic volumetric soil moisture contents at field capacity, maximum retentive capacity and permanent wilting point corresponding to soil water potentials at -33 kPa, 0 kPa and -1500 kPa, respectively. We have modified the text in Line 361 to 364 in the revised version.

**_RC 32_**: Line 393-410: Despite the fact that I have reviewed the STICS approach a number of times (and each time I expressed my bewilderment at the 'moisture equation') I still do not understand how you estimate M (Eq. 9), that is clearly a below-ground soil moisture related equation, for variables that are related to atmospheric moisture that is above ground? Also, why do you rearrange Eq. 6 in terms of M' if you are then still deriving M in Section 3.1.1.3?

**AC**: Equation 9 is a classical expression for estimating soil moisture availability (M) from the observations of actual soil moisture, soil moisture at the field capacity and permanent wilting point, respectively. The thermal inertia equation (eq. 6) was rearranged in terms of M to arrive at equation 9, and M is unknown here. Therefore, the estimation of M is done according to the procedure mentioned in 3.1.1.3, through which STIC is coupled with TI-based estimation of G. In the revised version, we have made this description more explicit.

**_RC 33_**: Line 416: Le and H are not conductances? They are fluxes! What is meant here? I believe the word 'and' is missing after the comma. Also, how were the conductances calculated? No detail is given about this?

**AC**: We intend to say that the initial estimates of conductances, LE, and H are obtained. The word 'and' is missing indeed. We have modified in line 400 to 408 in this section description as follows:

From the initial estimates of $G_i$(eq. 2) and $RN_i$ (equations in Appendix B), initial estimation of $LE_i$ and $H_i$ were obtained through the PMEB equation. The initial conductances for estimating H and LE through the PMEB equation was obtained by solving the state equations as described in the Appendix. The process was then iterated by updating $T_{0D}$ [$T_{0D} = T_D + (\gamma LE/\rho c_p g_a s_1)$] and M in every time step (as mentioned in Mallick et al., 2016, 2018a), and reestimating$G_i$(using eq. 3), net available energy ($RN_i$–$G_i$), conductances, $LE_i$, and $H_i$, until stable estimates of $LE_i$ were obtained. The conceptual block diagram and algorithm flow of STIC-TI is shown in Fig. 3a and Fig 3b, respectively.

**_RC 34_**: Line 429 (Figure 3). Is surface emissivity not used to calculate net radiation?

**AC:** Yes, surface emissivity was used to calculate net radiation. In the revised version, we have incorporated surface emissivity in the block showing inputs in Figure 3.

**_RC 35_**: Line 481: I am not sure that I fully understand the caption for Fig. 5b. How are the different years denoted?

**AC:** We have modified the caption for Fig. 5b as "Validation of the ecosystem-scale estimates of A from the above functions over different ecosystems"

**RC 36**: Line 520: Figure 7. It is actually not clear to me what these data are exactly. These are instantaneous values but for what time? You have used noon and midnight Ts values here, but your Gi cannot be a daily average because its G values would be much smaller. You mention noon time in the text, so I guess this must be it.

**AC:** These are noontime (1.30 pm) Gi estimates. We have modified in the caption of Figure 7 accordingly to read as 'Validation of STIC-TI derived noontime (1.30 pm) Gi estimates with respect to in-situ measurements in different ecosystems.

**RC 37**: Figures 10 & 11 look promising, but of course all these efforts only give you one value of the fluxes (around noon) for each day. How useful is this?

**AC:** Ideally diurnal latent heat fluxes are needed to estimate daily evapotranspiration (ET). In a cloudless day, the diurnal variation of latent heat flux between sunrise to sunset has sinusoidal pattern with its amplitude or peak corresponding to around noontime. Several studies revealed (Mallick et al, 2009, Bhattacharya et al, 2010) that the noontime latent heat flux computed through SEB model from polar orbiting remote sensing observation during a day can be converted to daily ET (in terms of mm depth of water) through sinusoidal integration. Validation of daily ET through such method is found to be close to in situ ET measurements. Therefore, validation of noontime LE and H fluxes and their accuracies are also very useful.

**RC 38**: Should Fig. 12 be part of the results section? Also why are you not plotting thermal intertia on the y-axis versus M? That would make more sense? Finally why are we not seeing the flattening off of the curve at large M values? And why is there so much scatter? Because of the different soil types at the different sites?

**AC:** Figure 12 is put in the discussion to highlight the reasons for reduced sensitivity of G due to Ts uncertainties. However, we agree with the suggestion to put thermal inertia in y-axis of Fig 12a. Overall, we have modified Figure 12 in the revised version which is as follows:

(i) We have merged Fig. 12 (a) and (b): thermal inertia in y-axis and noontime Ts in x-axis with color segregation according to M. The plot of thermal inertia versus Ts is flattening due to the effects of M and we see so much scatter due to different soil types across the sites.

(ii) We have merged Fig. 12 (c) and (d): G in y-axis and Amplitude (A) in x-axis with color segregation according to the thermal inertia.

The description in the text has been modified accordingly in line 641 to 647.

**RC 39**: Line 685-687: "...this is the first ever implementation of a coupled G-SEB model that does not require any empirical parameterization of aerodynamic and canopy-surface conductance". I am somewhat confused by this statement. I thought you were calculating g_A and g_S? I see equations for these in the Appendices?

**AC:** We are directly estimating ga and gs by solving the state equations (C1 to C4) as mentioned in the manuscript. While the estimation of ga does not require any parameterization of surface roughness and atmospheric stability, estimation of gs does not involve any empirical function or look-up table for biome or plant functional attributes. We have modified the sentence in line 756 to 759 for a better clarity.

**References:**

Bhattacharya, B.K., Mallick, K., Patel, N.K., & Parihar, J.S. (2010). Regional clear sky evapotranspiration over agricultural land using remote sensing data from Indian geostationary meteorological satellite. Journal of Hydrology, 387, 65-80.

Hillel, D (1982). Introduction to Soil Physics, San Diego, US, ISBN 9780123485205.

Mallick, K., et al. (2009). Latent heat flux estimation in clear sky days over Indian agroecosystems using noontime satellite remote sensing data. Agricultural and Forest Meteorology, 149 (10), 1646–1665, doi:10.1016/j.agrformet.2009.05.006.

Mihailovic, D. T., Kallos, G., Aresenic, I.D., Lalic, B., Rajkovic, B. and Papadopoulos, A. (1999). Sensitivity of soil surface temperature in a Force-Restore Equation to heat fluxes and deep soil temperature. International Journal of Climatology, 19, 1617-1632.

Schmid, H.P. (2002). Footprint modelling for vegetation atmosphere exchange studies: a review and perspective. Agricultural and Forest Meteorology 113, 159-183.

Vesala, T., Kljun, N., Rannik, U., Rinne, A. Sogachev, Markkanen, T., Sabelfeld, K., Foken, T. and Leclerc, M.Y. (2008). Flux and concentration footprint modelling: State of the art. Environmental Pollution, 152, 653-666

**Reviewer 2:**

**We thank Reviewer#2 for the detailed comments. All the line numbers correspond to the clean version of the manuscript. Please find the responses below.**

*RC 1*: The manuscript focusses on arid and semi-arid ecosystems where G can be a large component of the surface energy balance. Further, the authors have reported that the LE and G is sensitive to the LST observations. Over such arid or semi-arid ecosystems, it has been reported that the MYD21 LST (based on the Temperature Emissivity Separation algorithm) product might work better than the MYD11 product. May I suggest the authors to test if MYD21 product is better suited than the MYD 11 product for this coupled model?

**AC:** As per suggestions of reviewer, we have computed error statistics of STIC-TI flux estimates for four semi-arid and arid sites (Ind-Jai, Ind-Naw, US-Ton, AU-ASM) using MYD21 LST product. These error statistics for those four sites are now given in Table 5. The errors in flux estimates were found be less (2-8%) as compared to those with MYD11 LST product. The results of improvement are briefly narrated between line 598 to 602.

*RC 2*: Among the different components of surface energy fluxes, the in-situ observations of G and soil temperature are indeed strictly point observations when compared with LE, H or even Rn. In Equation 12, I think the ground observed soil temperature amplitude and satellite LST are combined towards developing the regression relationship. Is this valid considering that there can be varying soil conditions that can affect the amplitude of the soil temperature in a pixel?

**AC:** Soil temperature amplitude and noon-night LST as well as noontime surface albedo are taken from soils over varying soil types of Northern hemisphere (Indian site) and Southern hemisphere (Australian site), respectively. Based on the soil temperature amplitude model, STIC-TI estimate of G is independently validated over several US sites having varying soil types and vegetation cover for multiple years. This showed a good accuracy of G estimates. Therefore, we conclude that the proposed approach is valid for a wide range of soil, plant, and climate types.

*RC 3*: Further, satellite LST can represent soil temperature only over bare surface. Till what level of fraction vegetation cover, can we use LST to estimate the amplitude of soil temperature? Can the results in Section 4.1 be separated in terms of fc to see if there is any decrease in accuracy in estimating A with increasing vegetation cover?

**AC:** The present soil temperature amplitude model has been developed from noon-night LST difference as well as surface albedo for mean vegetation fraction around 0.5. We have now segregated the model bias for three different slabs of fractional vegetation cover (fc<0.3; 0.3≤fc≤0.5; fc>0.5). The results are discussed in line number 495 to 506.

*RC 4*: The STIC-TI model uses difference in LST between day and night. MODIS observations of LST can vary significantly in terms of viewing angle or increase in GIFOV due to the ground point being away from the nadir between day and night observations. Is this taken into consideration? Can the results be separated for LST observations made with similar viewing angles and different viewing angles during day and night?

**AC:** This is an excellent suggestion. We have done analysis on day-night view angle impact on deviations of STIC-TI flux estimates. The findings are described in line 582 to 590.

**RC 5**: As mentioned clearly, the method by Bastiaanssen (1998) was able to retrieve G with accuracy closer to that of the STIC-TI model. Is this due to the fact that both the models use LST, NDVI and albedo as key inputs to the 'G' model? Also, what benefit does the STIC-TI brings over the model by Bastiaanssen? We can look at the diurnal variation of LST to get more information about the ecosystem processes than the soil temperature amplitude, I think.

**AC:** This is a very good observation by the Reviewer followed by suggestion. The modelled soil temperature amplitude can be used to quantify diurnal variation of soil temperature and G fluxes which in turn can help to quantify diurnal latent heat fluxes and plant water stress.

Despite the comparable accuracy of current G estimates with the G model of Bastiaanssen et al (1998), the foundation of STIC-TI lies in the use of soil moisture characteristics with varying soil textural types which are known to influence the soil heat conductance and thereby G. Thus, the control of soil moisture on evaporation is explicitly included in STIC-TI as opposed to the semi-empirical G function of Bastiaanssen et al (1998). This explanation is included in line 683 to 687.

**RC 6**: There can be a considerable difference between the diurnal behavior of soil temperature at the surface (skin temperature) and soil temperature at 10 cm depth. Further, for modelling G, the soil skin temperature is the actual boundary condition. Can the authors please explain the reason for using soil temperature measured over 10 cm depth as the boundary condition rather than the skin temperature?

**AC:** Soil temperature measured at different depths within top 10 cm soil layer was averaged and considered as representative surface soil temperature (0 – 10 cm). Studies also showed that LST carries some signal beneath the skin of the surface(Johnston et al., 2022). We have clarified this in section 3.1.1.1 of the revised version of the manuscript in line 306 to 310.

**RC 7**: In addition, over two sites (Ind-Dha and AUS-Ade), soil temperature observations are done at or below 10 cm depth. How this has been used in the study in place of soil temperature up to 10 cm depth? When there are more than one soil temperature observations within 10 cm depth (AU-ASM, US-Ton and US-Var) were they averaged?

**AC:** Soil temperature observations at different depths within 10 cm were averaged (AU-ASM, USTon and US-Var). For Ind-Dha and AUS-Ade, single-depth (10 cm) soil temperature observation was used. We will state this explicitly in section 3.1.1.1 of the revised version in line 306 to 310.

**OTHER COMMENTS:**

**RC**: Lines 103-104, Please mention the key conclusions of the studies that you are referring to and explain why this study is needed.

**AC:** We have modified this portion in the revised manuscript mentioning the key conclusions from the previous studies and explained why the present study is needed in line 130 to 135.

**_RC_**: Lines 181-182 The fetch ratio of …90% of fetch area: This sentence is not clear. Are the authors trying to say that the assumed fetch area around the towers accounted for 90% of the energy fluxes? Figure 1 caption can be shortened and made precise.

**AC:** The flux footprint for EC towers in India varied from 500m-1km (Bhat et al. 2019). In present study, about 90% of fluxes came from an area within 500 m to 1 km from the EC tower. This represents that relative contribution of the vegetated land surface area to the flux is close to 90% (Schmid, 2002; Vesala et al, 2008). The remaining few percentage of flux is coming from an area beyond flux footprint.This is given in line 171 to 174.

We have shortened the Figure 1 caption by removing the text '(Supplement) map in PDF (Institute for Veterinary Public Health)' in the revised version.

**_RC_**: Lines 237-238: Non-availability of G over Indian sites is an issue. The authors have explained about the this at the end of the discussions section. But any studies supporting the use of unclosed energy budget from EC observations and lack of G observations will be helpful here.

**AC:** The findings from the relevant studies are included in line 721 to 728 and how present G-model can help in filling up the lack of G observations are included in line 743 to 747 in section 5.3

**_RC_**: Lines 268-269: The land surface emissivity was estimated from land cover types, atmospheric water vapour and …retrieval': I think MODIS emissivity is estimated from land cover type and anisotropy factor. The other variables mentioned here are used in the retrieval of LST with the split window algorithm. Please rewrite the sentence.

**AC:** Yes, we have rewritten the sentence in a new sub-section 3.1.3 in line number 432 to 433.

**_RC_**: Lines 271-272: How albedo was estimated from the white sky and black sky albedo available in the MCD43 product?

**AC:** Actual albedo is a value which is interpolated between white-sky and black-sky albedo as a function of the fraction of diffuse skylight which is itself a function of the aerosol optical depth.

We have added text in the new sub-section 3.1.3 (as also suggested by other reviewer) in line 435 to 439.

**RC**: Line 273-274: The line 'eight-day compositing … (MYD11A2)' appears little confusing. Do you mean that the compositing dates were obtained from MYD11 product? Please rewrite clearly.

**AC:** The 8-day average values of clear-sky LSTs retrieved from MYD11A2 data were used (Source: https://vip.arizona.edu/documents/viplab/MYD11A2.pdf). We have clarified this in line 256 – 258 of the revised manuscript.

**RC**: Lines 295-296: Is point (4) a contribution of this study? I think that is the nature of the TI model selected for coupling. Also, what is meant by 'moisture constants'?

**AC:** We agree that point (4) is not a contribution of this study but is a feature of STIC-TI. We have deleted this point to remove confusion.

We mean soil moisture characteristic limits. We have modified the text accordingly.

**RC**: Lines 303 – 304: Will the amplitude vary with each harmonic component? If yes, the A has to be replaced with An.

**AC:** In the present study, we have considered a single pair (noon-night corresponding to 1:30 pm and 1:30 am) of MODIS aqua LST data for amplitude. Therefore, 'A' is appropriate here. If we take multiple (n) LST pairs within a day, then 'An' will be appropriate.

**RC**: Lines 310-312: How $\varphi_n'$ can be taken as zero and k can be assumed to be one when noonnight data is considered? Please explain clearly. In equation (2), the first term in the inner bracket of sin term is '$n\omega t'$'. However, this becomes $\omega t'$ in eqn. (3). Are the symbols consistent here?

**AC:** Since we have considered a single pair (noon-night corresponding to 1:30 pm and 1:30 am) of MODIS aqua LST data for amplitude in the present study, the phase shift ($\varphi_n'$) is taken as zero and number of harmonics is taken as one (k=1) for estimating noontime $G_i$. We are not doing it here diurnally. The explanation is given in line 321 – 323.

The symbols are consistent between Eq. (2) and (3).

**RC**: Lines 331-332: It is mentioned that figure 2 contains theoretical and observed trajectories of soil temperature. However, only one curve is seen in the figure and I am not sure if it is theoretical or observed. Please check.

**AC**: This curve is based on observed soil temperature variation. We have clarified it in the caption of Figure 2.

**RC**: Lines 339-340: How TSTmin can be assumed to be closer top deep soil temperature as well as minimum soil temperature of other layers? Any data/studies to show this?

**AC:** This sentence has been deleted in the revised version to remove confusion.

*RC*: Lines 342-344: Have you used the OzFlux sites data to create Figure 2 or for the entire analysis?

**AC:** We have used OzFlux sites data to create Figure 2 to show example of diurnal variation of surface soil temperature.

*RC*: Line 438: 1.5 K repeats twice. Please check.

**AC:** The sensitivity was assessed by varying noon-time $T_S$ by ±0.5 K, ±1.0 K and ±1.5 K (keeping nighttime $T_S$ constant so that amplitude can vary automatically). We have modified the text accordingly in line 448 in the revised version.

*RC*: Line 579: What is 'clothesline effect'?

**AC:** The 'clothesline effect' is the effect on soil moisture when warm, dry air moves by advection through vegetation (e.g. field crops or forest). Close to the point of entry the warm air warms the soil-canopy system thus increasing the rate of evapotranspiration and drying the soil. Deeper into the vegetation canopy the temperature of moving air falls.

We have revised the text in the line 618 to 621 in the following way:

The intermittent EF spikes in the soil moisture dry down phase could be due to enhanced latent heat fluxes through evapotranspiration due to passage of warm air transported by micro-advection from surrounding vegetation causing additional evaporation than expected. This is known as the 'clothesline effect' which frequently occurs in semi-arid and arid ecosystems.

*RC*: Figure 12: Expect the relationship between M and thermal inertia in the first plot, there seem to be no direct link between other variables in the other plots. What is the key take home message from this figure?

**AC:** We have modified Figure 12 in the revised version considering comments from Reviewer 1 also. The revision is as follows:

> (i) We have merged Fig. 12 (a) and (b): thermal inertia in y-axis and noontime Ts in x-axis with color segregation according to M. The plot of thermal inertia versus Ts is flattening due to the effects of M and we see so much scatter due to different soil types across the sites.
> (ii) We have merged Fig. 12 (c) and (d): G in y-axis and Amplitude (A) in x-axis with color segregation according to the thermal inertia.

The description in the text has been modified accordingly in line 641 to 647.

*RC*: Line 627: …'range of 2-3 K with a standard deviation of 0.009'. I think the 0.009 corresponds to the standard deviation in surface emissivity in MODIS product. Please check.

**AC:** The standard deviations of errors in retrieved emissivity in bands 31 and 32 are 0.009, and the maximum error in retrieved LST values falls within 2-3 K. We have corrected this in the revised version in line 670 to 674.

Reference:

Dare-Idowu, O., Brut, A., Cuxart, J., Tallec, T., Rivalland, V., Zawilski, B., Ceschia, E. and Jarlan, L. (2021). Surface energy balance and flux partitioning of annual crops in southwestern France. Agricultural and Forest Meteorology, 308 – 309, 108529, https://doi.org/10.1016/j.agrformet.2021.108529.

Schmid, H.P. (2002). Footprint modelling for vegetation atmosphere exchange studies: a review and perspective. Agricultural and Forest Meteorology 113, 159-183.

Vesala, T., Kljun, N., Rannik, U., Rinne, A. Sogachev, Markkanen, T., Sabelfeld, K., Foken, T. and Leclerc, M.Y. (2008). Flux and concentration footprint modelling: State of the art. Environmental Pollution, 152, 653-666.

---

## Author Response (AR2)

**Author's response**

**We thank the reviewer for the additional suggestions in the revised version. The revisions are included and line numbers in the response corresponds to the 'track change' version of the manuscript.**

**Comment:** Though most of my comments are addressed properly, the comment that I raised on the variation of STIC-TI modelled fluxes with variation in MODIS sensor look angle has not been addressed completely. The authors have addressed it partially by mentioning that the flux estimates vary with the view angle of the sensor during daytime but not night time. Since the model needs day-night LST difference, both images could have been acquired with different sensor look angle. I asked the authors to check if the fluxes modelled by STIC-TI will be affected due to the sensor view angle changes between day and night observations.

**Reply**: Quantification of the modeled flux errors with respect to day-night view angle difference is done with additional analysis and is presented in Figure F in Appendix F. Relevant text is modified with new findings in section 4.4 between line 582 to 594.

**Comment:** Further, in the revised manuscript, Appendix-F is not really clear. What is meant by deviation? Instead of just presenting a histogram (which did not help at all), can the authors please quantify the errors in the modelled output with different look angles?

**Reply**: Quantification of the modeled flux errors with respect to day-night view angle difference is done with further analysis and is presented in Figure F in Appendix F. Relevant text is modified with new findings in section 4.4 between line 582 to 594.

**Comment:** It is assumed that MODIS aqua will collect data at 1:30 PM and 1:30 AM always. This may not be true always. This time can vary widely even by more than 30-45 minutes. I think this will affect the modelling of G through the mechanistic model. Can the authors please explain this more?

**Reply**: The standard deviations of MODIS Aqua day-night overpass time over study sites were found to be within 30-45 minutes. The possible impact of deviation of MODIS Aqua overpass time on LST and fluxes is explained in section 5.2 between line 727 to 731.

**Minor comments:**

**Line 39: SEB is not expanded in the first place of usage. Please expand.**

**Reply**: Expanded in line 32 in the abstract. It reads as follows:

One of the major undetermined problems in evaporation (ET) retrieval using thermal infrared remote sensing is the lack of a physically based ground heat flux (G) model and its integration within the surface energy balance (SEB) equation.

**Line 46: Replace 'was' with 'were'**

**Reply**: Corrected

**Lines 48-49: How overestimation of Rn will lead to over estimation of just LE? Overestimation of Rn need not only lead to over estimation of LE. It can also lead to over estimation of H. Can the authors explain this statement given in the manuscript?**

**Reply:** In the manuscript we stated that the overestimation of LE was associated with the overestimation of net available energy (i.e., $R_{Ni} - G_i$) and use of unclosed surface energy balance measurements.

It has two aspects. Firstly, net available energy is an important component for estimating LE through the Penman-Monteith equation. If an overestimation of net available energy leads to overestimation of LE by STIC-TI, then H will be obviously underestimated according to the complementary form of the Penman-Monteith equation. Secondly, the widespread lack of surface energy balance closure is well known, which leads to LE (H) being undermeasured (overmeasured) in the eddy covariance system. Therefore, the validation of overestimated LE (due to $R_{Ni} - G_i$) with undermeasured LE will lead to net overestimation in LE, and for H it will be vice-versa. However, to make it more clear, we corrected the sentence in the abstract (line 47 – 49) as follows:

Overestimation (underestimation) of $LE_i$ ($H_i$) was associated with the overestimation of net available energy ($R_{Ni} - G_i$) and use of unclosed SEB flux measurements in $LE_i$ ($H_i$) validation.

**Line 56: Will the TRISHNA mission provide both noon and night observations as needed by this model?**

**Reply**: Yes, it will. It is also applicable for the LSTM (Land Surface Temperature Monitoring) and SBG (Surface Biology and Geology). We modified the last sentence of the abstract as follows:

Findings from this parameter-sparse coupled G-ET model can make a valuable contribution to mapping and monitoring the spatiotemporal variability of ecosystem

water stress and evaporation using noon-night thermal infrared observations from future Earth Observation satellite missions such as TRISHNA, LSTM, and SBG.

**Line 73: Change 'By contrast' to 'in contrast'**

**Reply**: Corrected

**Line 77: Change 'remained' to 'remains'**

**Reply**: Corrected

**Lines 143 and 144: How such a small number of sites will represent the entire northern and the southern hemispheres? Please modify this suitably.**

**Reply**: We deleted the term 'representing'.

**Line 176: delete the word 'were' (The remaining percentage of fluxes originated from…)**

**Reply**: Corrected

**Line 191: leave a space between 3 trees ha–1. Also, is it only 3 trees per hectare?**

**Reply**: Corrected as 3 trees per hectare.

**Line 192: US-var is a (leave space between 'is' and 'a').**

**Reply**: Corrected

**Equation 3: What is t'? In the next equation it is mentioned as t.**

Reply: It is a typo error. It is 't' instead of t'. We have corrected it in equation 3.

**Line 298: Is $\Delta t$ a constant and held equal to 1.5? Then why is it necessary to calculate it as a function of fc?**

**Reply**: Murray and Verhoef (2007) initially proposed constant $\Delta t$ which is equal to 1.5. However, Maltese et al (2013) later modified as a function of fc. We have used $\Delta t$ as a function of fc. Text is modified accordingly in line 295 to 299.

**Figure 2: I had asked the same query previously. I think only the observed curve is mentioned in the figure but not the theoretical trajectory.**

**Reply:** Theoretical trajectory is published in literatures earlier, which we have mentioned. However, the observed curve is more important for the study to develop G model. Therefore, we have shown the observed curve.

**Lines 402 and 403: What is PMEB? Please expand the abbreviations in the first place of usage.**

**Reply**: Expanded in line 398 - 399.

**Line 497: 'slab' doesn't appear to be a right choice of word. You may use the word 'class'.**

**Reply**: Corrected.

**Lines 713 and 714: My same question regarding overestimation of H as raised in the abstract remains here too.**

**Reply**: Already explained above. To add better clarity, we added the following in line 717 – 721.

Since available energy is an important component for estimating LE through the PMEB equation, an overestimation of net available energy leads to an overestimation of LE by STIC-TI. Sensible heat flux will be consequently underestimated due to the complementary nature of the PMEB equation.